


# Ice injected into the tropopause by deep convection – Part 2: Over the Maritime Continent

Dion Iris-Amata[1], Dallet Cyrille[1], Ricaud Philippe[1], Carminati Fabien[2], Haynes Peter[3], and Dauhut Thibaut[4]

[1]CRNM, Meteo-France - CNRS, Toulouse, 31057, France
[2]Met Office, Exeter, Devon, EX1 3PB, UK
[3]DAMTP, University of Cambridge, Cambridge, CB3 0WA, UK
[4]Max Planck Institute for Meteorology, Hamburg, Germany

**Correspondence:** Iris-Amata Dion (iris.dion@meteo.fr)

**Abstract.** The amount of ice injected up to the tropical tropopause layer has a strong radiative impact on climate. In the tropics, the Maritime Continent (MariCont) region presents the largest injection of ice by deep convection into the upper troposphere (UT) and tropopause level (TL) (from results presented in the companion paper Part 1). This study focuses on the MariCont region and aims to assess the processes, the areas and the diurnal amount and duration of ice injected by deep

convection over islands and over seas using a $2° \times 2°$ horizontal resolution during the austral convective season of December, January and February. The model presented in the companion paper is used to estimate the amount of ice injected ($\Delta$IWC) up to the TL by combining ice water content (IWC) measured twice a day in tropical UT and TL by the Microwave Limb Sounder (MLS; Version 4.2), from 2004 to 2017, and precipitation (Prec) measurement from the Tropical Rainfall Measurement Mission (TRMM; Version 007) at high temporal resolution (1 hour). The horizontal distribution of $\Delta$IWC estimated from

Prec ($\Delta$IWC$^{Prec}$) is presented at $2° \times 2°$ horizontal resolution over the MariCont. $\Delta$IWC is also evaluated by using the number of lightnings (Flash) from the TRMM-LIS instrument (Lightning Imaging Sensor, from 2004 to 2015 at 1-h and $0.25°\times0.25°$resolutions). $\Delta$IWC$^{Prec}$ and $\Delta$IWC estimated from Flash ($\Delta$IWC$^{Flash}$) are compared to $\Delta$IWC estimated from the ERA5 reanalyses ($\Delta$IWC$^{ERA5}$) degrading the vertical resolution to that of MLS observations ($\langle\Delta IWC^{ERA5}\rangle$). Our study shows that, while the diurnal cycles of Prec and Flash are consistent to each other in timing and phase over lands and different

over offshore and coastal areas of the MariCont, the observational $\Delta$IWC range between $\Delta$IWC$^{Prec}$ and $\Delta$IWC$^{Flash}$ is small (to within 4 – 20% over land and to within 6 – 50% over ocean) in the UT and TL. The reanalysis $\Delta$IWC range between $\Delta$IWC$^{ERA5}$ and $\langle\Delta IWC^{ERA5}\rangle$ has been also found to be small in the UT (22 – 32 %) but large in the TL (68 – 71 %), highlighting the stronger impact of the vertical resolution on the TL than in the UT. Combining observational and reanalysis $\Delta$IWC ranges, the total $\Delta$IWC range is estimated in the UT between 4.17 and 9.97 mg m-3 (20 % of variability per study zone)

over land and between 0.35 and 4.37 mg m-3 (30% of variability per study zone) over sea, and, in the TL, between 0.63 and 3.65 mg m-3 (70% of variability per study zone) over land and between 0.04 and 0.74 mg m-3 (80% of variability per study zone) over sea. Finally, from IWC$^{ERA5}$, Prec and Flash, this study highlights 1) $\Delta$IWC over land has been found larger than $\Delta$IWC over sea, and 2) the Java Island is the area of the largest $\Delta$IWC in the UT (7.89 – 8.72 mg m-3 daily mean).



*Copyright statement.* TEXT

# 1 Introduction

In the tropics, water vapour (WV) and ice cirrus clouds near the cold point tropopause (CPT) have a strong radiative effect on climate (Stephens et al., 1991) and an indirect impact on stratospheric ozone (Stenke and Grewe, 2005). WV and water ice crystals are transported up to the tropopause layer by two main processes: a three-dimentional large-scale slow process (3-m month -1), and a small scale fast convective process (diurnal timescale) (Fueglistaler et al., 2009; Randel and Jensen, 2013). Many studies have already shown the impact of convective processes on the hydration of the atmospheric layers from the upper troposphere (UT) to the lower stratosphere (LS) (Liu and Zipser, 2005; Dauhut et al., 2018; Dion et al., 2019). However, the amount of total water (WV and ice) transported by deep convection up to the tropical UT and LS is still not well understood. The vertical distribution of total water in those layers is constrained by thermal conditions of the CPT (Randel et al., 2006). During deep convective events, Dion et al. (2019) have shown that air masses transported up to 146 hPa in the UT and up to 100 hPa in the tropopause layer (TL) have ice to total water ratios of more than 50% and 70%, respectively and that ice in the UT is strongly spatially correlated with the diurnal increases of deep convection, while WV is not. Dion et al. (2019) hence focused on the ice phase of total water to estimate the diurnal amount of ice injected into the UT and the TL over convective tropical areas showing that it is larger over land than over ocean, with maxima over lands of the Maritime Continent (MariCont), the region including Indonesian islands. For these reasons, the present study is focusing on the MariCont region in order to better understand small-scale processes impacting the diurnal injection of ice up to the TL.

The tropical distribution of the convection follows the Hadley cells and the Walker circulation. A method to estimate the amount of ice injected into the UT and up to the TL over convective areas and during convective seasons has been proposed by Dion et al. (2019). This method provides an estimation of the amplitude of the diurnal cycle of ice in those layers using the twice daily Ice Water Content (IWC) measurements from the Microwave Limb Sounder (MLS) instrument and the full diurnal cycle of precipitation (Prec) measured by the Tropical Rainfall Measurement Mission (TRMM) instrument, at one hour resolution. The method first focuses on the increasing phase of the diurnal cycle of Prec (peak to peak from the diurnal Prec minimum to the diurnal Prec maximum) and shows that the increasing phase of Prec is consistent in time and in amplitude with the increasing phase of the diurnal cycle of deep convection, over tropical convective zones and during convective season. The amount of ice ($\Delta$IWC) injected into the UT and the TL is estimated by relating IWC measured by MLS during the growing phase of the deep convection to the increasing phase of the diurnal cycle of Prec. Dion et al. (2019) conclude that deep convection over the MariCont region is the main process impacting the increasing phase of the diurnal cycle of ice in those layers.

The MariCont region is one of the main convective center in the tropics with the wettest troposphere and the coldest and driest tropopause (Ramage, 1968; Sherwood, 2000; Hatsushika and Yamazaki, 2001). Yang and Slingo (2001) have shown that over the Indonesian area, the phase of the convective activity diurnal cycle drifts from land to coastlines and to offshore areas. A comprehensive work has been done around the study of the diurnal cycle of precipitations and convection over the



MariCont, but the diurnal cycle of ice injected by deep convection up to the TL over this region is still not well understood. Millán et al. (2013) have tentatively evaluated the upper tropospheric diurnal cycle of ice from Superconducting Submillimeter-Wave Limb-Emission Sounder (SMILES) measurements over the period 2009-2010 but without differentiating land and sea

over the MariCont, which caused their analysis to show little diurnal variation over that region. Dion et al. (2019) have 1) highlighted that the MariCont must be considered as two separate areas: the MariCont land (MariCont_L) and the MariCont ocean (MariCont_O), with two distinct diurnal cycles of the Prec and 2) estimated the amount of ice injected in the UT and the TL. Over these two domains, it has also been shown that convective processes are stronger over MariCont_L than over MariCont_O. Consequently, the amount of ice injected in the UT and the TL is greater over MariCont_L than over

MariCont_O. Considering a higher horizontal resolution over small islands and seas of the MariCont and investigating other proxies of deep convection, the authors were expected a better characterisation of the amount of ice injected up to the TTL.

  Building upon the results of Dion et al. (2019), the present study is addressing the evaluation of ΔIWC at a resolution of 2° × 2° over 5 islands (Sumatra, Borneo, Java, Sulawesi and New Guinea) and 5 seas (West Sumatra Sea, Java Sea, China Sea, North Australia Sea, and Bismark Sea) of the MariCont during convective season (December, January and February, hereafter

DJF) from 2004 to 2017. ΔIWC will be first estimated from Prec measured by TRMM-3B42. A sensitivity study of ΔIWC based on the number of flashes (Flash) detected by the TRMM Lightning Imaging Sensor (TRMM-LIS), an alternative proxy for deep convection as shown by Liu and Zipser (2009), is also proposed. Finally, we will use IWC calculated by the ERA5 reanalyses from 2005 to 2016 to estimate ΔIWC in the UT and the TL over each study zone that will be compared it to ΔIWC estimated from Prec and Flash.

The observational datasets used in our study are presented in Sect. 2. Method is recalled in Sect. 3. The amount of ice (ΔIWC) injected up to the TL estimated from Prec is evaluated in Sect. 4. Diurnal cycles of Prec and Flash are compared to each other over different areas of the MariCont in Sect. 5. Results of the estimated ΔIWC injected up to the UT and the TL over five islands and five seas of the MariCont are presented and compared with the ERA5 reanalyses in Sect. 6. Results are discussed in Sect. 7, and conclusions are drawn in Sect. 8. This paper contains many abbreviations and acronyms. To facilitate

reading, they are compiled in the Acronyms list.

## 2 Datasets

This section presents the instruments and the reanalyses used for this study.

### 2.1 MLS Ice Water Content

The Microwave Limb Sounder (MLS, Version 4.2) instrument on board the NASA's Earth Observing System (EOS) Aura

platform (Livesey et al., 2017) launched in 2004 provides ice water content ($IWC^{MLS}$, mg m-3) measurements at 146 hPa (in the UT) and at 100 hPa (in the TL). MLS follows a sun-synchronous near-polar orbit, completing 233 revolution cycles every 16 days, with daily global coverage every 14 orbits. The instrument is crossing twice a day the equator at fixed time, measuring $IWC^{MLS}$ at 01:30 local time (LT) and 13:30 LT. The horizontal resolution of $IWC^{MLS}$ measurements is $\sim$ 300 and 7 km





along and across the track, respectively. The vertical resolution of IWC$^{MLS}$ is 4 and 5 km at 146 and 100 hPa, respectively.
Since the averaging kernels of $IWC^{MLS}$ are not provided by MLS, we will use an unitary triangular function centered at 146 and 100 hPa, having a width at half-maximum of 4 and 5 km, respectively to represent the averaging Kernels of MLS IWC at 146 and 100 hPa (see section 7.2). In our study, high spatial resolution study is now possible because we consider 13 years of MLS datasets, allowing to average the IWC$^{MLS}$ measurements within the bins of horizontal resolution of $2° \times 2°$ ($\sim$ 230 km). We select IWC$^{MLS}$ during all austral convective seasons DJF between 2004 and 2017.

## 2.2 TRMM-3B42 Precipitation

The Tropical Rainfall Measurement Mission (TRMM) has been launched in 1997 and has been able to provide measurements of Prec until 2015. TRMM is composed by five instruments, three of them are complementary sensor rainfall suite (PR, TMI, VIRS). TRMM had an almost circular orbit at 350 km altitude height performing a complete revolution in one and half hour. The 3B42 algorithm product (TRMM-3B42) (version V7) has been created to extend the precipitation product through 2019.
Thus, the TRMM-3B42, a Prec dataset is based on TRMM observations from 1997 to 2015 and provides Prec data from 1997 to 2019 at a $0.25° \times 0.25°$ ($\sim$ 29.2 km) horizontal resolution, extending from $50°$ S to $50°$ N (https://pmm.nasa.gov/data-access/downloads/trmm, last access: April 2019). Prec from TRMM-3B42 products depend on input from microwave and IR sensors (Huffman et al., 2007) not differentiate between stratiform and convective precipitations. In our study, Prec from TRMM-3B42 is selected over the austral convective seasons (DJF) from 2004 to 2017 and averaged to a horizontal grid of $2°$
$\times 2°$ to be compared to IWC$^{MLS}$. As the TRMM orbit precesses, the diurnal cycle of Prec averaged over the study period is calculated with a 1-h temporal resolution.

## 2.3 TRMM-LIS number of Flashes

The Lightning Imaging Sensor (LIS) aboard of the TRMM satellite measures several parameters relative to lightnings. LIS was using a Real-Time Event Processor (RTEP) that discriminates lightning event from Earth albedo light. The instrument
in itself was composed by a grid of $128 \times 128$ detectors allowing to observe a point within 90 seconds with a temporal resolution of 2 milliseconds. A lightning event corresponds to the detection of a light anomaly on a pixel representing the most fundamental detection of the sensor. After a spatial and temporal processing, the sensor was able to characterize a flash from several detected events. LIS horizontal resolution is provided at $0.25° \times 0.10°$. LIS is thus able to provide the number of flashes (Flash) measured. The LIS instrument performed measurements between 1 January 1998 and 8 April 2015. To be as consistent
as possible to the MLS and TRMM-3B42 period of study, we are using LIS measurements during DJF from 2004 to 2015. LIS spatial resolution varies between 3 km at nadir and 6 km off-nadir. The observation range of the sensor is between $38°$ N and $38°$ S. As LIS is on the TRMM platform, with an orbit that precesses, Flash from LIS can be averaged to obtain the full 24-h diurnal cycle of Flash over the study period with a 1-h temporal resolution. In our study, Flash measured by LIS is studied at $0.25° \times 0.25°$ horizontal resolution to be compared to Prec from TRMM-3B42.





### 2.4 ERA5 Ice Water Content

The European Centre for Medium-range Weather Forecasts (ECMWF) Reanalysis 5, known as ERA5, replaces the ERA-Interim reanalyses as the fifth generation of the ECMWF reanalysis providing global climate and weather for the past decades (from 1979) (Hersbach, 2018). ERA5 provides hourly estimates for a large number atmospheric, ocean and land surface quantities and covers the Earth on a 30 km grid with 137 levels from the surface up to a height of 80 km. Cloud ice water content from ERA5 reanalyses ($IWC^{ERA5}$) comes from the combination of a large amount of global historical observations, advanced modelling and data assimilation systems (https://cds.climate.copernicus.eu/cdsapp!/dataset/reanalysis-era5-pressure-levels-monthly-means?tab=form, last access: July 2019). The present study uses the $IWC^{ERA5}$ at 100 and 150 hPa averaged over DJF from 2005 to 2016 with one hour temporal resolution. $IWC^{ERA5}$ have been degraded along the vertical at 100 and 150 hPa ($\langle \Delta IWC^{ERA5} \rangle$) consistently with the MLS vertical resolution of $IWC^{MLS}$ (5 and 4 km at 100 and 146 hPa, respectively) using an unitary triangular function, in the absence of IWC averaging kernels by MLS (see section 7.2). $IWC^{ERA5}$ and $\langle \Delta IWC^{ERA5} \rangle$ will be both considered in this study. $IWC^{ERA5}$, initially provided in kg kg-1, has been converted into mg m-3 using the temperature provided by ERA5 in order to be compared with MLS IWC observations.

### 2.5 NOAA Winds

Because wind provided by ERA5 is not available at 100 and 150 hPa, our study uses the wind datasets from the National Centers For Environmental Prediction (NCEP) Global Data Assimilation System (GDAS), initialized analyses, provided by the National Oceanic and Atmospheric Administration (NOAA). NOAA provides vertical distribution of daily Est-West and North-South wind components in the range between -75 to 107 m s-1, from 1997 to 2019 (https://www.esrl.noaa.gov/psd/thredds/catalog/Datasets/ncep/catalog.html, last access: 8 July 2019). 12 vertical levels are available from 50 to 1000 hPa at global scale. Our study selects the daily mean of wind speed and direction at 100 and 150 hPa in DJF from 2004 to 2017.

## 3 Methodology

This section summarizes the method developed by Dion et al. (2019) to estimate $\Delta$IWC, the amount of ice injected into the UT and the TL. Dion et al. (2019) have presented a model relating Prec (as proxy of deep convection) from TRMM to $IWC^{MLS}$ over tropical convective areas during austral convective season DJF. The $IWC^{MLS}$ value measured by MLS during the growing phase of the convection (at x = 01:30 LT or 13:30 LT) is compared to the Prec value at the same time x in order to define the correlation coefficient (C) between Prec and $IWC^{MLS}$, as follows:

$$C = \frac{Prec_x}{IWC_x^{MLS}} \tag{1}$$





The diurnal cycle of IWC estimated ($IWC^{est}(t)$) can be calculated by using $C$ applied to the diurnal cycle of Prec (Prec(t)), where $t$ is the time, as follows:

$$IWC^{est}(t) = Prec(t) \times C \tag{2}$$

The amount of IWC injected up to the UT or the TL ($\Delta\mathrm{IWC}^{Prec}$) is defined by the difference between the maximum of IWC$^{est}$ (IWC$^{est}_{max}$) and its minimum (IWC$^{est}_{min}$).

$$\Delta IWC^{Prec} = C \times (Prec_{max} - Prec_{min}) = IWC^{est}_{max} - IWC^{est}_{min} \tag{3}$$

where $Prec_{max}$ and $Prec_{min}$ are the diurnal maximum and minimum of Prec, respectively. Figure 1 illustrates the relationship between the diurnal cycle of Prec and the two MLS measurements at 01:30 LT and 13:30 LT. The growing phase of the
convection is defined as the period of increase in precipitation from $Prec_{min}$ to $Prec_{max}$. The amplitude of the diurnal cycle is defined by the difference between $Prec_{max}$ and $Prec_{min}$. In Fig. 1, because the growing phase of the convection illustrated is happening during the afternoon, only the MLS measurement at 13:30 LT is used in the calculation of $\Delta$IWC. IWC at 01:30 LT is not used in that case.

## 4    Horizontal distribution of ΔIWC estimated from Prec over the MariCont

### 4.1    Prec from TRMM related to IWC from MLS

In order to identify the main areas of injection of ice in the TL over the MariCont, Figure 2 presents different parameters associated to this area: a) the name of the main islands and seas over the MariCont, b) the elevation (http://www.soda-pro.com/web-services/altitude/srtm-in-a-tile, last access: June 2019), c) the daily mean of Prec at $0.25° \times 0.25°$ horizontal resolution, d) the hour of the diurnal maxima of Prec at $0.25° \times 0.25°$ horizontal resolution, and e) the daily mean
($I\bar{W}C = (IWC_{01:30} + IWC_{13:30}) \times 0.5$) of IWC$^{MLS}$ at 146 hPa at $2° \times 2°$ horizontal resolution. Several points need to be highlighted. Daily means of Prec over lands and coastal parts are higher than over oceans (Fig. 2c). Areas where the daily mean of Prec is maximum are usually surrounding the highest elevation over lands (e.g. over the New Guinea) and near coastal areas (North West of Borneo in the China Sea and South of Sumatra in the Java Sea) (Fig. 2b and c). Qian (2008) explained that high precipitation is mainly concentrated over lands in the MariCont because of the strong sea-breeze convergence, combined
with the mountain–valley winds and cumulus merging processes. The diurnal maximum of Prec over land is observed between 18:00 LT and 00:00 LT (Fig. 2d) whereas, over coastal parts, it is in the early morning before 05:00 LT. Over seas, the time of the diurnal maximum varies as a function of the region. Java Sea and North of Australia Sea present maxima around 13:30 LT while the west Sumatra Sea and the Bismark Sea show maxima around 01:30 LT. Amplitudes of the diurnal cycles of Prec over the MariCont will be detailed as a function of islands and sea in section 5. The location of the largest concentration of
IWC (3.5 – 5.0 mg m-3, Fig. 2c) is consistent with that of Prec ($\sim 12 - 16$ mm day-1) over the West Sumatra Sea, and over



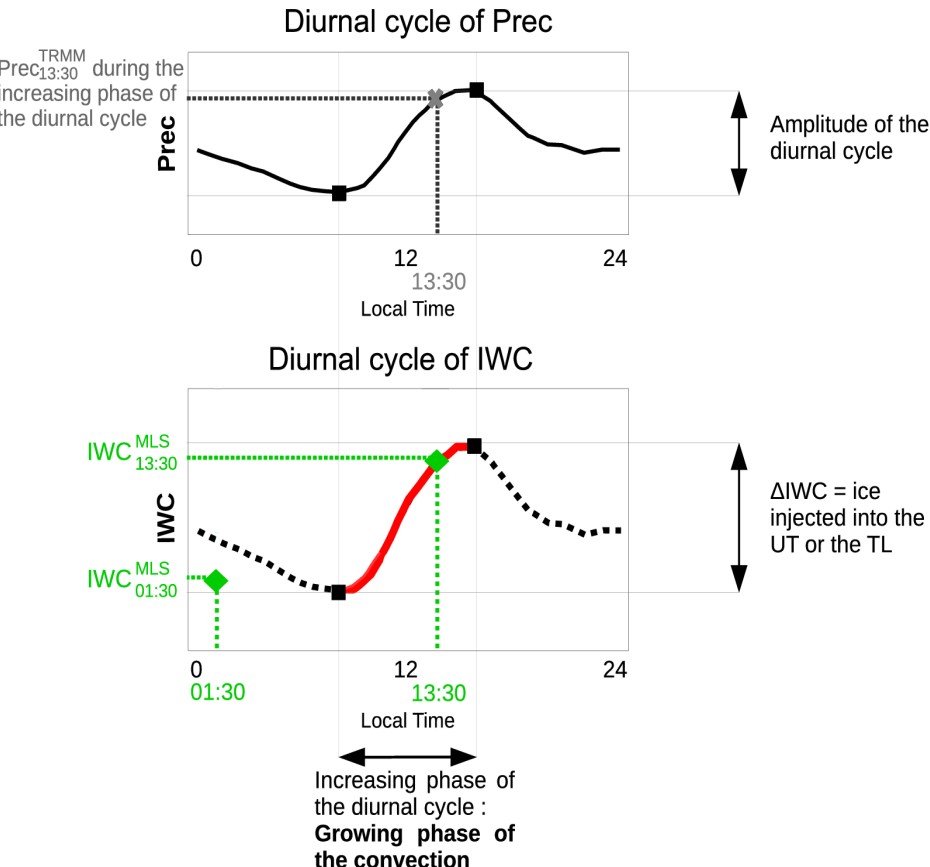

**Figure 1.** Illustration of the model used to estimate the amount of ice ($\Delta$IWC) injected into the UT or the TL. Diurnal cycle of a proxy of deep convection (Prec) (a), diurnal cycle of ice water content (IWC) estimated from diurnal cycle of the proxy of deep convection (b). In red line, the increasing phase of the diurnal cycle. In black dashed line, the decreasing phase of the diurnal cycle. The green diamonds are the two $IWC^{MLS}$ measurements from MLS. Grey thick cross represents the measurement of Prec during the growing phase of the convection ($Prec_x$), used in the model. Maximum and minimum of the diurnal cycles are represented by black squares. Amplitude of the diurnal cycle is defined by the differences between the maximum and the minimum of the cycle.

the South of Sumatra island. However, over North Australia seas (including the Timor Sea and the Arafura Sea), we observed large differences between Prec low values (4 – 8 mm day-1) and IWC large concentrations (4 – 7 mg m-3).



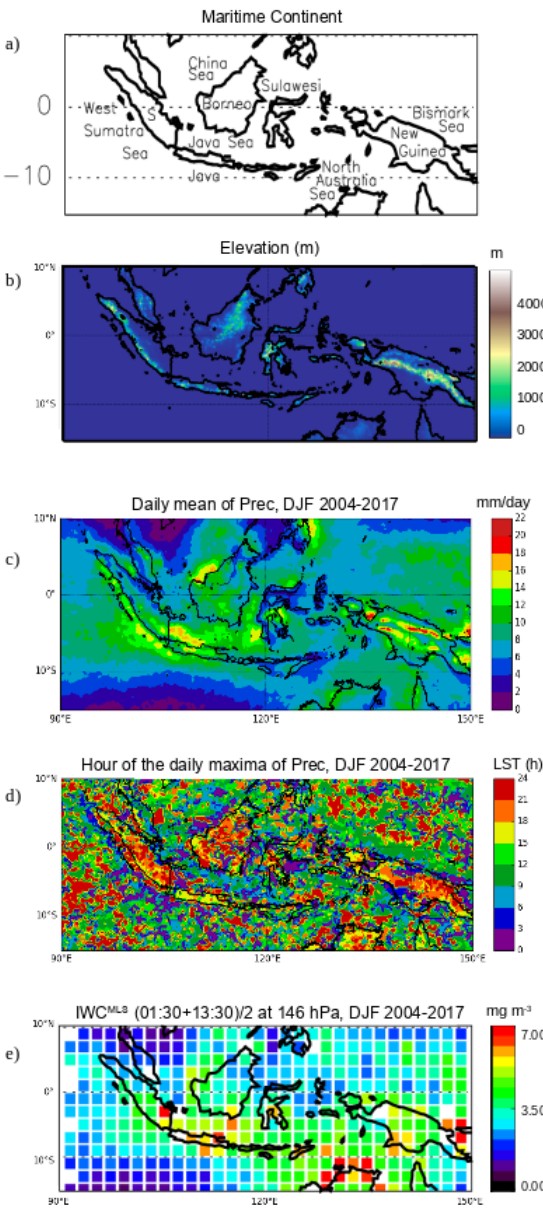

**Figure 2.** Main islands and seas of the MariCont (S is for Sumatra) (a), elevation from Solar Radiation Data (SoDa) (b); daily mean of Prec measured by TRMM over the Maritime Continent, averaged over the period of DJF 2004-2017 (c), hour (local solar time (LST)) of the diurnal maxima of Prec over the MariCont (d); daily mean (01:30 LT + 13:30 LT)/2 of IWC$^{MLS}$ at 146 hPa from MLS over the MariCont averaged over the period of DJF 2004-2017 (e).





## 4.2 Convective processes compared to IWC measurements

Although TRMM horizontal resolution is $0.25° \times 0.25°$, we require information at the same resolution that MLS IWC. From
the diurnal cycle of TRMM Prec measurements, each duration of the increasing phase of Prec can be known for each $2° \times 2°$ pixel. The duration of the growing phase of the convection can be defined from Prec over each pixel. Figures 3a and b present the anomaly (deviation from the mean) of Prec measured by TRMM–3B42 over the MariCont at 01:30 LT and 13:30 LT, respectively, only over pixels when the convection is in the growing phase. The anomaly of IWC measured by MLS over the MariCont is shown in Figs. 3c and d, over pixels when the convection is in the growing phase at 01:30 LT and 13:30 LT,
respectively. Note that some pixels can be present both in Figs. 3a and b (Figs. 3c and d, respectively) only when the onset of the convection is before 01:30 LT and the end is after 13:30 LT or when the onset of the convection is before 13:30 LT and the end is after 13:30 LT.

     The Prec anomaly at 01:30 LT and 13:30 LT varies between -0.15 and +0.15 mm h-1. At 13:30 LT, the growing phase of the convection is mainly over lands while, at 01:30 LT, the growing phase of the convection is mainly over seas and coastlines. The
strongest Prec anomaly (+0.15 mm h-1) at 13:30 LT is found over the Java island, while the strongest Prec anomaly at 01:30 LT is found over coastlines and seas close to the coasts such as the West Sumatra Sea. The IWC anomaly at 13:30 LT and 01:30 LT varies between -3 and +3 mg m-3. The strongest value of IWC anomaly at 13:30 LT is found over Java, while the strongest values of IWC anomaly at 01:30 LT is found over coastlines and seas close to the coasts, such as the North Australia Sea, Java Sea, China Sea and coast around the New Guinea. Three types of areas can be distinguished from Fig. 3: i) area where Prec
and IWC anomalies have the same sign (positive or negative either at 01:30 LT or 13:30 LT) (e.g. over Java, Borneo, Sumatra, Java Sea and coast of Borneo or the China Sea); ii) area where Prec anomaly is positive and IWC anomaly is negative (e.g. over West Sumatra Sea); and iii) area where Prec anomaly is negative and IWC anomaly is positive (e.g. over the North Australia Sea). Convective processes associated to these three types of areas over islands and seas of the MariCont are discussed in Sect. 6.

## 4.3 Horizontal distribution of ice injected into the UT and TL estimated from Prec

From the model developed in Dion et al. (2019) based on Prec from TRMM–3B42 and IWC from MLS and synthesized in section 2.4, we can calculate the amount of IWC injected ($\Delta$IWC) at 146 hPa (UT, Figure 4a) and at 100 hPa (TL, Figure 4b) by deep convection over the MariCont. In the UT, the amount of IWC injected over land is larger (> 10 – 20 mg m-3) than over seas (< 10 mg m-3). South of Sumatra, Sulawesi, North of New Guinea and North of Australia present the largest amounts of
$\Delta$IWC over lands (15 – 20 mg m-3). Java Sea, China Sea and Bismark Sea present the largest amounts of $\Delta$IWC over seas (7 – 15 mg m-3). West Sumatra Sea and North Australia Sea present low values of $\Delta$IWC (< 2 mg m-3). We can note that the anomalies of Prec and IWC during the growing phase over West Sumatra Sea are positive (< 0.15 mg m-3, Fig. 3a and b and > 2.50 mg m-3, Fig. 3c and d, respectively). In the TL, the maxima (up to 3 mg m-3) and minima (down to 2 – 3 mg m-3) of $\Delta$IWC are located within the same pixels as in the UT, although 3 – 4 times lower than in the UT. The decrease of $\Delta$IWC



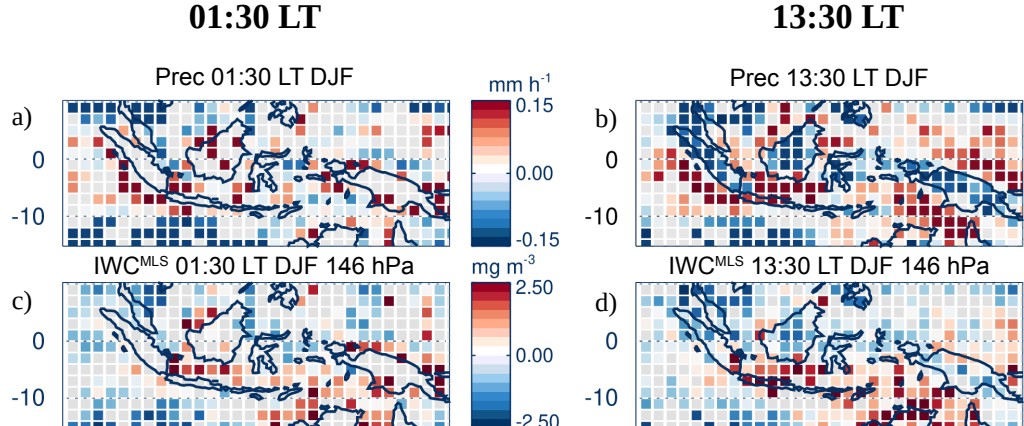

**Figure 3.** Anomaly (deviation from the mean) of Prec (a-b) and Ice Water Content (IWC$^{MLS}$) (c-d), at 01:30 LT (left) and at 13:30 LT (right) over pixels where 01:30 LT and 13:30 LT are during the growing phase of the convection, respectively averaged over the period of DJF 2004-2017.

with altitude is more important over lands (by a factor 6) than over sea (by a factor 3). Convective processes associated to these lands and seas are further discussed in Sect. 6.

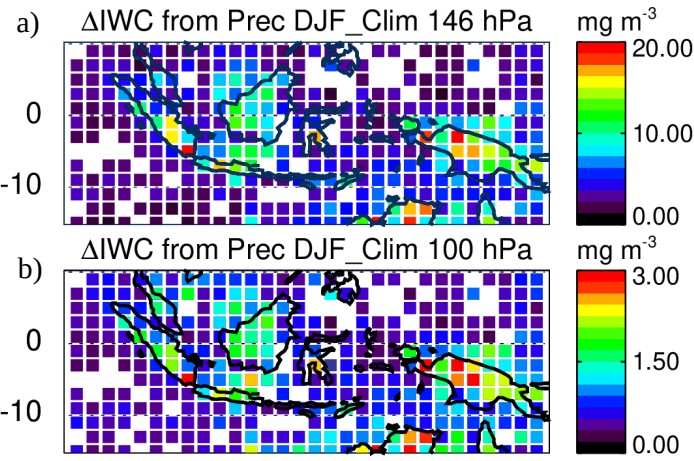

**Figure 4.** Daily amount of ice injected (ΔIWC) up to the UT (a) and up to the TL (b) estimated from Prec, averaged during DJF 2004-2017.

In order to better understand the impact of deep convection on the strongest ΔIWC injected per pixel into the UT and the TL, isolated pixels selected in Fig. 4 are presented separately in Figure 5a and f. This Figure shows the diurnal cycles of Prec





in four pixels selected for their large ΔIWC in the UT (≥ 15 mg m-3, Fig. 5b, c, d, e), and the diurnal cycle of Prec in four

pixels selected for their low ΔIWC in the UT (but large enough to observe diurnal cycles of Prec between 2.0 and 5.0 mg m-3, Fig. 5g, h, i, j). Pixels with low values of ΔIWC over lands (Figs. 5g, h and i) present small amplitude of diurnal cycles of Prec (∼ +0.5 mm h-1), with maxima between 15:00 LT and 20:00 LT and minima around 11:00 LT. The pixel with low value of ΔIWC over sea (Fig. 5j) presents an almost null amplitude of the diurnal cycle of Prec with low value of Prec all day long (∼ 0.25 mm h-1). Pixels with large values of ΔIWC over lands (Fig. 5b, c, d, e) present longer duration of the increasing phase of

the diurnal cycle (from ∼ 09:00 LT to 20:00 – 00:00 LT) than the increasing phase of Prec diurnal cycle over pixels with low values of ΔIWC (from 10:00 LT to 15:00 – 19:00 LT). More precisely, pixels labeled 1 and 2 over New Guinea (Fig. 5d and e) and the pixel over South of Sumatra (Fig. 5c) show amplitude of diurnal cycle of Prec reaching 1.0 mm h-1, while the pixel over North Australia (Fig. 5b) presents lower amplitude of diurnal cycle of Prec (0.5 mm h-1).

IWC measured by MLS during the growing phase of deep convection and the diurnal cycle of IWC estimated from Prec

are also shown on Fig. 5. For pixels with large values of ΔIWC, IWC observed by MLS is between 4.5 and 5.7 mg m-3. For pixels with low values of ΔIWC, IWC observed by MLS is found between 1.9 and 4.7 mg m-3. To summarize, large values of ΔIWC are observed in combination to i) longer growing phase of deep convection (> 9 hours), ii) high value of IWC (> ∼4.5 mg m-3) at 13:30 LT over land and 01:30 LT over seas, and/or iii) large diurnal amplitude of Prec (> 0.5 mm h-1). This shows that ΔIWC is strongly correlated with the shape of the diurnal cycle of Prec.

In the next section, we estimate ΔIWC using another proxy of deep convection, namely Flash measurements from LIS.

## 5 Relationship between diurnal cycle of Prec and Flash over MariCont land and sea

Lightnings are created into cumulonimbus clouds when the potential energy difference is large between the base and the top of the cloud. Lightnings can appear at the advanced stage of the growing phase of the convection and during the mature phase of the convection. For these reasons, in this section, we use Flash measured from LIS during DJF 2004-2015 as another proxy

of the deep convection in order to estimate ΔIWC ($\Delta$IWC$^{Flash}$) and check the consistency with ΔIWC obtained with Prec ($\Delta$IWC$^{Prec}$).

### 5.1 Flash distribution over the MariCont

Figure 6a presents the daily mean of Flash in DJF 2004-2015 at 0.25° × 0.25° horizontal resolution. Over land, Flash can reach a maximum of 10-1 flashes per day per pixel while, over seas, Flash are less frequent (∼ 10-3 flashes per day per pixel).

When compared to the distribution of Prec (Fig. 2c), maxima of Flash are found over the same areas as maxima of Prec (Java, East of Sulawesi coast, Sumatra and North Australia lands). Over Borneo and the New Guinea, coastlines present more Flash (∼10-2 flashes per day) than inland (∼10-3 flashes per day). Differences between Flash and Prec distributions are found over North Australia Sea, with relatively large number of Flash (∼ >10-2 flashes per day) compared to low Prec (4 – 10 mm day-1) (Fig. 2c), and over the New Guinea where the number of Flash is relatively low (∼10-2- 10-3 flashes day-1) while Prec is high

(∼14 – 20 mm day-1). Figure 6b shows the hour of the Flash maxima. Over land, the maximum of Flash is between 15:00



**Figure 5.** a) and f) location of $2° \times 2°$ pixels where $\Delta$IWC have been found higher than 15 mg m-3 (in Fig. 4) and where $\Delta$IWC have been found between 2 and 5 mg m-3 (in Fig. 4), respectively. Diurnal cycle of Prec: (b, c, d, e) over 4 pixels where $\Delta$IWC have been found higher than 15 mg m-3 (in Fig. 4), (g, h, i, j) over 4 pixels where $\Delta$IWC have been found between 2 and 5 mg m-3 (in Fig. 4), during DJF 2004-2017. The Diamond is IWC$^{MLS}$ measured by MLS during the increasing phase of the convection.





LT and 19:00 LT, slightly earlier than the maximum of Prec (Fig. 2c) observed between 16:00 LT and 24:00 LT. Coastal areas present similar hours of maximum of Prec and Flash, i.e between 00:00 LT and 04:00 LT although, over the West Sumatra Coast, diurnal maxima of both Prec and Flash happen 1–4 hours earlier (from 23:00-24:00 LT) than those of other coasts.

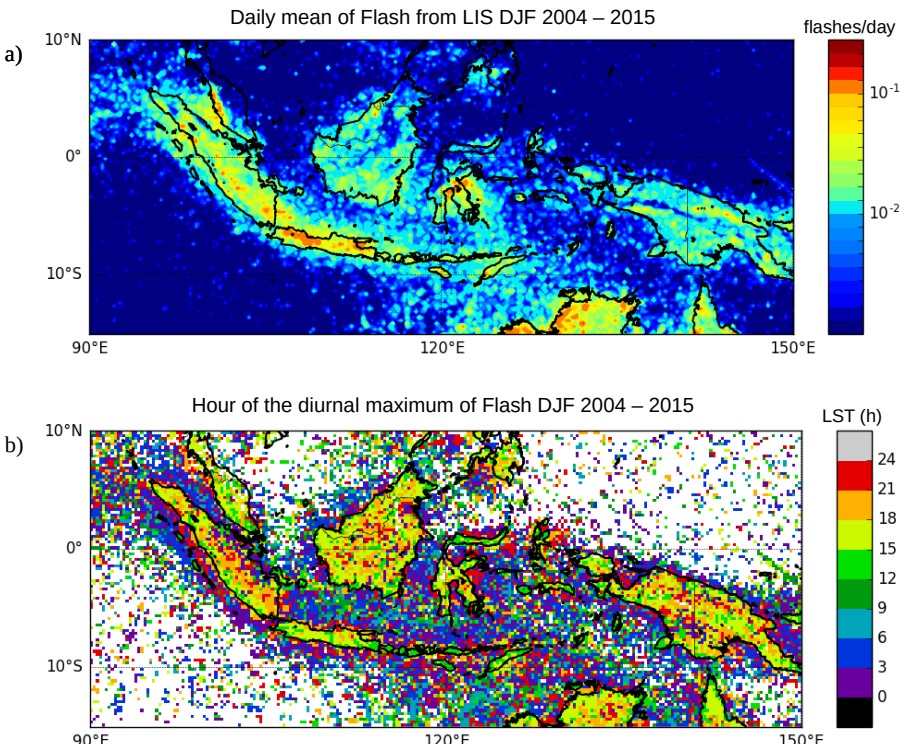

**Figure 6.** Daily mean of Flash measured by LIS averaged over the period DJF 2004-2015 (a); Hour (local solar time (LST)) of the diurnal maximum of Flash (b).

## 5.2 Prec and Flash diurnal cycles over the MariCont

This section compares the diurnal cycle of Flash with the diurnal cycle of Prec in order to assess the potential for Flash to be used as a proxy of deep convection over lands and seas of the MariCont. Diurnal cycles of Prec and Flash over the MariCont lands, offshore and coastlines (MariCont_L, MariCont_O MariCont-C, respectively) are shown in Figs. 7a–c, respectively. Within each $0.25° \times 0.25°$ bin, land/ocean/coast filters were applied from the Solar Radiation Data (SoDa, http://www.soda-pro.com/web-services/altitude/srtm-in-a-tile). MariCont-C is the average of all coastlines defined as 5 pixels

over the sea from the lands limits. The MariCont_O is the average of all offshore pixels defined as sea pixels excluding 10 pixels over the sea from the land coasts, thus coastline pixels are excluded. MariCont_L is the area of all land pixels.

Over land, during the growing phase of the convection, Prec and Flash start to increase at the same time (10:00 LT – 12:00 LT) but Flash reaches a maximum earlier (15:00 LT – 16:00 LT) than Prec (17:00 LT – 18:00 LT). This is consistent with the





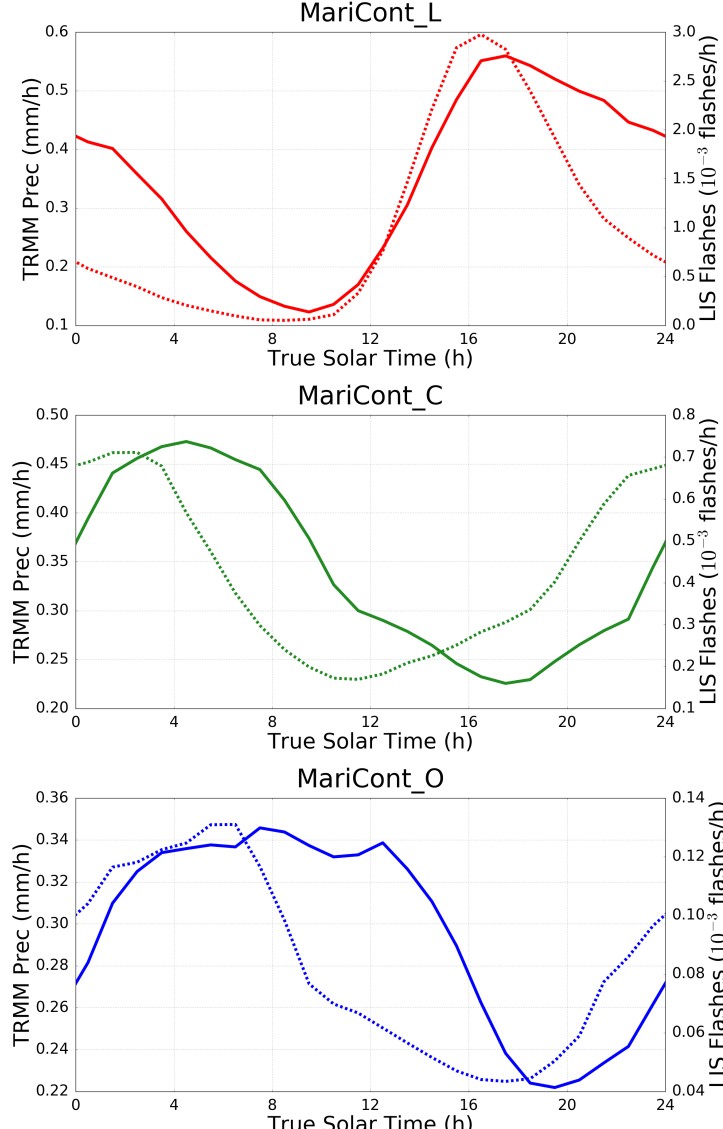

**Figure 7.** Diurnal cycle of Prec (full line) and diurnal cycle of Flash (dash line) over MariCont_L (top), MariCont_O (middle) and Mari-Cont_C (bottom).

finding of Liu and Zipser (2008) over the whole tropics. Different maximum times could come from the fact that, while the
deep convective activity intensity starts to decrease with the number of flashes, Prec is still high during the dissipating stage
of the convection and takes longer times to decrease than Flash. Consequently, combining our results with the ones presented
in Dion et al. (2019), Flash and Prec can be considered as good proxies of deep convection during the growing phase of the
convection over the MariCont_L.





Over offshore areas (Fig. 7b), minima of diurnal cycle of Prec and diurnal cycle of Flash are in the late afternoon, between

16:00 LT and 17:00 LT (Flash) and 17:00 LT and 18:00 LT (Prec), whilst maxima of diurnal cycle of Prec and Flash are reached

in the early morning, between 04:00 LT and 09:00 LT (Flash) and around 08:00 LT – 09:00 LT (Prec). Results over offshore

areas are consistent with diurnal cycle of Flash and Prec calculated by Liu and Zipser (2008) over the whole tropical ocean,

showing the increasing phase of the diurnal cycle of Flash starting 1–2 hours before the increasing phase of the diurnal cycle of

Prec. Over coastlines (Fig. 7c), the Prec diurnal cycle is delayed by about + 2 to 7 h with respect to the Flash diurnal cycle. Prec

minimum is around 18:00 LT while Flash minimum is around 11:30 LT. Maxima of Prec and Flash are found around 04:00

LT and 02:00 LT, respectively. This means that the increasing phase of Flash is 2-3 h longer than that of Prec. These results

are consistent with Mori et al. (2004) showing a diurnal maximum of precipitation in the early morning between 02:00 LT and

03:00 LT and a diurnal minimum of precipitation around 11:00 LT, over coastal zones of Sumatra. According to Petersen and

Rutledge (2001) and Mori et al. (2004), coastal zones are areas where precipitation results more from convective activity than

from stratiform activity and the amplitude of diurnal maximum of Prec decreases with the distance from the coastline.

The time of transition from maximum to minimum of Prec is always longer than that of Flash. The period after the maximum

of Prec is likely more representative of stratiform rainfall than deep convective rainfall. Consistently, over the MariCont ocean,

model results from Love et al. (2011) have shown that deep warm anomalies around noon are coming from a downwelling

wavefront which suppresses the convection offshore during early afternoon. To summarize, diurnal cycles of Prec and Flash

show that:

i) over land, Flash increases proportionally with Prec during the growing phase of the convection,

ii) over offshore areas, Flash increasing phase is advanced by about 1–2 hours compared to Prec increasing phase,

iii) over coastlines, Flash increasing phase is advanced by more than 6–7 hours compared to Prec increasing phase.

In section 7, we investigate whether this time difference impacts the estimation of ΔIWC over land, coasts, and offshore

areas.

### 5.3   Prec and Flash diurnal cycles and small-scale processes

In this subsection, we study the diurnal cycle of Prec and Flash at $0.25° \times 0.25°$ resolution over areas of deep convective

activity over the MariCont. In line with the distribution of large value of Prec (Fig. 2), IWC (Fig. 3) and ΔIWC (Fig. 4), we

have selected five islands and five seas areas over the MariCont. Diurnal cycles of Prec and Flash are presented over lands for

a) Java, b) Borneo, c) New Guinea, d) Sulawesi and e) Sumatra as shown in Figure 8 and over sea for the a) Java Sea, b) North

Australia Sea (NAuSea), c) Bismark Sea, d) West Sumatra Sea (WSumSea) and e) China Sea as shown in Figure 9.

Over land, the amplitude of the diurnal cycle of Prec is the largest over Java (Fig. 8a), consistently with Qian (2008) with a

maximum reaching 1 mm h-1 while, over the other areas, maxima are between 0.4 and 0.6 mm h-1. Furthermore, over Java,

the duration of the increasing phase in the diurnal cycle of Prec is 6-h consistent with that of Flash and elsewhere, the duration

of the increasing phase is longer in Prec than in Flash by 1–2 h. The particularity of Java is related to the increasing phase

of the diurnal cycle of Prec (6 h) that is faster than over all the other land areas considered in our study (8 – 10 h) and is

very consistent with the diurnal cycle of Prec over South America and South Africa (Dion et al., 2019). The strong and rapid



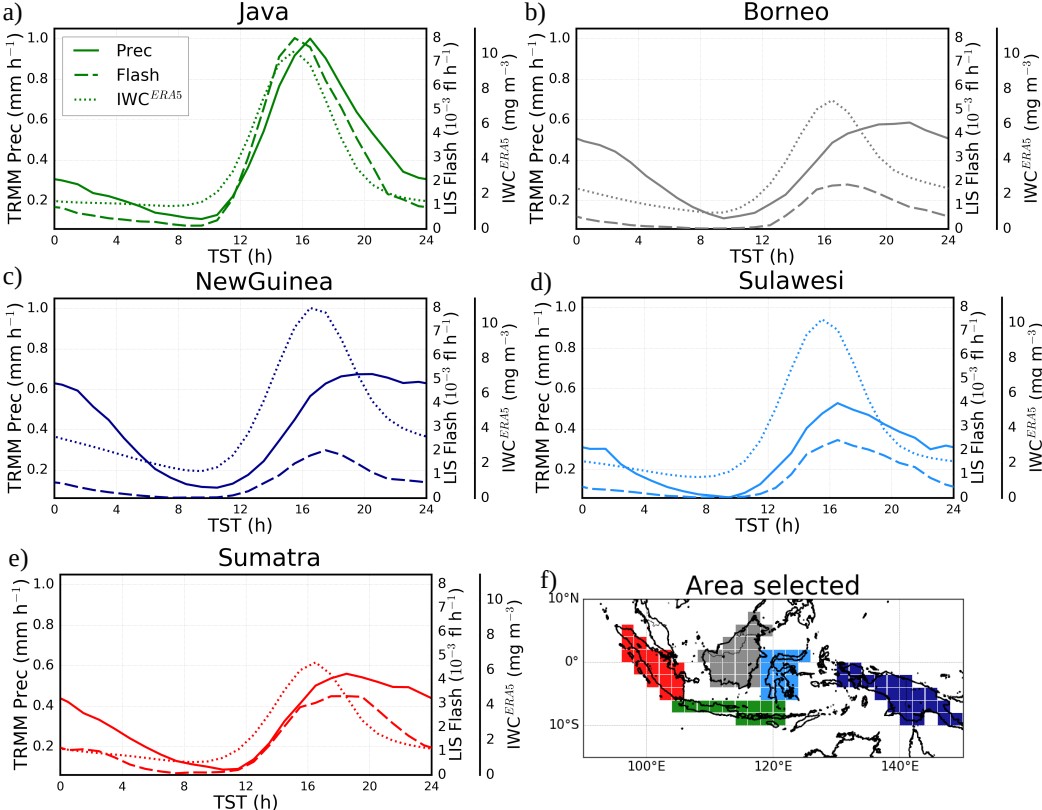

**Figure 8.** Diurnal cycles of Prec (solid line), Flash (dashed line) and $IWC^{ERA5}$ from ERA5 at 150 hPa (dotted line) over MariCont islands: Java (a), Borneo (b), New Guinea (c), Sulawesi (d) and Sumatra (e) and map of the study zones over land (f).

convective growing phase measured over Java might be explained by the fact that the island is narrow with high mountains (up to ∼ 2000 m of altitude) reaching the coast. The topography promotes the growth of intense and rapid convective activity.

The convection starts around 09:00 LT, rapidly elevating warm air up to the top of the mountains. Around 15:00 LT, air masses cooled in altitude are transported to the sea favoring the dissipating stage of the convection. Sulawesi is also a small island and presents the same onset of growing phase for the convection than Java, consistently with results presented in Nesbitt and Zipser (2003) and Qian (2008). Other islands, such as Borneo, New Guinea and Sumatra, have high mountains but also large lowland areas. Mountains promote deep convection at the beginning of the afternoon while lowlands help maintaining

the convective activity through shallow convection and stratiform rainfalls (Nesbitt and Zipser, 2003; Qian, 2008). Deep and shallow convections are then mixed during the slow dissipating phase of the convection (from ∼ 16:00 LT to 08:00 LT). However, because Flash are observed only in deep convective clouds, the decreasing phase of Flash diurnal cycles decreases more rapidly than the decreasing phase of Prec. The diurnal maxima of Prec found separately over the 5 islands of the MariCont are much higher than the diurnal maxima of Prec found over tropical lands (South America, South Africa and MariCont_L)





from Dion et al. (2019): $\sim 0.6 - 1.0$ mm h-1 and $\sim 0.4$ mm h-1, respectively. However, the duration of the increasing phase of the diurnal cycle of Prec is consistent with the one calculated over tropical lands by Dion et al. (2019).

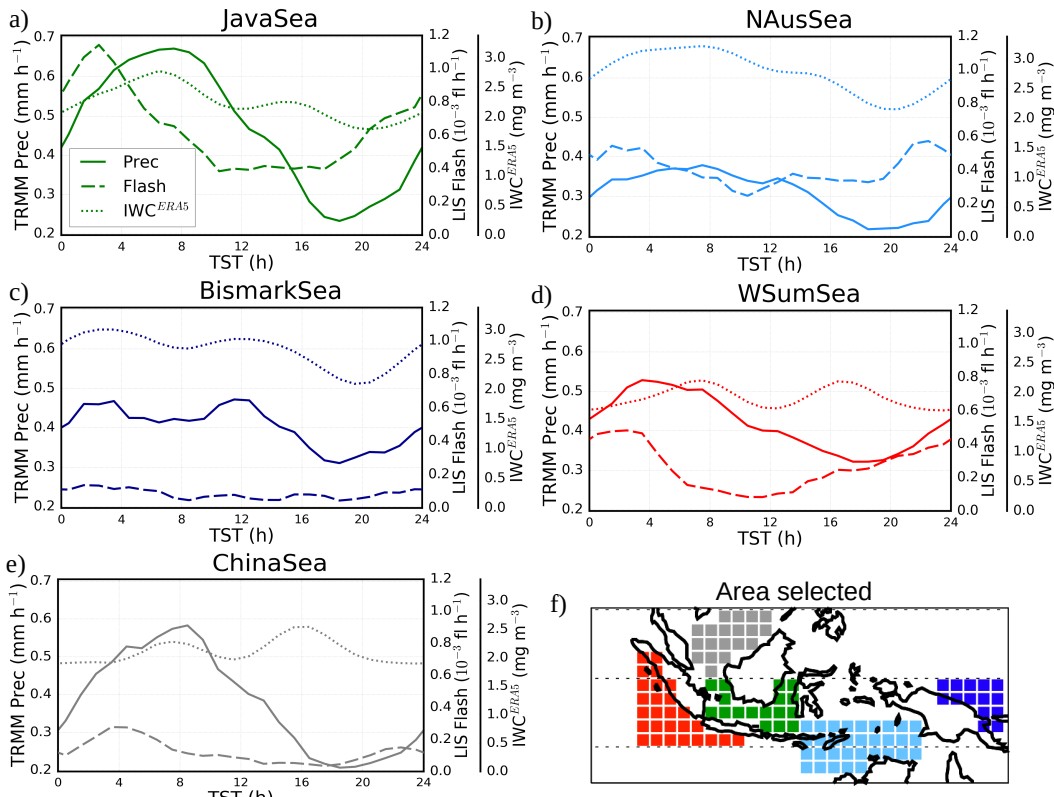

**Figure 9.** Diurnal cycles of Prec (solid line), Flash (dashed line) and IWC$^{ERA5}$ from ERA5 at 150 hPa (dotted line) over MariCont seas: Java Sea (a), North Australia Sea (NAusSea) (b), Bismark Sea (c), West Sumatra Sea (WSumSea) (d), China Sea (e) and map of the study zones over sea (f).

Over sea, the five selected areas (Fig. 9a–e) show a diurnal cycle of Prec and Flash consistent with coastline or offshore areas as a function of the area considered. The diurnal cycle of Prec and Flash over Java Sea is similar to the one over coastlines (Fig. 7c). Java Sea (Fig. 9a), an area mainly surrounded by coasts, shows the largest diurnal maximum of Prec ($\sim 0.7$ mm h-1)

and Flash ($\sim 1.1$ 10-3 flashes h-1) with the longest growing phase. In this area, land and sea breezes observed in coastal areas impact the diurnal cycle of the convection (Qian, 2008). During the night, land breeze develops from a temperature gradient between warm sea surface temperature and cold land surface temperature and conversely during the day. Over Java Sea, Prec is strongly impacted by land breezes from Borneo and Java islands (Qian, 2008), explaining why Prec and Flash reach largest values during the early morning. By contrast, NAuSea, Bismark Sea and WSumSea (Figs. 9b, c and d, respectively) present

small amplitude of diurnal cycle. In our analysis, these three study zones are the areas including the most of offshore pixels. Java Sea and WSumSea present a similar diurnal cycle of Prec and Flash, with Flash growing phase starting about 4 h earlier





than that of Prec. China Sea also shows a diurnal maximum of Flash shifted by about 4 hours before the diurnal maximum of Prec, but the time of the diurnal minimum of Prec and Flash is similar. Over NAuSea and Bismark Sea, the diurnal cycle of Flash shows a very weak amplitude with maxima reaching only 0.1 flashes h-1. However, the diurnal minima of Prec and

Flash over Bismark Sea are found to be at the same time (∼17:00 LT). Over NAuSea, the diurnal minimum of Prec is delayed by more than 7 hours compared to the diurnal minimum of Flash.

To summarize, over islands, Flash and Prec convective increasing phases start at the same time and increase similarly but the diurnal maximum of Flash is reached 1–2 hours before the diurnal maximum of Prec. Over seas, the duration of the convective increasing phase and the amplitude of the diurnal cycles are not always similar depending on the area considered. The diurnal

cycle of Flash is advanced by 4 hours over Java Sea and West Sumatra Sea and by more than 7 hours over North Australia Sea compared to the diurnal cycle of Prec. China Sea and Bismark Sea present the same time of the onset of the Flash and Prec increasing phase. In the next section, we estimate ΔIWC over the 5 selected island and sea areas from Prec and Flash as a proxy of deep convection.

## 6 Horizontal distribution of IWC from ERA5 reanalyses

The ERA5 reanalyses provide hourly IWC at 150 and 100 hPa ($IWC^{ERA5}$). Figures 10a, b, c and d present the daily mean and the hour of the diurnal maxima of $IWC^{ERA5}$ at 150 and 100 hPa. In the UT, the daily mean of $IWC^{ERA5}$ shows a horizontal distribution over the MariCont consistently with that of $IWC^{MLS}$ (Fig. 2e), except over the New Guinea where $IWC^{ERA5}$ (reaching 6.4 mg m-3) is much stronger than $IWC^{MLS}$ (∼ 4.0 mg m-3). The highest amount of $IWC^{ERA5}$ is located over the New Guinea mountain chain and in the West coast of North Australia (reaching 6.4 mg m-3 in the UT and 1.0 mg m-3 in the

TL). Over island in the UT and the TL, the hour of the $IWC^{ERA5}$ diurnal maximum is found between 12:00 LT and 15:00 LT over Sulawesi and New Guinea and between 15:00 LT and 21:00 LT over Sumatra, Borneo and Java, that is close to the hour of the diurnal maximum of Flash over island (Fig. 6). Over sea, in the UT and the TL, the hour of the $IWC^{ERA5}$ diurnal maximum is found between 06:00 LT and 09:00 LT over West Sumatra Sea, Java Sea, North Australia Sea, between 06:00 LT and 12:00 LT over China Sea and between 00:00 LT and 03:00 LT over Bismark Sea. There is no significant differences

between the hour of the maximum of $IWC^{ERA5}$ in the UT and in the TL.

The diurnal cycles of $IWC^{ERA5}$ at 150 hPa are presented in Figs. 8 and 9 over the selection of islands and seas of the MariCont together with the diurnal cycles of Prec and Flash. Over island (Fig. 8), the maximum of the diurnal cycle of $IWC^{ERA5}$ is found between 16:00 LT and 17:00 LT, consistently with the diurnal cycle of Prec and Flash. The duration of the increasing phase of the diurnal cycles of Prec, Flash and $IWC^{ERA5}$ are all consistent to each other, ∼ 4 − 5 h. Over sea

(Fig. 9), the maximum of the diurnal cycle of $IWC^{ERA5}$ is mainly found between 07:00 LT and 10:00 LT, consistently with the diurnal cycle of Prec (which is 2 − 3 hours after the diurnal maxima for Flash). Over Java Sea and North Australia Sea, the diurnal maxima and minima are found at the same hours as the diurnal maxima and minima of Prec. Thus, the duration of the increasing phase of the diurnal cycles of $IWC^{ERA5}$ is consistent with the one of Prec over these two sea study zone, ∼ 10 hours, but not with the one of Prec. Over Bismark Sea, West Sumatra Sea and China Sea, the diurnal cycles of $IWC^{ERA5}$ show





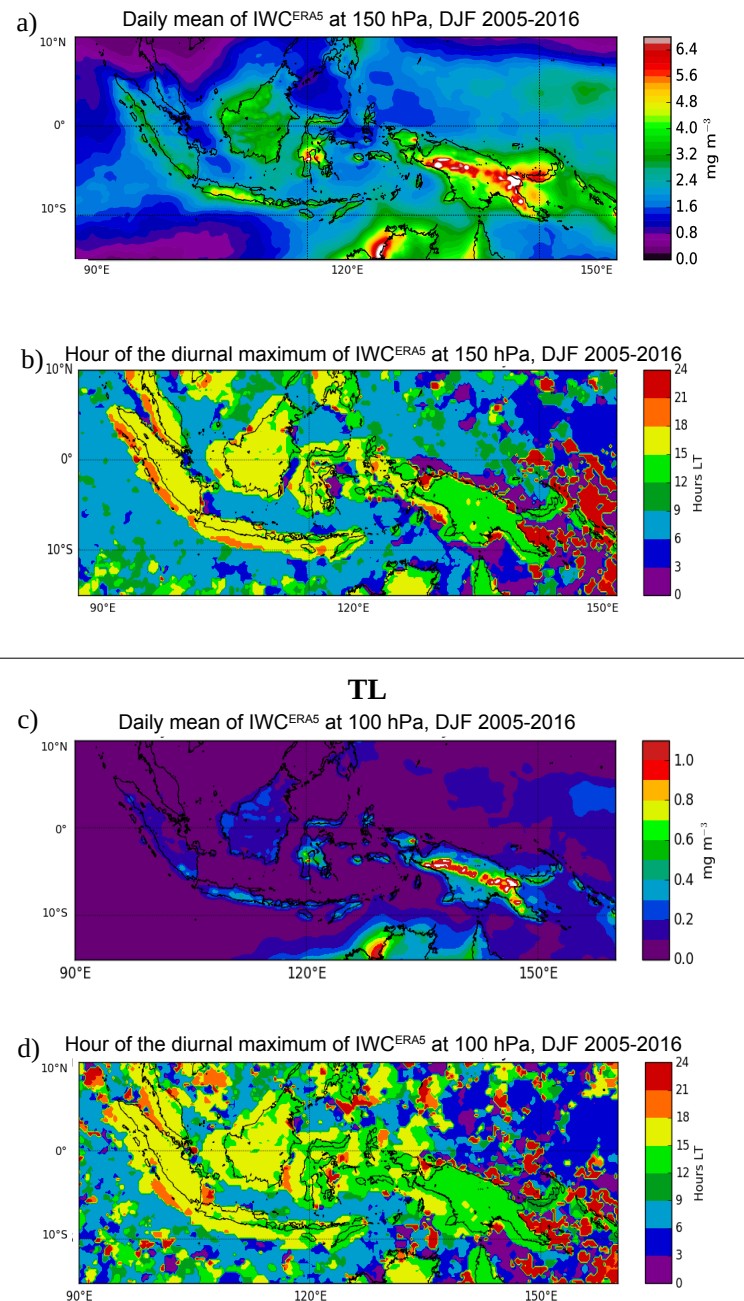

**Figure 10.** Daily mean of IWC$^{ERA5}$ averaged over the period DJF 2005-2016 at 150 hPa (a) and at 100 hPa (c); Time (hour, local time (LT)) of the diurnal maximum of IWC$^{ERA5}$ at 150 hPa (b) and at 100 hPa (d).





two maxima: one at the end of the local morning and the other one during the local afternoon. Over Bismark Sea, these two maxima are consistent with those observed in the diurnal cycle of Prec, consequently the increasing phases of the diurnal cycles of $IWC^{ERA5}$ and Prec are consistent to each other. Over West Sumatra Sea and China Sea, the two maxima in the diurnal cycle of $IWC^{ERA5}$ are not observed in the diurnal cycle of Prec and Flash, consequently the increasing phase of $IWC^{ERA5}$ is not consistent with the one of Prec nor the one of Flash.

## 7    Ice injected over a selection of island and sea areas

### 7.1    ΔIWC deduced from observations

Figure 11 synthesizes ΔIWC in the UT and the TL over the 5 islands and 5 seas of the MariCont studied in the previous section. Eqs. (1-3) are used to calculated ΔIWC from Prec ($\Delta IWC^{Prec}$) and from Flash ($\Delta IWC^{Flash}$). The observational ΔIWC range calculated between $\Delta IWC^{Prec}$ and $\Delta IWC^{Flash}$ provides an upper and lower bound of ΔIWC calculated from observational 365 datasets.

In the UT (Fig. 10a), over island, ΔIWC calculated over Sumatra, Borneo, Sulawesi and New Guinea varies from 4.87 to 6.86 mg m-3 whilst, over Java, ΔIWC reaches 7.89–8.72 mg m-3. $\Delta IWC^{Flash}$ is generally greater than $\Delta IWC^{Prec}$ by less than 1.0 mg m-3 (41%) for all the islands, excepted for New Guinea where the difference reaches 1.40 mg m-3 (20%). Conversely, over Java, $\Delta IWC^{Prec}$ is larger than $\Delta IWC^{Flash}$ by 0.71 mg m-3 (8%). Over sea, ΔIWC varies from 1.17 to 4.37 370 mg m-3. $\Delta IWC^{Flash}$ is greater than $\Delta IWC^{Prec}$ from 0.6 to 2.09 mg m-3 (31-50%), except for Java Sea, where $\Delta IWC^{Prec}$ is greater than $\Delta IWC^{Flash}$ by 0.21 mg m-3 (6 %). Over North Australia Sea, probably because of the 7-hours lagged diurnal cycle of Flash compared to Prec (Fig. 9), $\Delta IWC^{Flash}$ is almost twice greater than $\Delta IWC^{Prec}$ values.

In the TL (Fig. 10b), the observational ΔIWC range is found between 0.72 and 1.28 mg m-3 over island and between 0.22 and 0.74 mg m-3 over sea. The same conclusions apply to the observational ΔIWC range calculated between $\Delta IWC^{Prec}$ and 375 $\Delta IWC^{Flash}$ in the TL as in the UT with differences less than 0.39 mg m-3.

At both altitudes, Java shows the largest injection of ice over the MariCont. $\Delta IWC^{Prec}$ and $\Delta IWC^{Flash}$ are consistent to within 4-20 % over island and 6-50 % over sea in the UT and the TL. The largest difference over sea is probably due to the larger contamination of stratiform precipitation included in Prec over sea. Although Flash, is not contaminated by stratiform clouds, it could be a better proxy than Prec over sea but it is unfortunately negligible: less than 10-2 flashes per day (Fig. 6).

### 7.2    ΔIWC deduced from reanalyses

ΔIWC from ERA5 ($\Delta IWC^{ERA5}$) is calculated in the UT and the TL ($z_0 = 100$ and 150 hPa, respectively) as the max – min difference in the amplitude of the diurnal cycle. To be consistent with the MLS observations, we should degrade the ERA5 vertical resolution to assess the impact of the vertical resolution on $\Delta IWC^{ERA5}$. In the optimal estimation theory (**?**), the vertical distribution of $IWC^{ERA5}$ ($IWC^{ERA5}(z)$) should be convolved with the $IWC^{MLS}$ averaging kernels, but this





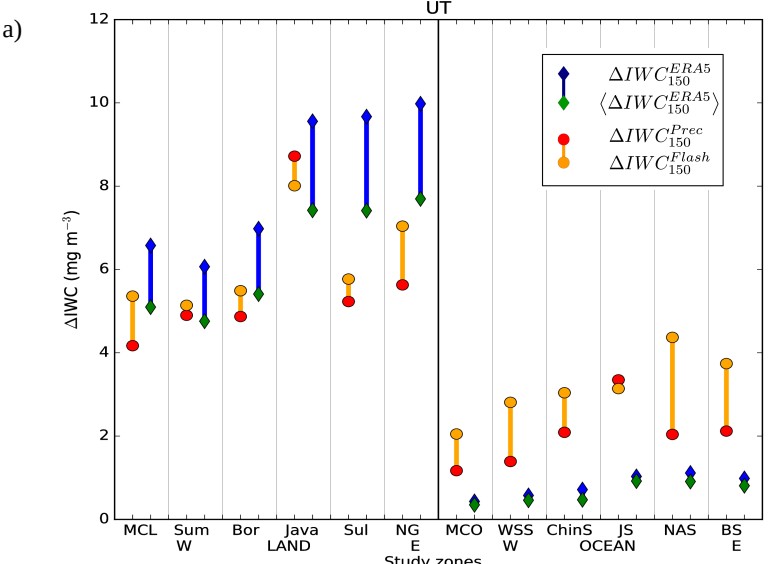

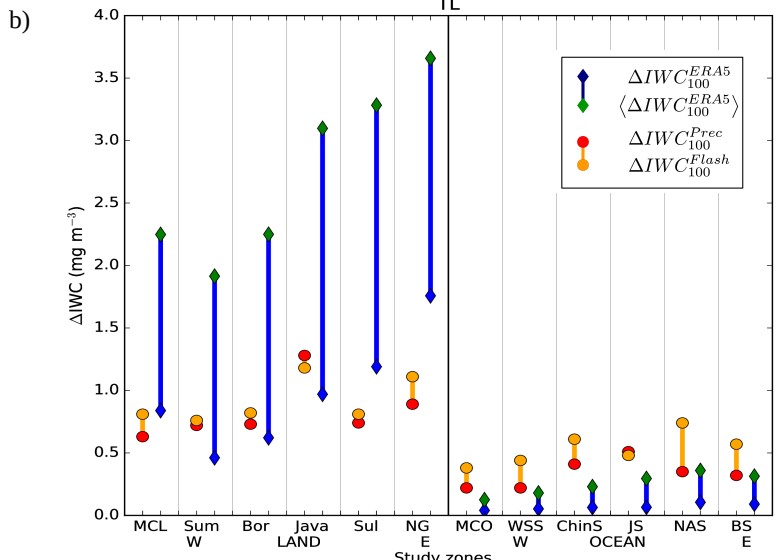

**Figure 11.** a) $\Delta$IWC (mg m-3) estimated from Prec (red) and Flash (orange) at 146 hPa and $\Delta$IWC estimated from ERA5 at the level 150 hPa and at the level 150 hPa degraded in the vertical, over islands and seas of the MariCont: MariCont_L (MCL) and MariCont_O (MCO); from West (W) to East (E) over land, Sumatra (Sum), Borneo (Bor), Java, Sulawesi (Sul) and New Guinea (NG); and over seas, West Sumatra Coast (WSS), China Sea (ChinS), Java Sea (JS), North Australia Sea (NAS) and Bismark Sea (BS). b), Same as a) but for 100 hPa.





information is not provided by MLS. Instead, we have created a unitary triangular function $\phi(z, z_0, \delta z)$ centered at $z_0$ with a width at half-maximum of $\delta z$. The convolved $IWC^{ERA5}(z)$ at $z_0$ ($\langle IWC_{z_0}^{ERA5}\rangle$) is calculated as:

$$\langle IWC_{z_0}^{ERA5}\rangle = \int \phi(z, z_0, \delta z) IWC^{ERA5}(z)dz \tag{4}$$

Consistently with Livesey et al. (2019), we have fixed $\delta z = 4$ and 5 km at $z_0 = 146$ and 100 hPa, respectively. The ice injected from ERA5 at $z_0 = 146$ and 100 hPa with degraded vertical resolution ($\langle \Delta IWC_{z_0}^{ERA5}\rangle$) is thus calculated from $\langle IWC_{z_0}^{ERA5}\rangle$.

Figure 11 shows $\Delta IWC^{ERA5}$ and ($\langle \Delta IWC_{z_0}^{ERA5}\rangle$) at 150 and 100 hPa, over the island and the sea study zones. In the UT (Fig. 11a), over island, $\Delta IWC_{150}^{ERA5}$ and $\langle \Delta IWC_{150}^{ERA5}\rangle$ calculated over Sumatra and Borneo vary from 4.75 to 6.97 mg m-3 ($\sim$22 % of variability per study zones) whilst $\Delta IWC_{150}^{ERA5}$ and $\langle \Delta IWC_{150}^{ERA5}\rangle$ over Java, Sulawesi and New Guinea reach 7.41–9.97 mg m-3 ($\sim$23 % of variability per study zones). Over sea, $\Delta IWC_{150}^{ERA5}$ and $\langle \Delta IWC_{150}^{ERA5}\rangle$ vary from 0.35 to 1.11 mg m-3 (9 – 32 % of variability per study zones). Over island and sea, $\Delta IWC_{150}^{ERA5}$ is greater than $\langle \Delta IWC_{150}^{ERA5}\rangle$. The

small differences between $\Delta IWC_{150}^{ERA5}$ and $\langle \Delta IWC_{150}^{ERA5}\rangle$ over island and sea in the UT support the fact that the vertical resolution at 150 hPa has a low impact on the estimated $\Delta IWC$.

In the TL, over land, $\Delta IWC_{100}^{ERA5}$ and $\langle \Delta IWC_{100}^{ERA5}\rangle$ vary from 0.46 to 3.65 mg m-3 ($\sim$68% of variability per study zones) with $\langle \Delta IWC_{100}^{ERA5}\rangle$ being lower than $\Delta IWC_{100}^{ERA5}$ by less than 2.12 mg m-3. Over sea, $\Delta IWC_{100}^{ERA5}$ and $\langle \Delta IWC_{100}^{ERA5}\rangle$ vary from 0.04 to 0.36 mg m-3 ($\sim$71 % of variability per study zones) with $\Delta IWC_{100}^{ERA5}$ lower than $\langle \Delta IWC_{100}^{ERA5}\rangle$ by less

than 0.25 mg m-3. The large differences between $\Delta IWC_{100}^{ERA5}$ and $\langle \Delta IWC_{100}^{ERA5}\rangle$ over island and sea in the TL support the fact that the vertical resolution at 100 hPa has a high impact on the estimation of $\Delta IWC$.

### 7.3  Synthesis

The comparison between the observational $\Delta IWC$ range and the reanalysis $\Delta IWC$ range is presented in Fig. 11. In the UT, over land, observation and reanalysis $\Delta IWC$ ranges agree to within 0 – 0.64 mg m-3 (8 %), except over Sulawesi with differences

of 1.63 mg m-3 (23%). Over sea, the observational $\Delta IWC$ range is systematically greater than the reanalysis $\Delta IWC$ range to within $\sim$ 1.00 – 2.19 mg m-3 (75 %). Combining observational and reanalysis ranges, the total $\Delta IWC$ variation range is estimated in the UT between 4.17 and 9.97 mg m-3 ($\sim$ 20 % of variability per study zones) over lands and between 0.35 and 4.37 mg m-3 ($\sim$ 30 % of variability per study zones) over sea, and in the TL, between 0.63 and 3.65 mg m-3 ($\sim$ 70 % of variability per study zones) over land, and between 0.04 and 0.74 mg m-3 ($\sim$ 80 % of variability per study zones) over sea.

The amount of ice injected in the UT deduced from observational and reanalysis are consistent to each other over Mari-Cont_L, Borneo and Java, with significant differences between observations and reanalyses over Sulawesi, New Guinea and MariCont_O. The impact of the vertical resolution on the estimation of $\Delta IWC$ in the TL is certainly non negligeable, with larger $\Delta IWC$ variability range in the TL (70 – 80 %) than in the UT (20 – 30 %). At both levels, observational and reanalyses $\Delta IWC$ estimated over land are more than twice larger than $\Delta IWC$ estimated over sea. Finally, whatever the level considered,

although Java has shown particularly high values in the observational $\Delta IWC$ range compared to other study zones, the reanalysis $\Delta IWC$ range shows that Sulawesi and the New Guinea would also be able to reach similar high values of $\Delta IWC$ as Java.





Whatever the datasets used, the vertical distribution of ΔIWC in the TTL has shown a gradient of - 6 mg m-3 between the UT and the TL over land compared to a gradient of - 2 mg m-3 between the UT and the TL over ocean.

## 8   Discussion on small-scale convective processes impacting ΔIWC over a selection of areas

Our results have shown that, in all the datasets used, Java island and Java Sea are the two areas with the largest amount of ice injected up to the UT and the TL over the MariCont lands and sea, respectively. In this section, processes impacting ΔIWC in the different study zones are discussed.

### 8.1   Java island, Sulawesi and New Guinea

Sulawesi, New Guinea and particularly Java island have been shown as the areas of the largest ΔIWC in the UT and TL.
Qian (2008) have used high resolution observations and regional climate model simulations to show the three main processes impacting the diurnal cycle of rainfall over the Java island. The main process explaining the rapid and strong peak of Prec during the afternoon over Java (Fig. 8a) is the sea-breeze convergence around midnight. This convergence caused by sea-breeze phenomenon increases the deep convective activity and impacts on the diurnal cycle of Prec and on the IWC injected up to the TL by amplifying their quantities. The second process is the mountain-valley wind converging toward the mountain
peaks, and reinforcing the convergence and the precipitations. The land breeze becomes minor compared to the mountain-valley breeze and this process is amplified with the mountain altitude. As shown in Fig. 2b, the New Guinea has the highest mountain chain of the MariCont. The third process shown by Qian (2008) is precipitation that is amplified by the cumulus merged processes which is more important over tiny islands such as Java (or Sulawesi) than over large islands such as Borneo or Sumatra. Another process is the interaction between sea-breeze and precipitation-driven cold pools that generates lines of
strong horizontal moisture convergence (Dauhut et al., 2016). Thus, IWC is increasing proportionally with Prec consistently with the results from Dion et al. (2019) and rapid convergence combined with deep convection transport elevated amounts of IWC at 13:30 LT (Fig. 3) producing high ΔIWC during the growing phase of the convection (Fig. 4 and Fig. 10) over Java.

### 8.2   West Sumatra Sea

In section 4.3, it has been shown that the West Sumatra Sea is an area with positive anomaly of Prec during the growing phase
of the convection but negative anomaly of IWC, which differs from other places. The diurnal cycle of Prec over West Sumatra Sea has been studied by Mori et al. (2004) using 3 years of TRMM precipitation radar (PR) datasets, following the 2A23 Algorithm Awaka (1998) separating stratiform to convective rainfall type. Mori et al. (2004) have shown that rainfall over Sumatra is characterized by convective activity with a diurnal maximum between 15:00 and 22:00 LT, while, over the West Sumatra Sea, the rainfall type is convective and stratiform, with a diurnal maximum during the early morning (as observed
in Fig. 9). Furthermore, their analyses have shown a strong diurnal cycle of 200-hPa wind, humidity and stability, consistent with the diurnal cycle of precipitation measured by TRMM Precipitation Radar (PR) over Sumatra West Sea and Sumatra island. Stratiform and convective clouds are both at the origin of heavy rainfall in the tropics (Houze and Betts, 1981; Nesbitt





and Zipser, 2003) and in the West Sumatra Sea, but stratiform clouds are mid-altitude clouds in the troposphere and do not transport ice up to the tropopause. Flash measured over the West Sumatra Sea would thus be a better proxy of deep convection

in order to estimate $\Delta$IWC than Prec because Prec is contaminated by stratiform rainfall.

### 8.3 North Australia Sea

Comparisons between Figs. 2c and 6a have shown strong daily mean of Flash (10-2 –10-1 flashes day-1) but low daily mean of Prec (2 – 8 mm day-1) over the North Australia Sea. Consequently, results in Fig. 10 have shown the strongest differences between $\Delta$IWC$^{Prec}$ and $\Delta$IWC$^{Flash}$ over the North Australia Sea, with $\Delta$IWC$^{Flash}$ greater than $\Delta$IWC$^{Prec}$ by 2.32 mg

m-3 in the UT and by 0.39 mg m-3 in the TL (53% of variability). Convective systems of the North Australian land and seas have been studied by Pope et al. (2008). The North Australia Sea, mainly composed by the Arafura Sea and the Timor Sea, has the strongest convective activity during the DJF season (monsoon months) (Pope et al., 2008). During this season, isolated pulse convection and many mesoscale convective systems (MCSs) are active, such as the famous "Hector" storm over the Tiwi Islands, North of Darwin (Carbone et al., 2000; Dauhut et al., 2016), squall lines (Chappel, 2001) and cloud line (Goler et al.,

2006). Pope et al. (2008), identifying clouds from IR imagery, have shown that the number of clouds over the North Australia Sea is the largest during the afternoon but the size of the cloud is larger in the early morning. Although the number of clouds is low near 01:30 LT, the deepest daily convective activity during the early morning would be at the origin of the strong relative value of IWC amount at 01:30 LT (see Fig. 3). Their studies have also shown that the diurnal maximum of number of clouds and cloud sizes over coastal part near the New Guinea is observed during the night, marked by land breeze convergence and

gravity waves. Thus, nighttime convective systems over coasts propagate through the North Australia Sea and reinforce the IWC amount in the UT in the early morning over sea. Figure 12 presents the daily average of horizontal wind in the UT at 150 hPa and in the TL at 100 hPa over the period DJF 2004-2017. The wind direction at both altitudes is Westward, from North Australia land to the North Australia Sea. Thus, we suggest that because 1) the diurnal maximum of deep convection over North Australia land is found during the afternoon (Pope et al., 2008), and 2) the wind prevalence is from the North Australia

land to sea, air masses charged with ice are advected from the North Australia land to the North Australia coastlines and seas at both altitudes. Thus, $\Delta$IWC is injected during the day over the North Australia land in the UT and the TL and would be transported over the sea from the evening to the night and the early morning.

### 9 Conclusions

The present study has combined observations of ice water content (IWC) measured by the Microwave Limb Sounder (MLS),

precipitation (Prec) from the algorithm 3B42 of the Tropical Rainfall Measurement Mission (TRMM), the number of flashes (Flash) from the Lightning Imaging Sensor (LIS) of TRMM with IWC provided by the ERA5 reanalyses in order to estimate the amount of ice injected ($\Delta$IWC) in the upper troposphere (UT) and the tropopause level (TL) over the MariCont, from the method proposed in a companion paper (Dion et al., 2019). $\Delta$IWC is firstly calculated using the IWC measured by MLS (IWC$^{MLS}$) in DJF from 2004 to 2017 at the temporal resolution of 2 observations per day and Prec from by TRMM-3B42





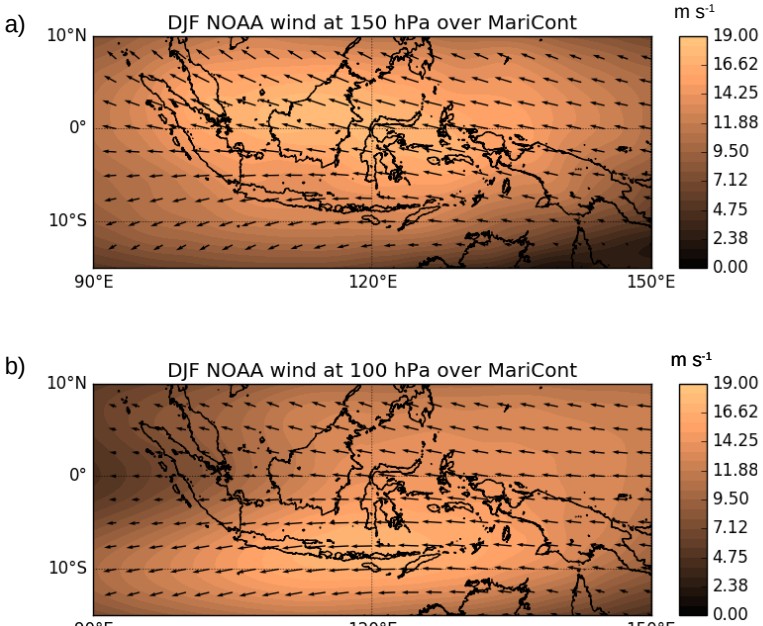

**Figure 12.** Daily mean of wind speed and direction at 150 hPa (UT) a), and at 100 hPa (TL) b), from NOAA averaged during DJF 2004-2017 over the MariCont. The arrows with arbitrary length unit represent the wind direction.

during the same period, to obtain a 1-hour resolution diurnal cycle. In the model used (Dion et al., 2019), Prec is considered as a proxy of deep convection to impact ice ($\Delta\text{IWC}^{Prec}$) into the UT and the TL. While Dion et al. (2019) have calculated $\Delta\text{IWC}^{Prec}$ over large convective study zones in the tropics, we show the spatial distribution of $\Delta\text{IWC}^{Prec}$ into the UT and the TL at $2° \times 2°$ horizontal resolution over the MariCont, highlighting local areas of strong injection of ice up to 20 mg m-3 in the UT and up to 3 mg m-3 in the TL. $\Delta\text{IWC}$ injected in the UT and the TL has also been evaluated by using another

proxy of deep convection: Flash measured by TRMM-LIS. Diurnal cycle of Flash has been compared to diurnal cycle of Prec, showing consistencies in 1) the spatial distribution of Flash and Prec over the MariCont (maxima of Prec and Flash located over land and coastline), and 2) their diurnal cycles over land (similar onset and duration of the diurnal cycle increasing phase). Differences have been mainly observed over sea and coastline areas, with the onset of the diurnal cycle increasing phase of Prec delayed by several hours depending on the considered area (from 2 to 7 h) compared to Flash. $\Delta\text{IWC}$ calculated by using

Flash as a proxy of deep convection ($\Delta\text{IWC}^{Flash}$) is compared to $\Delta\text{IWC}^{Prec}$ over five islands and five seas of the MariCont to establish an observational $\Delta\text{IWC}$ range over each study zone. $\Delta\text{IWC}$ is also estimated from IWC provided by the ERA5 reanalyses ($\Delta\text{IWC}^{ERA5}$ and IWC$^{ERA5}$, respectively) at 150 and 100 hPa over the study zones. We have also degraded the vertical resolution of IWC$^{ERA5}$ to be consistent with that of IWC$^{MLS}$ observations: 4 km at 146 hPa and 5 km at 100 hPa. The $\Delta\text{IWC}$ ranges calculated from observations and reanalyses were evaluated over the selected study zones (island and sea).





With the study of $\Delta \text{IWC}^{Prec}$, results show that the largest amount of ice injected in the UT and TL per $2° \times 2°$ pixels are related to i) an amplitude of Prec diurnal cycle larger than 0.5 mm h-1, ii) values of IWC measured during the growing phase of the convection larger than 4.5 mg m-3 and iii) duration of the growing phase of the convection longer than 9 hours. The largest $\Delta \text{IWC}^{Prec}$ has been found over areas where the convective activity is the deepest. The observational $\Delta \text{IWC}$ range calculated between $\Delta \text{IWC}^{Prec}$ and $\Delta \text{IWC}^{Flash}$ has been found to be within 4 – 20 % over land and to within 6 – 50 % over sea. The

largest differences between $\Delta \text{IWC}^{Prec}$ and $\Delta \text{IWC}^{Flash}$ over sea might be due to the combination between the presence of stratiform precipitation included into Prec and the very low values of Flash over seas (<10-2 flashes day-1). The diurnal cycle of $\text{IWC}^{ERA5}$ at 150 hPa is more consistent with that of Prec and Flash over land than over ocean. Finally, the observational $\Delta \text{IWC}$ range has been shown to be consistent with the reanalysis $\Delta \text{IWC}$ range to within 23 % over land and to within 75 % over sea in the UT and to within 49 % over land and to within 39 % over sea in the TL. Thus, thanks to the combination between the observational and reanalysis $\Delta \text{IWC}$ ranges, the total $\Delta \text{IWC}$ variation range has been found in the UT to be between 4.17

and 9.97 mg m-3 (to within 20 % per study zones) over land and between 0.35 and 4.37 mg m-3 (to within 30 % per study zones) over sea and, in the TL, between 0.63 and 3.65 mg m-3 (to within 70 % per study zones) over land and between 0.04 and 0.74 mg m-3 (to within 80% per study zones) over sea. The $\Delta \text{IWC}$ variation range in the TL is larger than that in the UT highlighting the stronger impact of the vertical resolution in the observations in the TL compared to the UT.

The study at small scale over islands and seas of the MariCont has shown that, $\Delta \text{IWC}$ from ERA5, Prec and Flash, in the UT agree to within 0 – 0.64 mg m-3 (8%) over MariCont_L, Sumatra, Borneo and Java with the largest values obtained over Java. However, while Java presents the largest amount of $\Delta \text{IWC}^{Prec}$ and $\Delta \text{IWC}^{Flash}$ in the UT and the TL (larger by about 1.0 mg m-3 in the UT and about 0.25 mg m-3 in the TL than other land study zones), New Guinea, and Sulawesi, reach similar range of values of ice injected with ERA5 as Java in the UT and even larger ranges of values as Java in the TL. Processes related to the

strongest amount of $\Delta \text{IWC}$ injected into the UT and the TL have been identified as the combination of sea-breeze, mountain-valley breeze and cumulus merged, such as over the New Guinea and accentuated over tiny islands with high topography such as Java or Sulawesi. Over sea areas, $\Delta \text{IWC}$ is a combination between the vertical transport of air masses by deep convection during night and early morning over offshore sea and by the westward horizontal transport of air masses near the tropopause, coming from land through coastline areas, during the end of the afternoon and at night (such as in North of Australia seas).

*Data availability.*   The observational datasets are available from the following websites: https://disc.gsfc.nasa.gov/datasets?age=1&keywords= ML2IWC_004 (last access: 1 January 2018, IWC from MLS), https://trmm.gsfc.nasa.gov/publications_dir/publications.html (last access: 1 January 2019, Prec from TRMM-3B42), https://ghrc.nsstc.nasa.gov/lightning/data/data_lis__trmm.html (last access: 1 January 2019, Flash from TRMM-LIS), https://cds.climate.copernicus.eu/cdsapp#!/dataset/reanalysis-era5-pressure-levels-monthly-means?tab=form (last access: July 2019, IWC from ERA5), and https://www.esrl.noaa.gov/psd/thredds/catalog/Datasets/ncep/catalog.html (last access : 17 June

2019, Wind from NOAA).



*Author contributions.* IAD analysed the data, formulated the model and the method combining MLS, TRMM and LIS data and took primary responsibility for writing the paper. CD has treated the LIS data, provided the Figures with Flash datasets, gave advices on data processing and contributed to the Prec and Flash comparative analysis. PR strongly contributed to the design of the study, the interpretation of the results and the writing of the paper. PR, FC, PH and TD provided comments on the paper and contributed to its writing.

*Acknowledgements.* We thank the National Center for Scientific Research (CNRS) and the Excellence Initiative (Idex) of Toulouse, France to fund this study and the project called Turbulence Effects on Active Species in Atmosphere (TEASAO – http://www.legos.obs-mip.fr/ projets/axes-transverses-processus/teasao, Peter Haynes Chair of Attractivy). We would like to thank the teams that have provided the MLS data (https://disc.gsfc.nasa.gov/datasets/ML2IWC_V004, last access: June 2019), the TRMM data (https://pmm.nasa.gov/data-access/ downloads/trmm), the LIS data (https://ghrc.nsstc.nasa.gov/lightning/data/data_lis_trmm.html, last access: June 2019), the ERA5 Reanalysis
data (https://cds.climate.copernicus.eu/cdsapp/dataset/reanalysis\discretionary{-}{}{}era5, last access: June 2019), and the NCEP Reanalysis data provided by the NOAA/OAR/ESRL PSD, Boulder, Colorado, USA, (https://www.esrl.noaa.gov/psd/, last access: June 2019).



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
