# Peer review of "Ice injected into the tropopause by deep convection – Part 2: Over the Maritime Continent"

_Atmospheric Chemistry and Physics, 2019_

## Referee Comment (RC1) · Michelle Santee (Referee) · 11 Nov 2019

This manuscript is a follow-on study from Dion et al. [ACP, 2019], which reported a novel method of correlating the twice-daily measurements of cloud ice water content (IWC) from the Aura Microwave Limb Sounder (MLS) with higher temporal resolution measurements of precipitation (Prec) from the Tropical Rainfall Measurement Mission (TRMM) to reconstruct the diurnal variation of ice in the upper troposphere (UT, 146 hPa) and tropopause level (TL, 100 hPa), thereby estimating the amount of ice injected at those levels by deep convection ($\Delta$IWC). Since the previous study found the largest convective injection of IWC over the Maritime Continent (MariCont), here that region is divided into separate island, sea, and coastal zones. The approach to deriving $\Delta$IWC in the UT and TL from MLS IWC and TRMM Prec data is also applied to TRMM lightning

(Flash) data. Results using both TRMM data sets are compared to those based on IWC from ERA5. Java island is found to be the area with the highest ∆IWC. The roles of small-scale processes in controlling the ∆IWC over the different areas are assessed.

In general, I think that this is a very interesting and valuable paper that demonstrates the great potential of the authors' innovative technique to "fill in" the climatological diurnal cycle of IWC and the estimates of ∆IWC in the UT and TL at $2° \times 2°$ horizontal resolution that have been derived from it. Thus I would very much like to see this paper in print. Unfortunately, however, the manuscript is riddled with inaccurate, erroneous, or inconsistent statements, many instances of unclear wording, and numerous typos. In my opinion, it requires a substantial amount of "cleaning up" before it can be published. A (fairly long) list of specific issues is detailed below. In most cases these concerns can be allayed simply by correcting and clarifying the discussion, with few if any requiring additional analysis. But, although each point is perhaps minor when considered in isolation, in aggregate they add up to major revisions. Moreover, even after the large number of minor corrections listed below, the manuscript will need copy-editing to improve the English.

Specific substantive comments and questions (in sequential order through the manuscript):

L9, L45-46, L105-106: I believe that the representation of the temporal resolution of the TRMM Prec measurements in the Abstract (L9) and Introduction (L45-46) is somewhat misleading. In both places it is stated that Prec data are available at 1-hr resolution. My understanding, however, is that the TRMM-3B42 data are provided as 3-hr averages. Only by taking advantage of the precessing orbit of TRMM and the long study period (13 years) are the authors able to average the data in 1-hr bins. This binning is obliquely alluded to in L105-106 in the TRMM description subsection, but it should be explained more clearly.

L72: Liu & Zipser [2009] is missing from the reference list, but actually it is not the

correct citation here anyway. The 2009 JGR paper did not use TRMM LIS data. A better reference here is Liu & Zipser [JGR, 2005].

Section 2.1: Several aspects of the MLS description require revision. The most significant issue is the implication that the MLS team should have but failed to provide averaging kernels for the IWC measurements (L90-92). This statement and related discussion in Section 2.4 (L130) and Section 7.2 (L383-385) suggest that the authors have misconstrued how the MLS IWC product is derived. In fact, although optimal estimation is used to retrieve almost all other MLS products, that is not the case for IWC, for which a cloud-induced radiance technique is used. Consequently, no averaging kernels are calculated for IWC. It would be appropriate to reference two of the first papers describing and validating the MLS IWC retrievals: Wu et al. [JGR, 2008] and Wu et al. [JGR, 2009]. According to Wu et al. [2008], the IWC measurements represent spatially averaged quantities whose volume can be approximated by a box with dimensions of ~4 km high by ~300 km long; a simple box like this could have been used to degrade the vertical resolution of the ERA5 IWC rather than the unitary triangular function the authors devised, likely leading to slightly different results. Other issues are: (1) Information on the quality of the IWC measurements and the screening steps taken to filter out poor-quality data points should be given. (2) MLS provides IWC measurements at 6 levels in the UTLS, not just at 146 and 100 hPa. (3) Although it is essential to specify the version of the MLS data being used in this study, as written the sentence in L84 makes it sound like it is Version 4.2 of the instrument itself and not the data processing algorithms. (4) It would be appropriate to cite the original paper describing the Aura MLS instrument, Waters et al. [2006], in addition to the MLS Data Quality Document. (5) The most up-to-date version of the latter document is Livesey et al. [2018], not 2017. (6) It might be better to say "horizontal" rather than "spatial" in L92.

Section 2.2: It is stated that TRMM provided observations until 2015 and that the Prec product has been extended through 2019, but the source of the data for the most recent

years is not explained (GPM?). No mention is made of Prec data quality (e.g., biases, random errors).

Section 2.3: Not a single reference for the LIS instrument is cited, nor is there any discussion of data quality, detection limits, etc. I do not understand what is meant by "allowing to observe a point within 90 seconds with a temporal resolution of 2 milliseconds" (L110-111). Within 90 seconds of what?

Section 2.4: As noted by Duncan & Eriksson [ACP, 2018], ERA5 differs from other reanalyses in that it differentiates between precipitating ice, classified as snow water, and non-precipitating ice, classified as cloud ice water. In their study, Duncan & Eriksson typically combined the two products. Presumably only cloud ice water was used here, so it would be good for the authors to comment on whether that approach has any impact on their results. In addition, it might be useful to discuss the conclusions of Duncan & Eriksson regarding the ability of ERA5 to capture both seasonal and diurnal variability in cloud ice.

L134: The statement that ERA5 does not provide winds at 100 and 150 hPa is incorrect.

Section 3: The algebra is backwards here: either the correlation should be flipped in Eqn. (1) or Prec(t) should be multiplied by 1/C in Eqn. (2).

Section 4.2: I am confused about exactly what message Fig. 3 is conveying. As I understand it, a pixel is represented in the maps for 1:30 and 13:30 LT only if it is experiencing the growing phase of convection at that time. Thus all pixels in the map for 1:30 LT are undergoing increasing deep convection then, and likewise for the map at 13:30 LT. The description is ambiguous, but when I read it I assumed that the mean was calculated for each individual pixel, as was done in Fig. 2c and 2e, and not over the MariCont as a whole. If so, then the sign of the deviations from the mean value in a particular pixel indicates whether deep convection is in the early stages (negative) or late stages (positive) of the increasing phase at that time, and the magnitude merely

identifies whether the convection is just getting started or is just about to reach its peak (large) versus whether it is near the middle of the increasing phase (small). If that is the case, then I do not see how the inferences being drawn from this plot are supported. It is stated (L188-189) that the growing phase of convection is mainly over land at 13:30 LT, but colored (i.e., non-grey, if indeed grey is meant to denote pixels for which convection is not ongoing, which is not at all clear) pixels seem to be present over nearly the entire domain in Fig. 3b and 3d, and IWC and, especially, Prec show fairly large anomalies over most of the sea areas. The strongest Prec anomaly at 13:30 LT is stated (L190) to be over Java Island, but (a) that may only mean that convection is not in the middle of the growing phase there, and (b) the one pixel with the largest deviation from the mean over the island of Java does not stand out above the similarly large anomalies in the surrounding seas. It is stated (L190-191) that the strongest Prec anomaly at 1:30 LT occurs over coastlines and coastal seas, but equally large anomalies are seen in several pixels over Borneo and New Guinea. It is stated (L192) that the strongest IWC anomaly at 13:30 LT is located over Java, but again comparably large values are located over North Australia and the North Australian Sea. Finally, the region over the North Australian Sea is identified as having a negative Prec anomaly and a positive IWC anomaly, but that is really only true at 1:30 LT – at 13:30 LT, both anomalies are largely positive in that area.

Section 4.3: The discussion is muddled in places. (1) It is not true that the anomalies of Prec and IWC during the growing phase are positive over the West Sumatran Sea (L207-208); in fact, this area was identified in Section 4.2 to fall into category #2, with positive Prec anomalies but negative IWC anomalies, and this discrepancy is why it is discussed in detail in Section 8.2. (2) In L207, "< 0.15 mg m-3" should be "> 0.15 mm h-1". (3) The sentence in L208-209 doesn't make sense: the quoted TL ∆IWC max and min values overlap (3 and 2-3 mg m-3, respectively), the min value in the TL is clearly much lower than 2 mg m-3 in Fig. 4, and the difference between the values in the TL and the UT is larger than a factor of 3-4 – indeed, it is stated to be a factor of 6 over land on L210. (4) The TL is mentioned in L213, but Fig. 5 shows only the UT.

(5) In L215, it should be "large enough to observe the diurnal cycle of IWC between 2 and 5 mg m-3", not Prec. (6) It is stated (L225) that pixels with large $\Delta$IWC have IWC values between 4.5 and 5.7 mg m-3, but that is not true for New Guinea point #2, for which the IWC is much lower. Moreover, the range of IWC values (1.9 to 4.7 mg m-3) for low-$\Delta$IWC points overlaps that of high-$\Delta$IWC points. Thus, large $\Delta$IWC values are not always associated with large IWC values at 13:30 LT over land, as asserted in L227-228. Nor is it possible on the basis of Fig. 5 to make a similar assertion for 1:30 LT over the seas, since no such cases were actually examined in that figure. (7) L228-229 states "This shows that $\Delta$IWC is strongly correlated with the shape of the diurnal cycle of Prec". But isn't that true by definition, since $\Delta$IWC is simply scaled from the min and max in the diurnal cycle of Prec (Eqn. 3)?

Section 5.2: A number of points need clarification. (1) The discussion throughout this section is inconsistent with Fig. 7, which shows the coastlines of the MariCont in the middle panel, not the bottom one. The figure caption is also incorrect. (2) I think the description of how coastlines are defined is unclear; it would help to say "extending into" rather than "over" the sea in L255. It is clear from previous figures that a number of pixels straddle coastlines – are they categorized into the land or the coastal bins? (3) Liu and Zipser [2008] is not included in the reference list, but it is unlikely to be the correct citation in any case. Perhaps the authors meant Liu et al. [JAMC, 2008], but I am not sure that that paper made the specific points about the diurnal cycles of Prec and Flash being made in L259 and L267. (4) The max in the diurnal cycle of Flash over MariCont_O is stated (L266) to be reached between 4 and 9 LT, but the peak is more like 6-7 LT and values are fairly low by 9 LT. (5) Petersen & Rutledge [2001] is also missing from the reference list. (6) I think that another sentence or two of discussion to put the results of the Love et al. [2011] paper into the context of this study would be helpful.

Section 5.3: (1) Sulawesi is singled out (L301-302) for exhibiting the same onset of the growing phase of convection as Java, but it seems to me that all of the islands in

Fig. 8 show fairly similar timing for the increase in Prec and Flash as Java; rather, it is the declining phase when Sulawesi more closely resembles the steeper decrease over Java than the other islands do. (2) It is stated (L287) that Prec and Flash are studied at $0.25° \times 0.25°$ resolution in this subsection. Therefore, couldn't the fact that the diurnal max in Prec over the 5 small islands in Fig. 8 is much higher than that reported by Dion et al. [2019] over the broad tropical regions of South America, South Africa, and MariCont_L – based on $2°$ bins – merely be a consequence of the much greater horizontal resolution used here? (3) In L323-324, it is stated that Flash reaches a max of only 0.1 flashes h-1 over the North Australian and Bismark Seas, but (a) the value should be 0.1 x 10-3 and (b) it is not true for NAuSea, for which the max is about 0.6 x 10-3 flashes h-1. (4) While the diurnal min in Prec is around 18:00 LT over the Bismark Sea, there are several local min in Flash (8, 14, 18 LT).

Section 6: (1) The duration of the increasing phase of the diurnal cycles of Prec, Flash, and ERA5 IWC is stated (L349) to be 4-5 h over islands, but in L296 this interval for Prec was given as 8-10 h over all land areas besides Java (6 h). (2) Over sea areas, the max of the diurnal cycle of ERA5 IWC is stated (L350) to occur mainly between 7 and 10 LT, but this is not true for the Bismark Sea ($\sim$3 LT), WSumSea (there is another essentially equal peak at 17 UT, as noted in L354-355), or China Sea (16 UT), nor is it true in those cases that the timing is consistent with the max in Prec. The statement that the max in the diurnal cycle comes 2-3 h after that in Flash is inconsistent with what was said in L330-331 (4-7 h). (3) The sentence in L353-355 appears to contradict itself ("consistent with the one of Prec . . . but not with the one of Prec") – perhaps "Flash" was meant in the latter case. (4) Although the comparisons with ERA5 IWC are interesting, I am wondering what the main goal in including them is. Is the intention to use ERA5 IWC, and the $\Delta$IWC estimated from it, to confirm the observationally derived values? Or, conversely, is the idea to use the Prec and Flash to "validate" the new ERA5 values?

Section 7.1: (1) It is very difficult for the reader to judge any of the $\Delta$IWC values stated here in the absence of any y-axis minor tick marks in Fig. 11. (2) It is not clear how

the quoted percentages are being calculated (i.e., relative to what). For example, a range of values of 4.87–6.86 mg m-3 is given for $\Delta$IWC over a subset of islands in the UT. It is then stated (L368) that $\Delta$IWC from Flash is greater than that from Prec by "less than 1.0 mg m-3 (41%)". I have no idea how a value of 41% could possibly have been calculated. (3) I am not convinced that the methodology and measurements employed in this study truly allow $\Delta$IWC to be estimated to three significant digits. (4) The fact that $\Delta$IWC from Flash is almost twice as large as that from Prec over the North Australia Sea is attributed to the lagged diurnal cycle of Flash compared to Prec (L371-372), but (a) this is backwards: it is Prec that is lagged compared to Flash, as noted in L325-326, and (b) I did not follow why a lag in the diurnal cycle would cause larger $\Delta$IWC values. (5) The third paragraph is confusing. It starts with a sentence about Java, but then the rest of the paragraph is about the differences between Prec and Flash $\Delta$IWC estimates in general, making the Java sentence seem out of place. The final sentence is badly written and difficult to parse, but it appears to say that Flash (unlike Prec) is not contaminated by stratiform precipitation and thus should serve as a better proxy than Prec over the sea, but because it is negligible there it cannot be used to calculate $\Delta$IWC in those regions – but of course Flash has been used to do exactly that, with the results shown in Fig. 11. And the statement that Flash is a better proxy for deep convection than Prec because it is not contaminated by stratiform rainfall is repeated in Section 8.2 for the West Sumatra Sea specifically. So the discussion here needs to be clarified.

Section 7.2: (1) the definitions of the UT and TL in L381 are switched. (2) The reference on L384, which was likely supposed to be Rodgers [2000], is missing, but as discussed above it is not relevant here anyway, as optimal estimation is not used to retrieve MLS IWC data. Thus the discussion related to that point needs to be rewritten. (3) There is no Livesey et al. [2019] document – the latest version of the MLS Data Quality Document is Livesey et al. [2018]. (4) It is not clear what is meant by the statement "xx% of variability per study zones", which appears in numerous places throughout this subsection and also in Sections 7.3 and 8.3, nor how those values are calculated.

Please clarify. (5) The convolved ERA5 ΔIWC values are greater than the unconvolved values in the TL, not lower as stated in L398.

Section 7.3: (1) I assume that the statement "observation and reanalysis ΔIWC ranges agree to within 0–0.64 mg m-3" (L404-405) is meant to indicate that the ranges generally overlap, not that the estimates precisely agree. I think it might be clearer to say "observation and reanalysis ΔIWC ranges overlap, except over New Guinea and Sulawesi, where the differences between the extrema of the two ranges are 0.64 mg m-3 and 1.63 mg m-3, respectively". (2) Does the fact that the observational ΔIWC range is more or less consistent with the reanalysis range over most islands but is systematically greater than the reanalysis range over all sea regions imply anything about either the validity of the methodology used here or the reliability of the ERA5 IWC values over offshore areas? (3) The combined ΔIWC range over land in the TL is stated (L408) to be 0.63–3.65 mg m-3, but the lowest value (which occurs over Sumatra) looks smaller than that (below 0.5) to me. Again I question whether the degree of precision in all of the ΔIWC values quoted throughout the manuscript is really supportable. (4) The consistency between ΔIWC estimates is discussed in L410-412. It is not clear to me why Sumatra was left off the list of specific land areas where agreement is good. On the other hand, although MariCont_O is identified as showing large differences, it seems to me that it should be noted that agreement is poor for all individual offshore areas.

Section 8.2: It is stated (L449-450) that Flash is a better proxy for deep convection over the West Sumatra Sea than Prec. I note that Flash shows higher ΔIWC than Prec over the WWS (as in almost all offshore areas) in Fig. 11. But I am puzzled about how the discussion of ΔIWC estimates in this section relates to the negative IWC anomaly in this region in Fig. 3, which is based directly on MLS IWC data, not estimates of ΔIWC derived from either Prec or Flash. More discussion tying the IWC / Prec anomalies of Fig. 3 (and how they differ over the WSS from other regions) to the ΔIWC estimates in Fig. 11 would be helpful here.

Section 8.3: I'm not sure that I follow the discussion in this section. The authors note

that daily mean Flash rates are higher than daily mean Prec values over the North Australian Sea, and that that difference is why ∆IWC estimates from the two sources differ most strongly in that region. They then go on to suggest that IWC injected during the day over North Australia land areas is transported to the coastlines and sea areas overnight. But the bottom-line point of this argument is not clear – what observations presented in this paper is it intended to explain? Are the authors contending that this transport of IWC somehow affects their ∆IWC estimates? That appears to be the case based on the final sentence (L517-519) of the Conclusions section. If so, then I find that very confusing, because the underlying basis for their approach in estimating ∆IWC is the assumption that deep convection is the dominant process driving the diurnal increase in IWC in the TTL and that other processes, such as horizontal advection, can be neglected. If indeed horizontal advection of IWC is a factor here, then wouldn't that mechanism operate in other regions as well? (Even just in the North Australia Sea, it seems that similar contributions from New Guinea might also play a role.) Fundamentally, it seems to me that this has potentially serious implications for the validity of their technique for deriving ∆IWC over any offshore areas that should be discussed in more detail here and stated more explicitly in the Conclusions.

References: (1) There is a pervasive lack of proper capitalization throughout the references listed, as well as several instances of bizarre (and unnecessary) hyphenation. (2) The correct reference for the MLS Data Quality Document is: Livesey, N.J., Read, W.G., Wagner, P.A., Froidevaux, L., Lambert, A., Manney, G.L., Millan, L.F., Pumphrey, H.C., Santee, M.L., Schwartz, M.J., Wang, S., Fuller, R.A., Jarnot, R.F., Knosp, B.W., Martinez, E., and Lay, R.R., Version 4.2x Level 2 data quality and description document, Tech. Rep. JPL D-33509 Rev. D, Jet Propulsion Laboratory, available at: http://mls.jpl.nasa.gov (last access: dd MMM yyyy), 2018.

Minor points of clarification, wording suggestions, and grammar / typo corrections:

L11: lightnings –> lightning events

[Figure]

L14 (also L38, 166, 167,169, 189, 205, 210, 211, 216, 219, 241, 252, 255, 289, 309, 311, 407, 421): lands –> land

L16: I think it would be clearer to add "they agree" in front of "to within 4-20%"

L28: dimentional –> dimensional

L29 and L31: add "e.g." at the beginning of the lists of references on these lines

L35: add a comma after "respectively"

L38: add a comma after "areas"

L41: The first sentence of this paragraph seems out of place, as it has nothing to do with the rest of the paragraph. It would be better to move it somewhere else or delete it.

L53: center –> centers

L56: Is the statement "a comprehensive work has been done around the study of the diurnal cycle of precipitations and convection over the MariCont" referring to previous studies other than Yang & Slingo (cited in the previous sentence)? If so, references are needed. In any case, the sentence needs to be clarified.

L56 (also L104, 430): precipitations –> precipitations

L65: It is not clear what is meant by "the authors were expected"

L73: that will be compared –> and compare

L84: the NASA's –> NASA's

L91: add a comma after "respectively"

L92-93: delete "study" after "resolution"; datasets –> data

L96: has been launched in 1997 and has been able to provide –> was launched in 1997 and provided

L97: composed by –> composed of

L102-103: depend –> depends; add "and does" in front of "not differentiate"

L108: lightnings –> lightning

L109: was using –> used

L123: number –> number of

L129 (also L292, 304, 337, 348, 350, 435): consistently –> consistent

L136: Est –> East

L167 (also L194, 241, 245, 337, 339, 416, 431, 464, 516): the New Guinea –> New Guinea

L175: Fig. 2c –> Fig. 2e

L176 (P7): it would be helpful if the Timor Sea and the Arafura Sea were also indicated on the map in Fig. 2a

P8, Fig. 2 caption: It would be helpful if the information about the horizontal resolution of the TRMM and MLS data were added to the caption in addition to being stated in the main text

L179: that –> as

L180-181: each duration –> the duration; can be defined –> can then be defined

L185: present both in Figs. 3a and b (Figs. 3c and d, respectively) –> present in both sets of Prec and IWC panels in Fig. 3

L187: this doesn't make sense – I think 13:30 LT –> 1:30 LT

L193: is –> are

P10, Fig. 3 caption: It would be good to state in the caption that the IWC plots are for

146 hPa. Also, add a comma after "respectively"

L210: more important –> larger

L232: lightnings are created into –> lightning is created in

L233: lightnings –> lightning

L244: the pervasive lack of superscripts in units (e.g., "month -1", "mg m-3", "m s-1", "mm h-1") is puzzling, given that superscripts are used for other purposes, but it is only a trivial annoyance in most places in the manuscript. In the case of Flash, however, it is a bigger issue, since it is hard to read "10-2- 10-3" in this line. Sometimes the units on Flash are given as per day and sometimes as day-1. Also, I don't think it is true that Flash values are lower than 10-2 per day over New Guinea, at least not in the interior of the island.

P12, Fig. 5: It would be more convenient if the y-axis for Prec had 4 (not 3) minor tick marks, as is the case for the IWC y-axis. The solid and dashed lines should also be described in the caption.

L246: Fig. 2c –> Fig. 2d

L252 (also L264, 269, 314): as noted above, the panels in Fig. 7 are mislabeled

P14, Figure 7: It would be very helpful to have more minor tick marks on the x-axis. In the caption: full line –> solid line; dash –> dashed

L289: delete "areas"

L289-290: add commas after "(2008)" and "1 mm h-1"

L294: add commas after "6h" and "Flash"

L299: it might be good to remind readers that the elevation is shown in Fig. 2b

L302: than Java –> as Java

L304: maintaining –> maintain

L305: rainfalls –> rainfall

L306: convections –> convection

P16-17, Figs. 8 and 9: again, it would be very helpful to have more minor tick marks on the x-axis. Also for Fig. 9, the label for the North Australia Sea is given in panel (b) and the figure caption as "NAusSea", but in the main text it is "NAuSea". The labels should be consistent.

L312-313: it would be clearer to say "either coastline or offshore areas depending on the area"

L320: most of –> most

L332: In the next section –> In Section 7

P19, Fig. 10: It would aid the comparisons with Fig. 2e discussed in the text if the same color bar were used, particularly since the ERA5 IWC values reach higher values than those of MLS, yet the color bar in Fig. 2e extends to larger values. This might also alleviate the issue that the highest values of ERA5 IWC over New Guinea and North Australia appear to saturate the color bar in Fig. 10 (that is, white colors appear in the map in those regions). Also, since panels (c) and (d) have been labeled "TL", it would be good to add "UT" to panels (a) and (b).

L340 (also L342, 347, 366, 373, 377, 391): island –> islands

L344: is –> are

L353: cycles –> cycle; zone –> zones

L363: calculated –> calculate

L366: Fig. 10a –> Fig. 11a

L368: excepted –> except

L370: from –> by

L372: twice greater than –> twice as large as; also delete "values"

L373: Fig. 10b –> Fig. 11b

P21, Fig. 11 caption: West Sumatra Coast (WSS) –> West Sumatra Se (WSS)

L390: why are there parentheses around the convolved ERA5 $\Delta$IWC term in this line?

L406: to within –> by

L410: observational –> observations

L412: negligeable –> negligible

L414: are –> is; twice larger than –> twice as large as

L433: merged –> merging

L433 (also L516): tiny –> small (not only does "tiny" not sound very scientific, but also it could come across as dismissive)

L436: transport –> transports

L437: Fig. 10 shows IWC, not $\Delta$IWC, so it should not be listed with Fig. 4 here; perhaps Fig. 11 was meant instead

L439: section 4.3 –> section 4.2

L442: Awaka (1998) –> (Awaka, 1998); to –> from

L446: "PR" has already been defined in L441

L453: Fig. 10 –> Fig. 11

L462: would be –> is

L479: from by –> from

L481: to impact –> injecting

L482: into –> in

L495: amount –> amounts

L500 (also L504, 517): combination between –> combination of

L510: delete commas after "that" and "Flash"

L513: delete commas after "Guinea" and "Sulawesi"; range –> ranges

L514: as Java –> than Java

L516: cumulus merged –> merged cumulus
* * *

---

## Referee Comment (RC2) · Anonymous Referee #2 · 19 Nov 2019

The paper by Dion et al. is the second part of a work aiming at quantifying the diurnal cycle of ice particles in the tropical tropopause layer (TTL), and more precisely, the amount of ice injected by deep convection up the tropospheric part of the TTL, and up to the tropopause. It is mostly based on the analysis of 13 years of ice water content (IWC) data from MLS onboard the AURA satellite, as well as precipitations data from TRMM, and lightning flashes from the LIS instrument onboard TRMM. While the first part of the study, already published in ACP, is dedicated to the study of all tropical regions over the globe, this companion paper only focuses on the Maritime Continent (MariCont) during the austral convective season of December January and February, because the MariCont has been shown to be one of the most efficient tropical regions to transport ice up to the TTL in Dion et al. (2019). Here the study focuses on each

sub-region of the Maritime Continent, that is the different islands and seas composing it. The main contribution from this study is to present the Maritime Continent not as a whole continent but as the sum of very different contributions. It was already shown in the first paper that the land parts had a different impact and cycle than the oceanic part. Here the authors are going further in estimating the climatological contribution of each main islands and seas of the MariCont. For example, Java and New Guinea are shown to be the main contributors in the transport of ice up to the TTL. In that sense, this innovative work and point of view deserve a publication. Before it can be done, I have major comments and minor comments that should be addressed before the paper can be accepted in ACP. Some of them can be easily addressed by adding explanations and references, some others may require additional calculations.

Major comments: Instrumental part: Information is missing in the description of the satellite products that are used. Most of all, I would expect information on the accuracy, precision, or biases for MLS IWC, TRMM prec, and TRMM LIS (detection limit, false detection of fhashes, etc). See "minor comments" for details.

The use of ERA 5 to estimate DIWC. I have a significant problem with this part. ERA5 is a relatively new reanalysis from ECMWF. The Authors are using ERA5 ice products to compare DIWC from satellite observations and from ERA5. But here, no reference is made on any estimation of the quality of ice from ERA5, nor how the ice is provided or calculated in ERA5. Is ice assimilated in the ECMWF model? If yes, from which instrument? If not, how it is calculated, is there a correlation between the ice product and any reported bias in ECMWF model? As a consequence, is DIWC_ERA5 used to validate DIWC_prec, and DIWC_flash, or should it be understood the other way round? To make a meaningful comparison between both types of estimation, all of this question should be addressed in the manuscript. Furthermore, to my knowledge, the ice in ERA5 is composed of 2 different variables. Are you using the total ice, or only one parameter (which should be non-precipitating ice)? The authors should be more explicit on this point and justify the choice of the variable they have used.

Winds from NCEP. In Section 8, winds from NCEP are used instead of winds of ERA5. It is stated that ERA5 winds are not available. For sure they are. What would be the results shown in Fig 12 if ERA5 winds were used? Does it impact the duration of transport of ice from North Australia land to seas westward?

Minor comments: - Abstract: one of the key highlights of the study is to present the MariCont as a jigsaw puzzle of different contributions with respect to the effect of deep convection on ice in the TTL. For example, Java and New Guinea are presented as very efficient locations for the injection of ice into the TTL. Thus to me, some key findings on the effect of subregions of the MariCont should appear in the abstract. - L11 Lightning is always singular. See also p4 and p11.

- Introduction L31. Jensen et al. (2007) are providing important inputs on the effect of deep convection on the hydration or dehydration in the TTL. It seems appropriate to cite this study here.

- Section 2.1: though the reference for the MLS ice product is given, no information is given on the accuracy and the uncertainties on the IWC. Please add it.

- Section 2.1 L90. This sentence is not clear to me. Please rephrase. At least it should be explained why you need the averaging Kernel at 100 hPa and 146 hPa.

- Section 4.1 L170: there is a very strong contrast in the maximum time of Prec between land and coastal region. If convection and Prec maxima are due to a sea breeze effect and orography over land, as stated before, why coastal region maximum of Prec is clearly not influenced by the sea breeze (otherwise the time of maximum of Prec should not be so different)? On the other hand, the coastal behavior seams relatively independent from the oceanic behavior since the oceanic behavior is very dependent on the sea considered, whereas the Prec maximum for coastal region is relatively well identified. A longer comment or hypothesis should be presented here to explain this behaviour

- Section 4.2 about Fig. 3. From what I understand, the number of occurrences (= cases per pixel that are during the growing phase at 13:30 or 01:30) on which the average/the anomaly is calculated depends on the pixel. What is the amplitude of the number of occurrences to get this figure?

- Section 4 Fig. 4. This is one of the key finding of the publication. However, I am surprised that qualitatively, the same patterns are found at 146 hPa and at 100 hPa, the only difference being the scale. I would expect a slightly different behavior at 100 hPa because the ice amount might be also driven by other processes than just deep convection (e.g. in situ formation of cirrus or ice particles close to the tropopause). Does the result mean that other processes than deep convection are negligible, or cannot be detected by this method or the instrument? A discussion should be given at the end of the comment of Fig. 4.

- Section 4.3 : 228 "this shows that…" Ok but what can we learn about the diurnal cycle or the intensity of deep convection from this correlation?

- Section 5 l231: potential energy –> electric potential energy

- Section 5.2, l254. The choice of 5 pixels over the sea from the land limits: why this choice? Was a sentivity test made on the number of pixel to infer the behavior of coastal regions?

- L255: 10 pixels offshore for oceanic behavior. Please justify.

- P14 fig 7 and results from it. There is a mismatch between the titles of the middle and bottom panels and the corresponding figure captions. Middle panel is entitled MariCon_C and It is captioned MariCont_O. The other way round for the bottom panel. In a general manner, the results for the coasts are relatively close to the one offshore. At least the time shift is weaker between MariCont_O Vs MariCont_C than for MariCont_C Vs. MariCont_L. Is the choice of 5 pixels from the land to define the MariCon_C has something to do with it? What if you had chosen 3 or 2 pixels only? Would the coastal

diurnal cycle of Prec and FLash be closer to the Land cycle?

- P15: References Liu and Zipser (2008) and (2009) appear in the text but not in the reference list.

- Fig 8 and 9. The ERA5 IWC is also presented in the figures and is not commented in section 5. This does not make any problem since it is commented later on in section 6 but at least, a sentence should inform that the ERA5 results would be commented later. About the same figs in section 6, L346: it would be interesting to overplot an equivalent value of the MLS IWC and comment it. As written in my major comment, there no real estimation of the ice product in ERA5. Adding the MLS IWC here could give an idea of a potential bias in the reanalysis. In section 6 the authors comment on the consistency or the inconsistency of the ERA5 IWC diurnal cycle with the Prec one. But no reason or hypothesis are given to explain this disagreement. I wish a discussion appeared at the end of section 6 on that point.

- Section 7. Before reading the whole paper, I did not understand why results from DIWC offshore could be given for flash since it was shown previously that IWC_flash are not synchronous over seas. Thus, one could deduce that the method to estimate IWC (and DIWC) cannot be applied offshore. We understand later that some regions are better described by the IWC_flash approach than by the IWC_Prec, due for example, to the higher contribution of stratiform Prec. So in section 7.1, the part where Fig 11 is presented and commented, it should be justified more clearly why DIWC_prec and DIWC_flash can be presented as a range of observational DIWC. In section 7.2 p 20 and 22. It would be interesting to recall here the number of model levels from 150 to 100 hPa to have an idea of the vertical resolution of the undegraded ERA5 data.

- 7.3 Obviously, a comparison section is needed here, but without describing how ERA5 ice is produced/calculated, the results presented here are meaningless. I do not say it is out of interest to do so. There is probably something to learn in this comparison (for example from the fact that over seas, ERA5 DIWC is systematically lower than the

observational DIWC), but here one must be aware the meaning of the product used.

- L418. Considering the large range of ERA5 to <ERA5>, the numbers given may not be representative.

- Section 8.3 L459. I would have added Corti et al. (2008) for the references concerning Hector.

- Section 8.3 from L465. See my major comment about the NCEP winds.

- L471 and 472. Ice in the UT and at the tropopause is not a passive tracer. So, to state that, an estimation of the lifetime of ice particles for both altitudes should be given.

- L503 "consistent to within 75 % over seas". This corresponds to a relatively fair agreement, and the use of "consistent" seams exaggerated.

- L517. "DIWC is a combination. . ." Again, you can write this statement only if you show that the lifetime of ice particle is long enough for such a transport.

Suggested references: - Jensen, E.J., A.S. Ackerman, and J.A. Smith, 2007: Can overshooting convection dehydrate the tropical tropopause layer? J. Geophys. Res., 112, D11209, doi:10.1029/2006JD007943. - Corti, T., et al. ( 2008), Unprecedented evidence for deep convection hydrating the tropical stratosphere, Geophys. Res. Lett., 35, L10810, doi:10.1029/2008GL033641.

---

## Author Comment (AC1) · 14 Feb 2020

Dear Dr Michelle Santee and Anonymous Referee #2, thank you very much for your very detailed comments that were very helpful for the improvement of our study. We tried as much as possible to answer all of your comments. Please consider in this document your comments in black, our answers in dark blue and the change on the text in clear blue. Pages 1 to 27 present the answers to Dr Michelle Santee's review, and pages 28 to the end present the answers to Anonymous Referee #2 's review.
* * *
Referee #1      Michelle Santee
(Referee) michelle.l.santee@jpl.nasa.gov

This manuscript is a follow-on study from Dion et al. [ACP, 2019], which reported a novel method of correlating the twice-daily measurements of cloud ice water content (IWC) from the Aura Microwave Limb Sounder (MLS) with higher temporal resolution measurements of precipitation (Prec) from the Tropical Rainfall Measurement Mission (TRMM) to reconstruct the diurnal variation of ice in the upper troposphere (UT, 146 hPa) and tropopause level (TL, 100 hPa), thereby estimating the amount of ice injected at those levels by deep convection (ΔIWC). Since the previous study found the largest convective injection of IWC over the Maritime Continent (MariCont), here that region is divided into separate island, sea, and coastal zones. The approach to deriving ΔIWC in the UT and TL from MLS IWC and TRMM Prec data is also applied to TRMM lightning (Flash) data. Results using both TRMM data sets are compared to those based on IWC from ERA5. Java island is found to be the area with the highest ΔIWC. The roles of small-scale processes in controlling the ΔIWC over the different areas are assessed. In general, I think that this is a very interesting and valuable paper that demonstrates the great potential of the authors' innovative technique to "fill in" the climatological diurnal cycle of IWC and the estimates of ΔIWC in the UT and TL at 2◦×2◦ horizontal resolution that have been derived from it. Thus I would very much like to see this paper in print. Unfortunately, however, the manuscript is riddled with inaccurate, erroneous, or inconsistent statements, many instances of unclear wording, and numerous typos. In my opinion, it requires a substantial amount of "cleaning up" before it can be published. A (fairly long) list of specific issues is detailed below. In most cases these concerns can be allayed simply by correcting and clarifying the discussion, with few if any requiring additional analysis. But, although each point is perhaps minor when considered in isolation, in aggregate they add up to major revisions. Moreover, even after the large number of minor corrections listed below, the manuscript will need copy-editing to improve the English.

Specific substantive comments and questions (in sequential order through the manuscript):

L9, L45-46, L105-106: I believe that the representation of the temporal resolution of the

TRMM Prec measurements in the Abstract (L9) and Introduction (L45-46) is somewhat misleading. In both places it is stated that Prec data are available at 1-hr resolution. My understanding, however, is that the TRMM-3B42 data are provided as 3-hr averages. Only by taking advantage of the precessing orbit of TRMM and the long study period (13 years) are the authors able to average the data in 1-hr bins. This binning is obliquely alluded to in L105-106 in the TRMM description subsection, but it should be explained more clearly.

Thank you, we decided to change the TRMM description in L105-106 by :

>  The granule temporal coverage of TRMM-3B42 data is 3 hours, but the temporal resolution of individual measurements is 1 minute. Thus, it is statistically possible to degrade the resolution to 1 hour. TRMM-3B42 are provided in Universal Time that we converted into local time (LT). Details of the binning methodology of TRMM-3B42 is provided by Huffman and Bolvin (2018).

The reference is : G. J. Huffman, D. T. Bolvin, E. J. Nelkin, D. B. Wolff, R. F. Adler, G. Gu, Y. Hong, K. P. Bowman, and E. F. Stocker. The TRMM multisatellite precipitation analysis (TMPA): quasi-global, multiyear, combined-sensor precipitation estimates at fine scales. Journal of hydrometeorology, 8(1):38–55, 2007.

L72: Liu & Zipser [2009] is missing from the reference list, but actually it is not the correct citation here anyway. The 2009 JGR paper did not use TRMM LIS data. Abetter reference here is Liu & Zipser [JGR, 2005].

Thank you, it is a mistake. We wanted to wrote Liu and Zipser (2008) with the following reference:

Liu, C., and Zipser, E. J. (2008), Diurnal cycles of precipitation, clouds, and lightning in the tropics from 9 years of TRMM observations, *Geophys. Res. Lett.*, 35, L04819, doi:10.1029/2007GL032437.

**Section 2.1:**
Several aspects of the MLS description require revision. The most significant issue is the implication that the MLS team should have but failed to provide averaging kernels for the IWC measurements (L90-92). This statement and related discussion in Section 2.4 (L130) and Section 7.2 (L383-385) suggest that the authors have misconstrued how the MLS IWC product is derived. In fact, although optimal estimation is used to retrieve almost all other MLS products, that is not the case for IWC, for which a cloud-induced radiance technique is used. Consequently, no averaging kernels are calculated for IWC. It would be appropriate to reference two of the first papers describing and validating the MLS IWC retrievals: Wu et al. [JGR, 2008] and Wu et al. [JGR, 2009]. According to Wu et al. [2008], the IWC measurements represent spatially averaged quantities whose volume can be approximated by a box with dimensions of∼4 km high by∼300 km long; a simple box like this could have been used to degrade the vertical resolution of the ERA5 IWC rather than the unitary triangular function the authors devised, likely leading to slightly different results.

In agreement with the reviewers' comments, we have used an unitary box function instead of a unitary triangular function to degraded the vertical resolution of ERA5 data.
L383-385, the incriminated sentence has been changed into:

> Consistently with the MLS observations, we have degraded the ERA5 vertical resolution to assess the impact of the vertical resolution on $\Delta IWC^{ERA5}$. According to Wu et al. (2008), $IWC^{MLS}$ estimation derived from MLS represent spatially-averaged quantities within a volume that can be approximated by a box of ~ 300 x 7 x 4 km³ near the pointing tangent height.
> In order to compare $IWC^{MLS}$ and $IWC^{ERA5}$, we degraded: 1) the horizontal resolution of ERA5 from 0.25°x0.25° to 2°x2° and 2) ERA5 data by connecting the vertical profiles of $IWC^{ERA5}$ with a unitary box function whose width is 5 and 4 km at 100 and 146 hPa, respectively.

L. 168 has been changed as follow:

> $IWC^{ERA5}$ have been degraded along the vertical at 100 and 150 hPa ($$) consistently with the MLS vertical resolution of $IWC^{MLS}$ (5 and 4 km at 100 and 146 hPa, respectively) using an unitary box function (see section 7.2).

Other issues are: (1) Information on the quality of the IWC measurements and the screening steps taken to filter out poor-quality data points should be given.
(1) L94, we added the following sentence:
> The IWC measurements were filtered following the recommendations of the MLS team described in Livesey et al. (2018).

(2) MLS provides IWC measurements at 6 levels in the UTLS, not just at 146 and 100 hPa.
(2) L88, the following sentence has been added:
> The Microwave Limb Sounder (MLS, Version 4.2) instrument on board the NASA's Earth Observing System (EOS) Aura platform (Livesey et al., 2017) launched in 2004 provides ice water content ($IWC^{MLS}$, mg m-3) measurements . MLS provides $IWC^{MLS}$ are given at 6 levels in the UTLS (82, 100, 121, 146, 177 and 215 hPa). However, we have chosen to study only two levels: an upper and a lower level of the TTL. Because the level at 82 hPa does not provide enough significant measurements of IWC to have a good signal-to-noise, we have selected 2 levels: 1) at 100 hPa as the uppermost level of the TTL (named TL for tropopause level). Then, the level at 146 hPa has been chosen as the lowermost level of the TTL (named UT for upper troposphere). MLS follows a sun-synchronous near-polar orbit, ...

(3) Although it is essential to specify the version of the MLS data being used in this study, as written the sentence in L84 makes it sound like it is Version 4.2 of the instrument itself

and not the data processing algorithms.
(3) L84 the sentence has been changed into:
    The Microwave Limb Sounder (MLS, data processing algorithm version 4.2) instrument …

(4) It would be appropriate to cite the original paper describing the Aura MLS instrument, Waters et al. [2006], in addition to the MLS Data Quality Document.
(4) L85 the following reference has been added:

    … on board the NASA's Earth Observing System (EOS) Aura platform (Waters et al., 2006; Livesey et al., 2018) launched in 2004

    *Waters, J. W., Froidevaux, L., Harwood, R. S., Jarnot, R. F., Pickett, H. M., Read, W. G., ... & Holden, J. R. (2006). The earth observing system microwave limb sounder (EOS MLS) on the Aura satellite. IEEE Transactions on Geoscience and Remote Sensing, 44(5), 1075-1092.*

(5) The most up-to-date version of the latter document is Livesey et al. [2018], not 2017.
(5) Thank you, it has been changed.

(6) It might be better to say "horizontal" rather than "spatial" in L92.
(6) L92, Thank you, we changed the incriminated word as follow:
    In our study, high horizontal resolution study is now possible because…

**Section 2.2:**
It is stated that TRMM provided observations until 2015 and that the Prec product has been extended through 2019, but the source of the data for the most recent years is not explained (GPM?). No mention is made of Prec data quality (e.g., biases, random errors).

Yes, this is a GPM. L99, we added the following sentences to clarify this point:

    The Tropical Rainfall Measurement Mission (TRMM) has been launched in 1997 and has been able to provide measurements of Prec until 2015. TRMM is composed by five instruments, three of them are complementary sensor rainfall suite (PR, TMI,VIRS). TRMM had an almost circular orbit at 350 km altitude height performing a complete revolution in one and a half hour. Since, the TRMM satellites re-entered the Earth's atmosphere on 2015, the 3B42 algorithm product (TRMM-3B42) (version V7) has been created to estimate the precipitation and extend the precipitation product through 2019. TRMM-3B42 is a multi-satellite precipitation analysis composing a Global Precipitation Measurement (GPM) Mission. TRMM-3B42 is computed from the various precipitation-relevant satellite passive Microwave (PMW) sensors using GPROF2017 computed at the Precipitation Processing System (PPS) (e.g., GMI, DPR, Ku, Ka, Special Sensor Microwave Imager/Sounder [SSMIS], etc.) and including TRMM measurements from 1997 to 2015 (Huffman et al., 2007, 2010; and Huffman and Bolvin, 2018). Work is currently underway with NASA funding to develop more appropriate estimators for random error, and to introduce estimates of bias error (Huffman and Bolvin, 2018).  Prec data are provided  at a 0.25°×0.25°

(~29.2 km) horizontal resolution, extending from 50°S to 50°N (https://pmm.nasa.gov/data-access/downloads/trmm, last access: April 2019).

**Section 2.3:**
Not a single reference for the LIS instrument is cited, nor is there any discussion of data quality, detection limits, etc. I do not understand what is meant by "allowing to observe a point within 90 seconds with a temporal resolution of 2 milliseconds" (L110-111). Within 90 seconds of what?

L110-111, we changed the paragraph by the following one:

The Lightning Imaging Sensor (LIS) aboard of the TRMM satellite measures several parameters relative to lightning. According to Christian et al. (2000), LIS used a Real-Time Event Processor (RTEP) that discriminates lightning event from Earth albedo light. . A lightning event corresponds to the detection of a light anomaly on a pixel representing the most fundamental detection of the sensor. After a spatial and temporal processing, the sensor was able to characterize a flash from several detected events. The instrument detects lightning with storm-scale resolution of 3-6 km (3 km at nadir, 6 km at limb) over a large region (550-550 km) of the Earth's surface. A significant amount of software filtering has gone into the production of science data to maximize the detection efficiency and confidence level. Thus, each datum is a lightning signal and not noise. Furthermore, the weak lightning signals that occur during the day are hard to detect because of background illumination. A real-tile event processor removes the background signal to enable the system and detect weak lightning and achieve a 90% detection efficiency during the day. LIS horizontal resolution is provided at 0.25°×0.10°. LIS is thus able to provide the number of flashes (Flash) measured. The TRMM LIS detection efficiency ranges from 69% near noon to 88% at night. The LIS instrument performed measurements between 1 January 1998 and 8 April 2015. To be as consistent as possible to the MLS and TRMM-3B42 period of study, we are using LIS measurements during DJF from 2004 to 2015.  The observation range of the sensor is between 38°N and 38°S. As LIS is on the TRMM platform, with an orbit that precesses, Flash from LIS can be averaged to obtain the full 24-h diurnal cycle of Flash over the study period with a 1-h temporal resolution. In our study, Flash measured by LIS is studied at 0.25°×0.25° horizontal resolution to be compared to Prec from TRMM-3B42.

Reference added: *Christian, H. J., Blakeslee, R. J., Goodman, S. J., & Mach, D. M. (2000). Algorithm Theoretical Basis Document (ATBD) for the Lightning Imaging Sensor (LIS), 53 pp. NASA/Marshall Space Flight Cent., Alabama.*

**Section 2.4:**
As noted by Duncan & Eriksson [ACP, 2018], ERA5 differs from other reanalyses in that it differentiates between precipitating ice, classified as snow water, and non-precipitating ice,

classified as cloud ice water. In their study, Duncan & Eriksson typically combined the two products. Presumably only cloud ice water was used here, so it would be good for the authors to comment on whether that approach has any impact on their results. In addition, it might be useful to discuss the conclusions of Duncan & Eriksson regarding the ability of ERA5 to capture both seasonal and diurnal variability in cloud ice.

L121-132, we changed sentences in the section 2.4 as follow:

> …. ERA5 provides hourly estimates for a large number atmospheric, ocean and land surface quantities and covers the Earth on a 30 km grid with 137 levels from the surface up to a height of 80 km. Reanalyses such as ERA5 provide a physically constrained, continuous, global, and homogeneous representation of the atmosphere through a large number of observations (space-borne, air-borne, and ground-based) with short-range forecasts. Although there is no direct observation of atmospheric ice content in ERA5, the specific cloud ice water content (mass of condensate / mass of moist air) ($IWC^{ERA5}$) corresponds to the changes in the analysed temperature (and at low levels, humidity) which is mostly driven by the assimilation of temperature-sensitive radiances from satellite instruments (https://cds.climate.copernicus.eu/cdsapp!/dataset/reanalysis-era5-pressure-levels-monthly-means?tab=form, last access: July 2019). $IWC^{ERA5}$ used in our analysis is representative of non-precipitating ice, classified as cloud ice water. Precipitating ice,  classified as snow water, is also provided by ERA5 but not used in this study in order to focus only on the injected and non-precipitating ice into the TTL. Furthermore, results from Duncan and Eriksson (2018) have highlighted that ERA5 is able to capture both seasonal and diurnal variability in cloud ice water but the reanalyses exhibit noisier and higher amplitude diurnal variability than borne out by the satellite estimates. The present study uses the $IWC^{ERA5}$ at 100 and 150 hPa averaged over DJF from 2005 to 2016 with one-hour temporal resolution. ….

L134: The statement that ERA5 does not provide winds at 100 and 150 hPa is incorrect.
     This was a mistake. As explained below (P18 of the present document), we decided to suppress the paragraph 2.5 as well as the Fig. 12.

Section 3: The algebra is backwards here: either the correlation should be flipped in Eqn. (1) or Prec(t) should be multiplied by 1/C in Eqn. (2).

Section 3, thank you. This is a mistake, we changed the equation 1 into:
$$C = IWC_x^{MLS} / Prec_x$$

**Section 4.2:**
I am confused about exactly what message Fig. 3 is conveying. As I understand it, a pixel is represented in the maps for 1:30 and 13:30 LT only if it is experiencing the growing phase of convection at that time. Thus all pixels in the map for 1:30 LT are undergoing increasing deep convection then, and likewise for the map at 13:30 LT. The description is

ambiguous, but when I read it I assumed that the mean was calculated for each individual pixel, as was done in Fig. 2c and 2e, and not over the MariCont as a whole.

Thank you to highlight that it is not clear about which « mean » we are talking about. This is the average of the whole IWC or Prec at 01:30 LT or 13:30 LT over the whole MariCont. Thus, the anomaly (deviation from the mean) shows the areas where Prec of IWC at 01:30 LT or 13:30 LT (per pixel) deviate from the MariCont mean of Prec of IWC at these hours.

In order to detail the explanations about these figures, we propose to describe the Figure 3a only. In Fig. 3a, the Prec values at 01:30 LT are presented as an anomaly (i. e. deviation from the average of the Prec values at 01:30 LT for the entire MariCont area). However, in this figure, it is shown only the pixels where the values from Prec to 01:30 LT are during the increasing phase of convection. Since we know the diurnal cycle of Prec for some pixels, the value of Prec at 01:30 LT is during the decreasing phase of convection. However, we decided to highlight the pixels when 01:30 LT is during the decreasing phase of the convection in grey. Furthermore, pixels with a reddish tending color indicates regions where precipitation (Prec) is greater than the average at 01:30 LT (when observations at 01:30 LT is during the increasing phase of convection). Conversely, pixels with a bluish color indicate regions where there is little precipitation compared to the Prec average at 01:30 LT.
Finally, Figures 3b, c and d are similar to Figure 3a but for Prec at 13:30 LT, IWC at 01:30 LT and IWC at 13:30 LT, respectively

We modified the incriminated sentences (L181) into:

Figures 3a and b present the anomaly (deviation from the mean) of Prec measured by TRMM–3B42 over the MariCont at 01:30 LT and 13:30 LT, respectively, only over pixels when the convection is in the growing phase. The anomaly of IWC measured by MLS over the MariCont is shown in Figs. 3c and d, over pixels when the convection is in the growing phase at 01:30 LT and 13:30 LT, respectively. Thus, e Each pixel of Prec at 01:30 LT or 13:30 LT during the growing phase of the convection deviates from the average of the all Prec at 01:30 LT or 13:30 LT during the growing phase of the convection over the whole MariCont. The gray color denotes pixels for which convection is not ongoing. Some pixels can be presented on both sets of Prec and IWC panels in Figs. 3 when: 1) the onset of the convection is before 01:30 LT and the end is after 13:30 LT or 2) the onset of the convection is before 13:30 LT and the end is after 01:30 LT. Note that, whithin each 2°x2° pixel, at least 60 measurements of Prec or IWC at 13:30 LT or 01:30 LT over the period 2004-2017 have been selected for the average.

The caption of the Fig. 3 has been modified as follow:

Anomaly (deviation from the mean) of Prec (a-b) and Ice Water Content (IWC $_{MLS}$) at 146 hPa (c-d), at 01:30 LT (left) and at 13:30 LT (right) over pixels where 01:30 LT and 13:30 LT are during the growing phase of the convection,

respectively, averaged over the period of DJF 2004-2017. The gray color denotes pixels for which convection is not ongoing.

If so, then the sign of the deviations from the mean value in a particular pixel indicates whether deep convection is in the early stages (negative) or late stages (positive) of the increasing phase at that time, and the magnitude merely identifies whether the convection is just getting started or is just about to reach its peak (large) versus whether it is near the middle of the increasing phase (small). If that is the case, then I do not see how the inferences being drawn from this plot are supported.

We realize that our first explanation of the Fig.3 has not been easy to understand. For that reason, we have better explained the Fig. 3 in the previous answer.

It is stated (L188-189) that the growing phase of convection is mainly over land at 13:30 LT, but colored (i.e., non-grey, if indeed grey is meant to denote pixels for which convection is not ongoing, which is not at all clear) pixels seem to be present over nearly the entire domain in Fig. 3b and 3d, and IWC and, especially, Prec show fairly large anomalies over most of the sea areas.

We clarified this point in the previous answer Section 4.2. Furthermore, the sentence (L188-189) has been changed into:

> At 13:30 LT, t The growing phase of the convection over land is mainly at 13:30 LT while, at 01:30 LT, the growing phase of the convection is mainly over seas and coastlines.

The strongest Prec anomaly at 13:30 LT is stated (L190) to be over Java Island, but (a) that may only mean that convection is not in the middle of the growing phase there,

See answer above (P7-8), explaining that the colors do not refer to the timing of the growing phase but to the anomalies of Prec.

and (b) the one pixel with the largest deviation from the mean over the island of Java does not stand out above the similarly large anomalies in the surrounding seas. It is stated (L190-191) that the strongest Prec anomaly at 1:30 LT occurs over coastlines and coastal seas, but equally large anomalies are seen in several pixels over Borneo and New Guinea.

Our analyses are describing the land and the sea separately. To be clearer, the sentence L190-191 has been changed as follow:

> At 13:30 LT, over land, the strongest Prec and IWC anomalies (+0.15 mm h$^{-1}$ and 2.50 mg m$^{-3}$, respectively) are found over the Java island, (and north of Australia for IWC). At 01:30 LT, the growing phase of the convection is found mainly over sea (while the pixels of the land are mostly gray), with maxima of Prec and IWC anomalies over coastlines and seas close to the coasts such as the Java Sea and the Bismark Sea.

It is stated (L192) that the strongest IWC anomaly at 13:30 LT is located over Java, but again comparably large values are located over North Australia and the North Australian Sea.

The previous answer presents the changes we performed to clarify L192:

Furthermore, L192, we canceled the following sentence :

>

Finally, the region over the North Australian Sea is identified as having a negative Prec anomaly and a positive IWC anomaly, but that is really only true at 1:30 LT – at 13:30 LT, both anomalies are largely positive in that area.

We changed the sentence L197 into:

> … iii) area where Prec anomaly is negative and IWC anomaly is positive (e.g. over the North Australia Sea at 01:30 LT).

**Section 4.3:**
The discussion is muddled in places. (1) It is not true that the anomalies of Prec and IWC during the growing phase are positive over the West Sumatran Sea (L207-208); in fact, this area was identified in Section 4.2 to fall into category #2, with positive Prec anomalies but negative IWC anomalies, and this discrepancy is why it is discussed in detail in Section 8.2.

It is a mistake. It should be North Australia Sea instead of West Sumatra Sea. We changed the incriminated sentence L207 as follow:

> We can note that the anomalies of Prec and IWC during the growing phase of the convection over North Australia Sea at 13:30 LT are positive (> 0.2  mg m$^{-3}$, Fig. 3a and b and > 2.5 mg m$^{-3}$, Fig. 3c and d, respectively).

(2) In L207, "< 0.15 mg m$^{-3}$" should be "> 0.15 mm h$^{-1}$".

This has been corrected.

(3) The sentence in L208-209 doesn't make sense: the quoted TL ΔIWC max and min values overlap (3 and 2-3 mg m$^{-3}$, respectively), the min value in the TL is clearly much lower than 2 mg m$^{-3}$ in Fig. 4, and the difference between the values in the TL and the UT is larger than a factor of 3-4 – indeed, it is stated to be a factor of 6 over land on L210.

We change the min value into 0.2-0.3 mg m$^{-3}$ and the factor values as follow :

In the TL, the maxima (up to 3.0 mg m$^{-3}$) and minima (down to 0.2 – 0.3 mg m$^{-3}$) of ΔIWC are located within the same pixels as in the UT, although 3 to 6 times lower than in the UT.

(4) The TL is mentioned in L213, but Fig. 5 shows only the UT.

The sentence L213 has been changed into:

In order to better understand the impact of deep convection on the strongest ΔIWC injected per pixel up to the TTL, into the UT isolated pixels selected in Fig. 4a are presented separately in Figure 5a and f.

(5) In L215, it should be "large enough to observe the diurnal cycle of IWC between 2 and 5 mg m$^{-3}$", not Prec.

The sentence has been changed.

(But large enough to observe the diurnal cycles of IWC between 2.0 and 5.0 mg m$^{-3}$, Fig. 5g, h, i, j).

(6) It is stated (L225) that pixels with large ΔIWC have IWC values between 4.5 and 5.7 mg m$^{-3}$, but that is not true for New Guinea point #2, for which the IWC is much lower.

We changed the sentence L225:

For pixels with large values of ΔIWC, IWC observed by MLS is between 4.5 and 5.7 mg m$^{-3}$ over North Australia Sea, South Sumatra and New Guinea.

Moreover, the range of IWC values (1.9 to 4.7 mg m$^{-3}$) for low-ΔIWC points overlaps that of high-ΔIWC points. Thus, large ΔIWC values are not always associated with large IWC values at 13:30 LT over land, as asserted in L227-228. Nor is it possible on the basis of Fig. 5 to make a similar assertion for 1:30 LT over the seas, since no such cases were actually examined in that figure.

We changed the sentence L227-228:
To summarize, large values of ΔIWC are observed over land in combination to i) longer growing phase of deep convection (> 9 hours), ii) high value of IWC (>~4.5mg m-3, excepted ) at 13:30 LT over land and 01:30 LT over seas, and/or i ii) large diurnal amplitude of Prec (> 0.5 mm h$^{-1}$).

(7) L228-229 states "This shows that ΔIWC is strongly correlated with the shape of the diurnal cycle of Prec". But isn't that true by definition, since ΔIWC is simply scaled from the min and max in the diurnal cycle of Prec (Eqn. 3)?

We suppressed the incrimitated sentence:

This shows that ΔIWC is strongly correlated with the shape of the diurnal cycle of Prec.

**Section 5.2:**
A number of points need clarification. (1) The discussion throughout this section is inconsistent with Fig. 7, which shows the coastlines of the MariCont in the middle panel, not the bottom one. The figure caption is also incorrect.

We changed the Fig. 7 caption as follow:

Figure 7. Diurnal cycle of Prec (solid line) and diurnal cycle of Flash (dashed line) over MariCont_L (top), MariCont_C (middle) and Mari-Cont_O (bottom).

We changed the L251-253 as follow:

Diurnal cycles of Prec and Flash over the MariCont land, coastlines and offshore (MariCont_L, MariCont_C, MariCont_O, respectively) are shown in Figs. 7a–c, respectively. Within each 0.25°×0.25° bin, land/coast/ocean filters were applied from the Solar Radiation Data (SoDa, \url{http://www.soda-pro.com/web-services/altitude/srtm-in-a-tile}). MariCont\_C is the average of all coastlines defined as 5 pixels extending into the sea from the land limit.

And we changed the organisation of the paragraph L264-275 in order to describe results over coastline before results over ocean as follow:

Over coastlines (Fig. 7c), the Prec diurnal cycle is delayed by about + 2 to 7 h with respect to the Flash diurnal cycle. Prec minimum is around 18:00 LT while Flash minimum is around 11:30 LT. Maxima of Prec and Flash are found around 04:00 LT and 02:00 LT, respectively. This means that the increasing phase of Flash is 2-3 h longer than that of Prec. These results are consistent with Mori et al. (2004) showing a diurnal maximum of precipitation in the early morning between 02:00 LT and 03:00 LT and a diurnal minimum of precipitation around 11:00 LT, over coastal zones of Sumatra. According to Petersen and Rutledge (2001) and Mori et al. (2004), coastal zones are areas where precipitation results more from convective activity than from stratiform activity and the amplitude of diurnal maximum of Prec decreases with the distance from the coastline.
Over offshore areas (Fig. 7b), minima of diurnal cycle of Prec and diurnal cycle of Flash are in the late afternoon, between 16:00 LT and 17:00 LT (Flash) and 17:00 LT and 18:00 LT (Prec), whilst maxima of diurnal cycle of Prec and Flash are reached in the early morning, between 06:00 LT and 07:00 LT (Flash) and around 08:00 LT − 09:00 LT (Prec). Results over offshore areas are consistent with diurnal cycle of Flash and Prec calculated by Liu and Zipser (2008) over the whole tropical ocean,showing the increasing phase of the diurnal cycle of Flash starting 1–2 hours before the increasing phase of the diurnal cycle of Prec.

Finally we changed the sentence (L280-285) as follow:
To summarize, diurnal cycles of Prec and Flash show that:
i) over land, Flash increases proportionally with Prec during the growing phase of the convection,

ii) over coastlines, Flash increasing phase is advanced by more than 6–7 hours compared to Prec increasing phase,
iii) over offshore areas, Flash increasing phase is advanced by about 1–2 hours compared to Prec increasing phase.

(2) I think the description of how coastlines are defined is unclear; it would help to say "extending into" rather than "over" the sea in L255.

We changed the sentence L254-255 as follow:

MariCont-C is the average of all coastlines defined as 5 pixels  extending into the sea from the lands limits.

It is clear from previous figures that a number of pixels straddle coastlines – are they categorized into the land or the coastal bins?

The calculation of the pixel value is not well described. We added the following sentences to make it clearer:

Within each 0.25°×0.25° bin, land/ocean/coast filters were applied from the Solar Radiation Data (SoDa,http://www.soda-pro.com/web-services/altitude/srtm-in-a-tile). MariCont-C is the average of all coastlines defined as 5 pixels  extending into the sea from the lands limits. The MariCont_O is the average of all offshore pixels defined as sea pixels excluding 10 pixels over the sea from the land coasts, thus coastline pixels are excluded, as well as all the coastal influences. MariCont_L is the area of all land pixels. A given 0.25°x0.25° pixel can contain information from different origins : land/coastlines or sea/coastlines. In that case, we can easily discriminate between land and coastlines or sea and coastlines by applying the land/ocean/coastlines filter. Consequently, this particular pixel will be flagged both as land and coastlines or sea and coastlines.

(3) Liu and Zipser [2008] is not included in the reference list, but it is unlikely to be the correct citation in any case. Perhaps the authors meant Liu et al. [JAMC, 2008], but I am not sure that that paper made the specific points about the diurnal cycles of Prec and Flash being made in L259 and L267.

Liu and Zipser [2008] has been forgotten in the reference list. In this paper, authors made similar comparison between diurnal cycle of Prec and Flash showing consistent results. We included them in the reference list:
*Liu, C., and Zipser, E. J. ( 2008), Diurnal cycles of precipitation, clouds, and lightning in the tropics from 9 years of TRMM observations, Geophys. Res. Lett., 35, L04819, doi:10.1029/2007GL032437.*

(4) The max in the diurnal cycle of Flash over MariCont_O is stated (L266) to be reached between 4 and 9 LT, but the peak is more like 6-7 LT and values are fairly low by 9 LT.

We changed the sentence L266 :

Over offshore areas (Fig. 7b), minima of diurnal cycle of Prec and diurnal cycle of Flash are in the late afternoon, between 16:00 LT and 17:00 LT (Flash) and 17:00 LT and 18:00 LT (Prec), whilst maxima of diurnal cycle of Prec and Flash are reached in the early morning, between 06:00 LT and 07:00 LT (Flash) and around 08:00 LT – 09:00 LT (Prec).

(5) Petersen & Rutledge [2001] is also missing from the reference list.

We inserted them in the reference list:

Petersen, W. A., & Rutledge, S. A. (2001). Regional variability in tropical convection: Observations from TRMM. *Journal of Climate*, *14*(17), 3566-3586.

(6) I think that another sentence or two of discussion to put the results of the Love et al. [2011] paper into the context of this study would be helpful.

We updated the incriminated paragraph:

The time of transition from maximum to minimum of Prec is always longer than that of Flash. The period after the maximum of Prec is likely more representative of stratiform rainfall than deep convective rainfall. Consistently, over the MariCont ocean, model results from Love et al. (2011) have shown the suppression of the deep convection over offshore area in West of Sumatra from the early afternoon due to downwelling wavefront highlighted by deep warm anomalies around noon. According to the authors, later in the afternoon, gravity waves are forced by the stratiform heating profile and propagate slowly offshore. They also highlighted that the diurnal cycle of the offshore convection responds strongly to the gravity wave forcing at the horizontal scale of 4 km.

**Section 5.3:**
(1) Sulawesi is singled out (L301-302) for exhibiting the same onset of the growing phase of convection as Java, but it seems to me that all of the islands in Fig. 8 show fairly similar timing for the increase in Prec and Flash as Java; rather, it is the declining phase when Sulawesi more closely resembles the steeper decrease over Java than the other islands do.

We changed the incriminated sentence L301-302 into:

Sulawesi is also a small islandand presents the same onset of growing phase for the convection than Java, consistently with results presented in Nesbittand Zipser (2003) and Qian (2008).
Sulawesi is also a small island with high topography as Java. However, the amplitude of the diurnal cycle of Prec and Flash over Sulawesi is not as strong as over Java.

(2) It is stated (L287) that Prec and Flash are studied at 0.25°×0.25° resolution in this subsection. Therefore, couldn't the fact that the diurnal max in Prec over the 5 small islands in Fig. 8 is much higher than that reported by Dion et al. [2019] over the broad

tropical regions of South America, SouthAfrica, and MariCont_L – based on 2° bins – merely be a consequence of the much greater horizontal resolution used here?

You are right, thank you. We modified the following sentence L295-297 into:
> The particularity of Java is related to the increasing phase of the diurnal cycle of Prec (6 h), that is faster than over all the other land areas considered in our study (7 – 8 h).

(3) In L323-324, it is stated that Flash reaches a max of only 0.1 flashes $h^{-1}$ over the North Australian and Bismark Seas, but (a) the value should be 0.1 x $10^{-3}$ and (b) it is not true for NAuSea, for which the max is about 0.6 x $10^{-3}$ flashes $h^{-1}$.

We corrected L323-324 as follow:

> Over China Sea and Bismark Sea, the diurnal cycle of Flash shows a  weak amplitude with maxima reaching only 0.1-0.2 x$10^{-3}$ flashes $h^{-1}$.

(4) While the diurnal min in Prec is around 18:00 LT over the Bismark Sea, there are several local min in Flash (8, 14, 18 LT).

We changed the sentence L324-325 by the following one:
> ~~However, the diurnal minima of Prec and Flash over Bismark Sea are found to be at the same time (~17:00 LT).~~
> Furthermore, over the Bismark Sea, while the diurnal minimum in Prec is around 18:00 LT, there are several local minima in Flash (08:00, 14:00 and 18:00 LT).

**Section 6:**
(1) The duration of the increasing phase of the diurnal cycles of Prec, Flash, and ERA5 IWC is stated (L349) to be 4-5 h over islands, but in L296 this interval for Prec was given as 8-10 h over all land areas besides Java (6 h).

We firstly corrected L296 as follow:

> The particularity of Java is related to the increasing phase of the diurnal cycle of Prec (6 h) that is faster than over all the other land areas considered in our study (7 – 8 h).

Then, we changed L349 as follow:

> The duration of the increasing phase of the diurnal cycles of Prec, Flash and IWCERA5 are all consistent to each other (6 – 8 h).

(2) Over sea areas, the max of the diurnal cycle of ERA5 IWC is stated (L350) to occur mainly between 7 and 10 LT, but this is not true for the Bismark Sea (~3 LT), WSumSea (there is another essentially equal peak at 17 UT, as noted in L354-355), or China Sea (16

UT), nor is it true in those cases that the timing is consistent with the max in Prec. The statement that the max in the diurnal cycle comes 2-3 h after that in Flash is in consistent with what was said in L330-331 (4-7 h).

We changed L349-351 as follow:

> Over sea (Fig. 9), the maximum of the diurnal cycle of IWC$^{ERA5}$ is mainly found between 07:00 LT and 10:00 LT over Java Sea and North Australia Sea, consistently with the diurnal cycle of Prec and a second peak is found around 16:00 LT.  Thus, the duration of the increasing phase of the diurnal cycles of IWC$^{ERA5}$ is consistent with the one of Prec over these two sea study zones (~10 hours), but not with the one of Flash. Over Bismark Sea, the diurnal maxima of IWC$^{ERA5}$ are found at 04:00 LT with a second peak later at noon. Over West Sumatra Sea, two diurnal maxima are found at 08:00 LT and 17:00 LT. Over China Sea, the diurnal maxima of IWC$^{ERA5}$ are found at 16:00 LT with a second peak at 08:00 LT.

(3) The sentence in L353-355 appears to contradict itself ("consistent with the one of Prec...but not with the one of Prec") – perhaps "Flash" was meant in the latter case.

Yes, it should be 'Flash' instead of 'Prec'. The incriminated sentence has been corrected.

(4) Although the comparisons with ERA5 IWC are interesting, I am wondering what the main goal in including them is. Is the intention to use ERA5IWC, and the ΔIWC estimated from it, to confirm the observationally derived values? Or, conversely, is the idea to use the Prec and Flash to "validate" the new ERA5 values?

A sentence has been added L335 to clarify the motivations to use IWC from ERA5:

> The ERA5 reanalyses provide hourly IWC at 150 and 100 hPa (IWC$^{ERA5}$). The diurnal cycle of IWC over the MariCont from ERA5 will be used to calculate ΔIWC from ERA5 in order to assess the horizontal distribution and the amount of ice injected in the UT and the TL deduced from our model combining MLS ice and TRMM Prec or MLS ice and LIS flash. Figures 10a, b, c and d present …

 **Section 7.1:**
(1) It is very difficult for the reader to judge any of the ΔIWC values stated here in the absence of any y-axis minor tick marks in Fig. 11.

The Fig. 11 has been changed as follow:

[Figure]

[Figure]

(2) It is not clear how the quoted percentages are being calculated (i.e., relative to what). For example, a range of values of 4.87–6.86 mg m$^{-3}$ is given for ΔIWC over a subset of islands in the UT. It is then stated (L368) that ΔIWC from Flash is greater than that from Prec by "less than 1.0 mg m$^{-3}$ (41%)". I have no idea how a value of 41% could possibly have been calculated.

The percentage has been recalculated as follow: for each land study zone, the percentage is calculated as the difference between the Flash value minus the Prec value divided by the Flash value. For each study zone, results are detailled as follow:
MCL=22%, Sum = 4%, Bor = 11%, Java=-8%, Sul = 9% and NG =19% over island and MCO=43 %, WSS=50 %, ChinS=31 %, JS =-7 %, NAS=53 % and BS = 43 % over sea).
As a consequence, the previous sentences L367-372 have been changed as follow:

> ΔIWC$^{Flash}$ is generally greater than ΔIWC$^{Prec}$ by less than 1.0 mg m$^{-3}$ (((ΔIWC$^{Flash}$-ΔIWC$^{Prec}$)/ΔIWC$^{Flash}$)x100 ranges from 4 to 22%)  for all the islands, except  for Java where ΔIWC$^{Prec}$ is larger than ΔIWC$^{Flash}$ by 0.7 mg m-3 (-8%).  Over sea, ΔIWC varies from 1.72 to 4.74 mg m$^{-3}$. ΔIWC$^{Flash}$ is greater than ΔIWC$^{Prec}$ from 0.6 to 2.1 mg m$^{-3}$ (31-5%), except for Java Sea, where ΔIWC$^{Prec}$ is greater than ΔIWC$^{Flash}$ by 0.2 mg m$^{-3}$ (-7%). Over North Australia Sea,  ΔIWC$^{Flash}$ is almost twice as large as than ΔIWC$^{Prec}$  (53%).

(3) I am not convinced that the methodology and measurements employed in this study truly allow ΔIWC to be estimated to three significant digits.

We changed all results to two significant digits.

(4) The fact that ΔIWC from Flash is almost twice as large as that from Prec over the North Australia Sea is attributed to the lagged diurnal cycle of Flash compared to Prec (L371-372), but (a) this is backward: it is Prec that is lagged compared to Flash, as noted in L325-326, and (b) I did not follow why a lag in the diurnal cycle would cause larger ΔIWC values.

You are right, there is no specific reason why a lag in the diurnal cycle would cause larger ΔIWC values. We deleted the part of the sentence L371-372:

(5) The third paragraph is confusing. It starts with a sentence about Java, but then the rest of the paragraph is about the differences between Prec and Flash ΔIWC estimates in general, making the Java sentence seem out of place. The final sentence is badly written and difficult to parse, but it appears to say that Flash (unlike Prec) is not contaminated by stratiform precipitation and thus should serve as a better proxy than Prec over the sea, but because it is negligible there it cannot be used to calculate ΔIWC in those regions – but of course Flash has been used to do exactly that, with the results shown in Fig. 11. And the statement that Flash is a better proxy for deep convection than Prec because it is not contaminated by stratiform rainfall is repeated in Section 8.2 for the West Sumatra Sea specifically. So the discussion here needs to be clarified.

We clarified the last paragraph (L376-379) as below and replaced the last sentence which was confusing by a new one concluding that both proxies can be used in our model, with more confidence over land:

> At both altitudes, To summarize, independently of the proxies used for the calculation of ΔIWC, and at both altitudes, Java shows the largest injection of ice over the MariCont. ΔIWC$^{Prec}$ and ΔIWC$^{Flash}$ are consistent to within 4-20 \% over islands and 6-50 \% over seas in the UT and the TL. Furthermore, it has been shown that both proxies can be used in our model, with more confidence over land: IWC$^{Prec}$ and ΔIWC$^{Flash}$ are consistent to each other to within 4-22% over island and 7-53% over sea in the UT and the TL. The largest difference over sea is probably due to the larger contamination of stratiform precipitation included in Prec over sea. Although Flash, is not contaminated by stratiform clouds, it could be a better proxy than Prec over sea but it is unfortunately negligible: less than 10-2 flashes per day (Fig. 6).

**Section 7.2:**
(1) the definitions of the UT and TL in L381 are switched.

The definitions have been corrected:

> ΔIWC from ERA5 (ΔIWC$^{ERA5}$) is calculated in the UT and the TL ($z_0$= 150 and 100 hPa, respectively) as the max–min difference in the amplitude of the diurnal cycle.

(2) The reference on L384, which was likely supposed to be Rodgers [2000], is missing, but as discussed above it is not relevant here anyway, as optimal estimation is not used to retrieve MLSIWC data. Thus the discussion related to that point needs to be rewritten.

Correction has been done and answers has been detailed previously page 3 of this document.

(3) There is no Livesey et al. [2019] document – the latest version of the MLS Data Quality Document is Livesey et al. [2018].

The reference has been changed.

(4) It is not clear what is meant by the statement "xx% of variability per study zones", which appears in numerous places throughout this subsection and also in Sections 7.3 and 8.3, nor how those values are calculated. Please clarify.

Similarly as the percentage previously calculated in section 7.1 (comparing $\Delta IWC^{Prec}$ and $\Delta IWC^{Flash}$), the percentage calculated here is an average of all percentages of difference between $\Delta IWC^{ERA5}$ and $<\Delta IWC^{ERA5}>$ calculated for each study zone as follow: $((\Delta IWC^{ERA5}-<\Delta IWC^{ERA5}>/\Delta IWC^{ERA5})\times100)$. For each study zone, results over land are :
MCL=21%, Sum=21%, Bor=27%, Java=22%, Sul=23%, NG=23%.

To clarify, this we changed the paragraph as follow:

> Figure 11 shows $\Delta IWC^{ERA5}$ and $(\langle\Delta IWC^{ERA5}z_0\rangle)$ at $z_0 = 150$ and 100 hPa, over the island and the sea study zones. In the UT (Fig. 11a), over island, $\Delta IWC^{ERA5}_{150}$ and $\langle\Delta IWC^{ERA5}_{150}\rangle$ calculated over Sumatra and Borneo vary from  4.9 to  7.0 mg m$^{-3}$ (the relative variation calculated as $((\Delta IWC^{ERA5}-<\Delta IWC^{ERA5}>)/\Delta IWC^{ERA5})\times100$ is 18-19%)  whilst $\Delta IWC^{ERA5}_{150}$ and $\langle\Delta IWC^{ERA5}_{150}\rangle$ over Java, Sulawesi and New Guinea reach  7.4-10.0 mg m$^{-3}$ (~19-22% of variability per study zone).

(5) The convolved ERA5$\Delta IWC$ values are greater than the unconvolved values in the TL, not lower as stated in L398.

The sentence has been corrected. :

> In the TL, over land, $\Delta IWC^{ERA5}_{100}$ and $\langle\Delta IWC^{ERA5}_{100}\rangle$ vary from  0.5 to  3.7 mg m$^{-3}$ (~68% of variability per study zone) with $\langle\Delta IWC^{ERA5}_{100}\rangle$ being larger than $\Delta IWC^{ERA5}_{100}$ by less than 2.1 mg m$^{-3}$.

**Section 7.3:**
(1) I assume that the statement "observation and reanalysis $\Delta IWC$ ranges agree to within 0–0.64 mg m$^{-3}$" (L404-405) is meant to indicate that the ranges generally overlap, not that the estimates precisely agree. I think it might be clearer to say "observation and reanalysis $\Delta IWC$ ranges overlap, except over New Guinea and Sulawesi, where the differences between the extrema of the two ranges are 0.64 mg m$^{-3}$ and 1.63 mg m$^{-3}$, respectively".

The sentence L403-405 has been changed as follow :

> The comparison between the observational $\Delta IWC$ range and the reanalysis $\Delta IWC$ range is presented in Fig. 11. In the UT, over land, observation and reanalysis $\Delta IWC$ ranges overlap (agree to within 0.1 to 1.0 mg m$^{-3}$), which

highlights the robustness of our model over land, except over Sulawesi and New Guinea, where the observational ΔIWC range and the reanalysis ΔIWC range differ by 1.7 and 0.7 mg m$^{-3}$, respectively).

(2) Does the fact that the observational ΔIWC range is more or less consistent with the reanalysis range over most islands but is systematically greater than the reanalysis range over all sea regions imply anything about either the validity of the methodology used here or the reliability of the ERA5 IWC values over offshore areas?

As noted in the sentence L403-405, these results highlighted the robustness of our model over land, but over sea, a systematic positive bias and a too large variability range were depicted in our model compared to ERA5.

To clarify, the sentence L405 has been completed as follow:

Over sea, the observational ΔIWC range is systematically greater than the reanalysis ΔIWC range to within ~1.0~~0~ – 2.2 mg m$^{-3}$ (75 %), showing a systematic positive bias and a too large variability range in our model over sea compared to ERA5.

(3) The combined ΔIWC range over land in the TL is stated (L408) to be 0.63–3.65 mg m$^{-3}$, but the lowest value (which occurs over Sumatra) looks smaller than that (below 0.5) to me. Again I question whether the degree of precision in all of the ΔIWC values quoted throughout the manuscript is really supportable.

Thank you. We checked all the values quoted in the manuscript. We changed the value of 0.6 (which is the minimum for the MCL study zones) to 0.5 which is the minimum among all the study zones. Here are the changes:

Combining observational and reanalysis ranges, the total ΔIWC variation range is estimated in the UT between  4.2 and 10.0 mg m$^{-3}$ (~20% of variability per study zone) over land and between 0.4 and 4.4 mg m$^{-3}$ (~30% of variability per study zone) over sea and, in the TL, between 0.5 and 3.7 mg m$^{-3}$ (~70% of variability per study zone) over land, and between 0.1 and 0.7 mg m$^{-3}$ (~80 % of variability per study zone) over sea.

(4) The consistency between ΔIWC estimates is discussed in L410-412. It is not clear to me why Sumatra was left off the list of specific land areas where agreement is good. On the other hand, although MariCont_O is identified as showing large differences, it seems to me that it should be noted that agreement is poor for all individual offshore areas.

Sumatra has been forgotten in the sentence L140-412. The sentence L410-412 has been modified into:

The amounts of ice injected in the UT deduced from observations and reanalyses are consistent to each other over MariCont_L, Sumatra, Borneo and Java, with significant differences  over Sulawesi, New Guinea (within 1.7 to 0.7 mg m$^{-3}$, respectively) and all individual offshore study zones (within 0.7 to 2.1 mg m$^{-3}$).

**Section 8.2:** It is stated (L449-450) that Flash is a better proxy for deep convection over the West Sumatra Sea than Prec. I note that Flash shows higher ΔIWC than Prec over the WWS (as in almost all offshore areas) in Fig. 11. But I am puzzled about how the discussion of ΔIWC estimates in this section relates to the negative IWC anomaly in this region in Fig. 3, which is based directly on MLS IWC data, not estimates of ΔIWC derived from either Prec or Flash. More discussion tying the IWC / Prec anomalies of Fig. 3 (and how they differ over the WSS from other regions) to the ΔIWC estimates in Fig. 11 would be helpful here.

The whole paragraph has been changed for a better clarification:

In section 4.3, it has been shown that the West Sumatra Sea is an area with positive anomaly of Prec during the growing phase of the convection but negative anomaly of IWC, which differs from other places.  These results suggest that Prec is representative not only of convective precipitation but also of stratiform precipitation. The diurnal cycle of stratiform and convective precipitations over West Sumatra Sea has been studied by Mori et al. (2004) using 3 years of TRMM precipitation radar (PR) datasets, following the 2A23Algorithm (Awaka, 1998). The authors have shown that rainfall over Sumatra is characterized by convective activity with a diurnal maximum between 15:00 and 22:00 LT while, over the West Sumatra Sea, the rainfall type is convective and stratiform, with a diurnal maximum during the early morning (as observed in Fig. 9). Furthermore, their analyses have shown a strong diurnal cycle of 200-hPa wind, humidity and stability, consistent with the diurnal cycle of precipitation measured by TRMM Precipitation Radar (PR) over Sumatra West Sea and Sumatra island. Stratiform and convective clouds are both at the origin of heavy rainfall in the tropics (Houze and Betts, 1981; Nesbitt and Zipser, 2003) and in the West Sumatra Sea, but stratiform clouds are mid-altitude clouds in the troposphere and do not transport ice up to the tropopause. Thus, over the West Sumatra Sea, the calculation of ΔIWC estimated from Prec is possibly overestimated because Prec include a non-negligible amount of stratiform precipitation over this area.

**Section 8.3:** I'm not sure that I follow the discussion in this section. The authors note that daily mean Flash rates are higher than daily mean Prec values over the North Australian Sea, and that difference is why ΔIWC estimates from the two sources differ most strongly in that region. They then go on to suggest that IWC injected during the day over North Australia land areas is transported to the coastlines and sea areas over night. But the bottom-line point of this argument is not clear – what observations presented in this paper is it intended to explain? Are the authors contending that this transport of IWC some how affects their ΔIWC estimates? That appears to be the case based on the final sentence

(L517-519) of the Conclusions section. If so, then I find that very confusing, because the underlying basis for their approach in estimating ΔIWC is the assumption that deep convection is the dominant process driving the diurnal increase in IWC in the TTL and that other processes, such as horizontal advection, can be neglected. If indeed horizontal advection of IWC is a factor here, then wouldn't that mechanism operate in other regions as well? (Even just in the North Australia Sea, it seems that similar contributions from New Guinea might also play a role.). Fundamentally, it seems to me that this has potentially serious implications for the validity of their technique for deriving ΔIWC over any offshore areas that should be discussed in more detail here and stated more explicitly in the Conclusions.

We deleted Fig. 12 and paragraph 2.5. We decided not to discuss about the assumptions of the impact of processes than the vertical deep convective processes on the ice injected over offshore because we do not have enough information to assess their implications and make hypothese. Thus, we changed and simplified the whole paragraph as follow:

> ### 8.3 North Australia Sea and seas with nearby islands
>
> The comparisons between Figs. 2c and 6a have shown strong daily mean of Flash ($10^{-2}$ –$10^{-1}$ flashes day$^{-1}$) but low daily mean of Prec (2.0 – 8.0 mm day$^{-1}$) over the North Australia Sea. Additionally, Fig. 11 shows that the strongest differences between ΔIWC$^{Prec}$ and ΔIWC$^{Flash}$ are found over the North Australia Sea, with ΔIWC$^{Flash}$ greater than ΔIWC$^{Prec}$ by 2.3 mg m$^{-3}$ in the UT and by 0.4 mg m$^{-3}$ in the TL (53% of variability between ΔIWC$^{Flash}$ and ΔIWC$^{Prec}$). These results imply that the variability range in our model is too large and highlight the difficulty to estimate ΔIWC over this study zone.
> Furthermore, as for Java Sea or Bismarck Sea, North Australia Sea has the particularity to be surrounding by several islands. According to the study from Pope et al. (2009), the cloud size is the largest during the afternoon over the North Australia land, during the night over North Australia coastline and during the early morning over the North Australia sea. These results suggest that deep convective activity moves from the land to the sea during the night. Over the North Australia Sea, it seems that the deep convective clouds are mainly composed by storms with lightnings but precipitations are weak or do not reach the surface and evaporating before.

Furthermore, the sentence L517 has been deleted.

**References:**

(1) There is a pervasive lack of proper capitalization throughout the references listed, as well as several instances of bizarre (and unnecessary) hyphenation. Done

(2) The correct reference for the MLS Data Quality Document is: Livesey, N.J., Read,W.G., Wagner, P.A., Froidevaux, L., Lambert, A., Manney, G.L., Millan, L.F., Pumphrey,H.C., Santee, M.L., Schwartz, M.J., Wang, S., Fuller, R.A., Jarnot, R.F., Knosp, B.W.,Martinez, E., and Lay, R.R., Version 4.2x Level 2 data quality and description document, Tech. Rep.

JPL D-33509 Rev. D, Jet Propulsion Laboratory, available at:http://mls.jpl.nasa.gov (last access: dd MMM yyyy), 2018. Done

Minor points of clarification, wording suggestions, and grammar / typo corrections:

L11: lightnings –> lightning events OK

L14 (also L38, 166, 167,169, 189, 205, 210, 211, 216, 219, 241, 252, 255, 289, 309,311, 407, 421): lands –> land OK

L16: I think it would be clearer to add "they agree" in front of "to within 4-20%" OK

L28: dimentional –> dimensional OK

L29 and L31: add "e.g." at the beginning of the lists of references on these lines OK

L35: add a comma after "respectively"L38: add a comma after "areas" OK

L41: The first sentence of this paragraph seems out of place, as it has nothing to dowith the rest of the paragraph. It would be better to move it somewhere else or delete it.

OK we deleted it.

L53: center –> centers  OK

L56: Is the statement "a comprehensive work has been done around the study of the diurnal cycle of precipitations and convection over the MariCont" referring to previous studies other than Yang & Slingo (cited in the previous sentence)? If so, references are needed. In any case, the sentence needs to be clarified.  OK we added the missing reference as follow:

> Yang and Slingo (2001) have shown that over the Indonesian area, the phase of the convective activity diurnal cycle drifts from land to coastlines and to offshore areas. Even though authors have done a comprehensive work around the study of the diurnal cycle of precipitation and convection over the MariCont, the diurnal cycle of ice injected by deep convection up to the TL over this region is still not well understood.

L56 (also L104, 430): precipitations –> precipitations  OK

L65: It is not clear what is meant by "the authors were expected"

We clarified this point as follow:

> Consequently, the amount of ice injected in the UT and the TL is greater over MariCont_L than over MariCont_O. Considering a higher horizontal resolution over small islands and seas of the MariCont and investigating other proxies of deep convection, the authors were expected a better characterisation of the amount of ice injected up to the TTL .
> Building upon the results of Dion et al. (2019), the present study is

addressing the evaluation of ΔIWC at a resolution of the present study aims to improve the methodology of Dion et al. (2019) by i) studying smaller study zones than in Dion et al. (2019) and by distinguishing island and sea of the MariCont, ii) assessing the sensibility of model to different proxies of deep convection and iii) assessing the amount of ice injected in the UT and the TL inferred by our model by comparinf with that of ERA5 reanalyses. Based on space-borne observations and meteorological reanalyses, ΔIWC is provided at a horizontal resolution of 2°×2° over 5 islands (Sumatra, Borneo, Java, Sulawesi and New Guinea) and 5 seas (West Sumatra Sea, Java Sea, China Sea,North Australia Sea, and Bismark Sea) of the MariCont during convective season (December, January and February, here after DJF) from 2004 to 2017. ΔIWC will be first estimated from Prec measured by TRMM-3B42. A sensitivity study of ΔIWC based on the number of flashes (Flash) detected by the TRMM Lightning Imaging Sensor (TRMM-LIS), an alternative proxy for deep convection as shown by Liu and Zipser (2009), is secondly proposed. Finally, we will use IWC calculated by the ERA5 reanalyses from 2005 to 2016 to estimate ΔIWC in the UT and the TL over each study zone and compare it to ΔIWC estimated from Prec and Flash.

L73: that will be compared –> and compare  OK
L84: the NASA's –> NASA's  OK
L91: add a comma after "respectively"  OK
L92-93: delete "study" after "resolution"; datasets –> data OK

L96: has been launched in 1997 and has been able to provide –> was launched in 1997 and provided OK
L97: composed by –> composed of OK
L102-103: depend –> depends; add "and does" in front of "not differentiate" OK
L108: lightnings –> lightning OK
L109: was using –> used OK
L123: number –> number of OK
L129 (also L292, 304, 337, 348, 350, 435): consistently –> consistent OK
L136: Est –> East OK
L167 (also L194, 241, 245, 337, 339, 416, 431, 464, 516): the New Guinea –> NewGuinea OK
L175: Fig. 2c –> Fig. 2[e] OK
L176 (P7): it would be helpful if the Timor Sea and the Arafura Sea were also indicated on the map in Fig. 2a

These two seas compose what we named the North Australia Sea. However, the map on Fig. 2a is too small to show these two seas separately.

P8, Fig. 2 caption: It would be helpful if the information about the horizontal resolution of the TRMM and MLS data were added to the caption in addition to being stated in the main text

The caption has been changed as follow:

Main islands and seas of the MariCont (S is for Sumatra) (a), elevation from Solar Radiation Data (SoDa) (b); daily mean of Prec measured by TRMM over

the Maritime Continent, averaged over the period of DJF 2004-2017 (c), hour (local solar time (LST)) of the diurnal maxima of Prec over the MariCont (d); daily mean (01:30 LT + 13:30 LT)/2 of IWC$^{MLS}$ at 146 hPa from MLS over the MariCont averaged over the period of DJF 2004-2017 (e). Observations are presented with a horizontal resolution is 0.25°x0.25° for (b, c and d) and 2°x2° for (e).

L179: that –> as OK
L180-181: each duration –> the duration; can be defined –> can then be defined OK
L185: present both in Figs. 3a and b (Figs. 3c and d,  OK

L187: this doesn't make sense – I think 13:30 LT –> 1:30 LT OK

L193: is –> are OK
P10, Fig. 3 caption: It would be good to state in the caption that the IWC plots are for 146 hPa. Also, add a comma after "respectively"

OK :

Anomaly (deviation from the mean) of Prec (a-b) and Ice Water Content (IWC$^{MLS}$) at 146 hPa (c-d), at 01:30 LT (left) and at 13:30 LT (right) over pixels where 01:30 LT and 13:30 LT are during the growing phase of the convection, respectively, averaged over the period of DJF 2004-2017.

L210: more important –> larger OK
L232: lightnings are created into –> lightning is created in OK
L233: lightnings –> lightning OK

L244: the pervasive lack of superscripts in units (e.g., "month -1", "mg m-3", "m s-1","mm h-1") is puzzling, given that superscripts are used for other purposes, but it is only a trivial annoyance in most places in the manuscript. In the case of Flash, however, it is a bigger issue, since it is hard to read "10-2- 10-3" in this line. Sometimes the units on Flash are given as per day and sometimes as day-1. Also, I don't think it is true that Flash values are lower than 10-2 per day over New Guinea, at least not in the interior of the island.

We changed the superscripts in units.

P12, Fig. 5: It would be more convenient if the y-axis for Prec had 4 (not 3) minor tickmarks, as is the case for the IWC y-axis. The solid and dashed lines should also be described in the caption. OK

L246: Fig. 2c –> Fig. 2d OK
L252 (also L264, 269, 314): as noted above, the panels in Fig. 7 are mislabeled Corrected, see below.

P14, Figure 7: It would be very helpful to have more minor tick marks on the x-axis. In the caption: full line –> solid line; dash –> dashed OK

L289: delete "areas" OK
L289-290: add commas after "(2008)" and "1 mm h$^{-1}$" OK
L294: add commas after "6h" and "Flash" OK
L299: it might be good to remind readers that the elevation is shown in Fig. 2b OK
L302: than Java –> as Java  OK
L304: maintaining –> maintain  OK
L305: rainfalls –> rainfall  OK
L306: convections –> convection  OK
P16-17, Figs. 8 and 9: again, it would be very helpful to have more minor tick marks on the x-axis. Also for Fig. 9, the label for the North Australia Sea is given in panel (b) and the figure caption as "NAusSea", but in the main text it is "NAuSea". The labels should be consistent. OK

L312-313: it would be clearer to say "either coastline or offshore areas depending onthe area" OK

L320: most of –> most OK

L332: In the next section –> In Section 7 OK

P19, Fig. 10: It would aid the comparisons with Fig. 2e discussed in the text if the same color bar were used, particularly since the ERA5 IWC values reach higher values than those of MLS, yet the color bar in Fig. 2e extends to larger values. This might also alleviate the issue that the highest values of ERA5 IWC over New Guinea and North Australia appear to saturate the color bar in Fig. 10 (that is, white colors appear in the map in those regions).

Fig. 10 is presented at 0.25°x0.25° while Fig. 2 is presented at 2°x2°. According to us, it is not needed to use the same color bar, because the horizontal degradation from 0.25°x0.25° to 2°x2° tends to decrease the averaged values per pixel. Thus it is not pertinent to compare the scale value between the Fig. 10 and Fig. 2. However, it is interesting to compare the areas of maxima and minima over these two maps.

 Also, since panels (c) and (d) have been labeled "TL", it would be good to add "UT" to panels (a) and (b).

OK

L340 (also L342, 347, 366, 373, 377, 391): island –> islands  OK
L344: is –> are OK
L353: cycles –> cycle; zone –> zonesOK
L363: calculated –> calculateOK
L366: Fig. 10a –> Fig. 11aOK
L368: excepted –> except  OK
L370: from –> byOK
L372: twice greater than –> twice as large as; also delete "values"OK
L373: Fig. 10b –> Fig. 11bOK
P21, Fig. 11 caption: West Sumatra Coast (WSS) –> West Sumatra Sea (WSS)OK
L390: why are there parentheses around the convolved ERA5ΔIWC term in this line?OK
L406: to within –> by OK

L410: observational –> observations OK
L412: negligeable –> negligible OK
L414: are –> is; twice larger than –> twice as large as OK
L433: merged –> merging OK
L433 (also L516): tiny –> small (not only does "tiny" not sound very scientific, but also it could come across as dismissive) OK
L436: transport –> transports OK
L437: Fig. 10 shows IWC, not ΔIWC, so it should not be listed with Fig. 4 here; perhaps Fig. 11 was meant instead OK
L439: section 4.3 –> section 4.2 OK
L442: Awaka (1998) –> (Awaka, 1998); to –> fromOK
L446: "PR" has already been defined in L441OK
L453: Fig. 10 –> Fig. 11OK
L462: would be –> isOK
L479: from by –> from  OK
L481: to impact –> injecting OK
L482: into –> in OK
L495: amount –> amounts OK
L500 (also L504, 517): combination between –> combination of OK
L510: delete commas after "that" and "Flash" OK
L513: delete commas after "Guinea" and "Sulawesi"; range –> ranges OK
L514: as Java –> than Java OK
L516: cumulus merged –> merged cumulus OK
* * *
Anonymous Referee #2

The paper by Dion et al. is the second part of a work aiming at quantifying the diurnal cycle of ice particles in the tropical tropopause layer (TTL), and more precisely, the amount of ice injected by deep convection up the tropospheric part of the TTL, and up to the tropopause. It is mostly based on the analysis of 13 years of ice water content (IWC) data from MLS onboard the AURA satellite, as well as precipitations data from TRMM, and lightning flashes from the LIS instrument onboard TRMM. While the first part of the study, already published in ACP, is dedicated to the study of all tropical regions over the globe, this companion paper only focuses on the Maritime Continent (MariCont) during the austral convective season of December January and February, because the MariCont has been shown to be one of the most efficient tropical regions to transport ice up to the TTL in Dion et al. (2019). Here the study focuses on each sub-region of the Maritime Continent, that is the different islands and seas composingit. The main contribution from this study is to present the Maritime Continent not as awhole continent but as the sum of very different contributions. It was already shownin the first paper that the land parts had a different impact and cycle than the oceanicpart. Here the authors are going further in estimating the climatological contributionof each main islands and seas of the MariCont. For example, Java and New Guinea are shown to be the main contributors in the transport of ice up to the TTL. In thatsense, this innovative work and point of view deserve a publication. Before it can be done, I have major comments and minor comments that should be addressed

before the paper can be accepted in ACP. Some of them can be easily addressed by adding explanations and references, some others may require additional calculations.

**Major comments:**

**Instrumental part:**
Information is missing in the description of the satellite products that are used. Most of all, I would expect information on the accuracy, precision, or biases for MLS IWC, TRMM prec, and TRMM LIS (detection limit, false detection of fashes, etc). See "minor comments" for details.

We added more information for MLS IWC, TRMM prec and TRMM Flash.
Firstly, we added to the 2.1 MLS IWC paragraph, L94:

> … In our study, high spatial resolution study is now possible because we consider 13 years of MLS datasets, allowing to average the IWC$^{MLS}$ measurements within the bins of horizontal resolution of 2°×2°(~230 km). We select IWC$^{MLS}$ during all austral convective seasons DJF between 2004 and 2017. The IWC measurements were filtered following the recommendation of the MLS team described in Liversey et al. (2018). The resolutions of IWC$^{MLS}$ (horizontal along the path, horizontal perpendicular to the path, vertical) measured at 146 and 100 hPa are 300x7x4 km and 250x7x5 km, respectively. The precision of the measurement is 0.10 mg m$^{-3}$ at 146 hPa and 0.25 to 0.35 mg m$^{-3}$ at 100 hPa. The accuracy is 100 % for values less than 10 mg m$^{-3}$ at both levels and the valid range is 0.02-50.0 mg.m$^{-3}$ at 146 hPa and 0.1-50.0 mg m$^{-3}$ at 100 hPa (Wu et al., 2008).

We added the following sentence into the paragraph 2.3 TRMM-LIS, previously modified P.4-5 of the present document:

> LIS is thus able to provide the number of flashes (Flash) measured. The TRMM LIS detection efficiency ranges from 69% near noon to 88% at night.

More details about the accuracy, precision, or biases of each instrument has been described P4-5 and 6 of the present document.

The use of ERA 5 to estimate ΔIWC. I have a significant problem with this part. ERA5 is a relatively new reanalysis from ECMWF. The Authors are using ERA5 ice products to compare ΔIWC from satellite observations and from ERA5. But here, no reference is made on any estimation of the quality of ice from ERA5, nor how the ice is provided or calculated in ERA5. Is ice assimilated in the ECMWF model? If yes, from which instrument?

IWC is not assimilated but modeled in ECMWF. See explanations below (P30) providing more information and references about the use of ERA5.

If not, how it is calculated, is there a correlation between the ice product and any reported bias in ECMWF model ? As a consequence, is ΔIWC$^{ERA5}$ used to validate ΔIWC$^{Prec}$, and ΔIWC$^{Flash}$, or should it be understood the other way round? To make a meaningful comparison between both types of estimation, all of this question should be addressed in

the manuscript.

We do not use $\Delta IWC^{ERA5}$ in order to 'validate' $\Delta IWC^{Prec}$, and $\Delta IWC^{Flash}$, but in order to assess the amounts estimated by our model.

We added more detail on ERA5 in the paragraph 2.4 ERA5, L127:

> … The present study uses the $IWC^{ERA5}$ at 100 and 150 hPa averaged over DJF from 2005 to 2016 with one-hour temporal resolution. $IWC^{ERA5}$ is governed by the model microphysics which allows ice supersaturation with respect to ice (100-150% RH) but not with respect to liquid water. Although microwave radiances at 183 GHz (sensitive to atmospheric scattering induced by ice particles) (Geer et al., 2017) are assimilated, cloud and precipitations are used as control variable in the 4D-Var assimilation system and cannot be adjusted independently in the analysis (Geer et al., 2017). The microwave data have sensitivity to the frozen phase hydrometeors but mainly to larger particles, such as those in the cores of deep convection (Geer et al., 2017), but the sensitivity to cirrus clouds in ERA5 is strongly dependent on microphysical assumptions on the shape and size of the cirrus particles. Indirect feedbacks are also acting on cirrus representation in the model – e.g. changing the intensity of the convection will change the amount of outflow cirrus generated. This is why observations that affects the troposphere by changing for example the stability, the humidity, or the synoptic situation can affect the upper level ice cloud indirectly (Geer et al., 2017). $IWC^{ERA5}$ is used to assess the amount of ice injected in the UT and the TL as estimated by the model developed in Dion et al., (2019) and in the present study.

Geer, A. J., Baordo, F., Bormann, N., Chambon, P., English, S. J., Kazumori, M., ... & Lupu, C. (2017). The growing impact of satellite observations sensitive to humidity, cloud and precipitation. *Quarterly Journal of the Royal Meteorological Society*, *143*(709), 3189-3206.

Lopez, P. (2011). Direct 4D-Var assimilation of NCEP stage IV radar and gauge precipitation data at ECMWF. *Monthly Weather Review*, *139*(7), 2098-2116.

A sentence, described P.15 of the present document, has also been added L335 to remind the motivation to use IWC from ERA5.

Furthermore, to my knowledge, the ice in ERA5 is composed of 2 different variables. Are you using the total ice, or only one parameter (which should be non-precipitating ice)? The authors should be more explicit on this point and justify the choice of the variable they have used.

This point has been already developped in the replies to the reviewer « 1 's comments L121-132. We have changed sentences in the section 2.4.

More details about the accuracy, precision, or biases of each instrument has been described P4-5 and 6 in the replies to the reviewer #1.

Winds from NCEP. In Section 8, winds from NCEP are used instead of winds of ERA5. It is stated that ERA5 winds are not available. For sure they are. What would be the results shown in Fig 12 if ERA5 winds were used? Does it impact the duration of transport of ice from North Australia land to seas westward?

We have choosen to delete the Fig. 12 as well as the discussion about this figure in section 8. Thus, the paragraph describing NCEP has also been deleted. See more justification P.6 and 20-21 in the replies to the reviewer # 1.

**Minor comments:**

**- Abstract:** one of the key highlights of the study is to present the MariCont as a jigsaw puzzle of different contributions with respect to the effect of deep convection on ice in the TTL. For example, Java and New Guinea are presented as very efficient locations for the injection of ice into the TTL. Thus to me, some key findings on the effect of subregions of the MariCont should appear in the abstract.

We added some information on the abstract as follow L22-23:

Finally, from $IWC^{ERA5}$, Prec and Flash, this study highlights 1) ΔIWC over land has been found larger than ΔIWC over sea with a limit at 4.0 mg m$^{-3}$ in the UT between minimum of ΔIWC estimated over land and maximum of ΔIWC estimated over sea, and 2) small islands with high topography present the strongest amounts of ΔIWC such as the Java Island, the area of the largest ΔIWC in the UT (7.9 – 8.7 mg m$^{-3}$ daily mean).

**- L11** Lightning is always singular. See also p4 and p11. OK

**- Introduction L31.** Jensen et al. (2007) are providing important inputs on the effect of deep convection on the hydration or dehydration in the TTL. It seems appropriate to cite this study here.

We added Jensen et al., 2007 L.31 and in the reference list.

Jensen, E. J., Ackerman, A. S., & Smith, J. A. (2007). Can overshooting convection dehydrate the tropical tropopause layer?. *Journal of Geophysical Research: Atmospheres*, *112*(D11).

**- Section 2.1:** though the reference for the MLS ice product is given, no information is given on the accuracy and the uncertainties on the IWC. Please add it.

We added information in the Section 2.1 MLS IWC. See the replies to the reviewer #1 (P.26).

- Section 2.1 L90. This sentence is not clear to me. Please rephrase. At least it should be explained why you need the averaging Kernel at 100 hPa and 146 hPa.

We decided to delete the sentence. Since IWC from MLS is retrieved using optimal estimate theory, averaging kernels do not exist for IWC. We changed the previous sentence L90 by the following one:

 Although optimal estimation is used to retrieve almost all other MLS products, a cloud-induced radiance technique is used to estimate the MLS IWC (Wu et al., 2008; Wu et al., 2009).

See replies to the reviewer #1 to have more information about the ERA5 vertical degradation as a function the IWC MLS box function.

- **Section 4.1** L170: there is a very strong contrast in the maximum time of Prec between land and coastal region. If convection and Prec maxima are due to a sea breeze effect and orography over land, as stated before, why coastal region maximum of Prec is clearly not influenced by the sea breeze (otherwise the time of maximum of Prec should not be so different)? On the other hand, the coastal behavior seams relatively independent from the oceanic behavior since the oceanic behavior is very dependent on the sea considered, whereas the Prec maximum for coastal region is relatively well identified. A longer comment or hypothesis should be presented here to explain this behaviour.

Sea breeze impacts the land convection at the end of the day, when land temperature surface is higher than oceanic temperature surface: maximum of Prec is observed at the end of the day (15-24h). Over coasts, because the sea-breeze transports air masses from the sea to the land at the end of the day, the conditions are not favorable for the development of the convection. This is only once the sea breeze is stopped/reduced that the convection can be strong over coast: time of the observed maximum of Prec is during night-morning (0-6h) over coasts. Then, during the night-morning, the sea surface temperature become larger than the land surface temperature (water releases its heat much slower than land causing the air over the water to be warmer than the air over the land) and the land-breeze favours the convection development over coasts and sea (time of the maximum of Prec over sea is observed mainly in the morning-noon: 09-12h and 15-24h depending of the sea considered).

We have inserted the following sentence L.200:

Areas where the daily mean of Prec is maximum are usually surrounding the highest elevation over lands (e.g. over the New Guinea) and near coastal areas (North West of Borneo in the China Sea and South of Sumatra in the Java Sea) (Fig. 2b and c). ~~Qian (2008) explained that high precipitation is mainly concentrated over lands in the MariCont because of the strong sea-breeze convergence, combinedwith the mountain–valley winds and cumulus merging processes. The diurnal maximum of Prec over land is observed between 18:00 LT and 00:00 LT (Fig. 2d) whereas, over coastal parts, it is in the early morning before 05:00 LT. Over seas, the time of the diurnal maximum varies as a function of the region. Java Sea and North of Australia Sea present maxima around 13:30LT while the west Sumatra Sea and the Bismark Sea show maxima around 01:30 LT.~~ The times of the maxima of Prec are over land, during

the evening (18:00-00:00 LT), over coast during the night-morning (00:00-06:00 TL) and over sea during the morning-noon and even evening depending of the sea considered (09:00-12:00 LT and 15:00-00:00 LT). These differences could illustrate the impact of the land/sea breeze within the 24 hours. The sea breeze during the day favours the land convection at the end of the day when land temperature surface is higher than oceanic temperature surface. During the night, the coastline sea surface temperature becomes larger than the land surface temperature and the land-breeze favours systematically the convection development over coast. These observations are consistent with results presents in Qian (2008), explaining that high precipitation is mainly concentrated over land in the MariCont because of the strong sea-breeze convergence, but also because of the combination with the mountain–valley winds and cumulus merging processes. Amplitudes of the diurnal cycles of Prec over the MariCont will be detailed as a function of islands and sea in section 5.

- **Section 4.2** about Fig. 3. From what I understand, the number of occurrences (=cases per pixel that are during the growing phase at 13:30 or 01:30) on which the average/the anomaly is calculated depends on the pixel. What is the amplitude of the number of occurrences to get this figure?

We selected at least 60 measurements of Prec or IWC at 13:30 LT or 01:30 LT per pixel of 2°x2° over the period 2004-2017. We already provided more information about this question into the replies to the rewiever #1.

- **Section 4** Fig. 4. This is one of the key finding of the publication. However, I am surprised that qualitatively, the same patterns are found at 146 hPa and at 100 hPa, the only difference being the scale. I would expect a slightly different behavior at 100hPa because the ice amount might be also driven by other processes than just deep convection (e.g. in situ formation of cirrus or ice particles close to the tropopause). Does the result mean that other processes than deep convection are negligible, or cannot be detected by this method or the instrument? A discussion should be given at the end of the comment of Fig. 4.

ΔIWC is the amount of ice injected by deep convection (during the growing phase of the convection). Thus other processes are not considered into the calculation of ΔIWC. Dion et al. (2019) have suggested that the main process controlling ΔIWC is the deep convective process. Authors also suggest that other processes such as the in situ formation of ice, the sublimation, precipitation, horizontal advection, … are minor during the growing phase of the deep convection and could become major after the growing phase of the deep convection (namely during the decreasing phase). However, ot is out of the scope of the present paper to estimate the impact of each process on the diurnal cycle of IWC during the decreasing phase. Thus, authors have suggested that the growing phase of the diurnal cycle of Prec, representative of the growing phase of the deep convection, is correlated with the growing phase of ice into the UT and the TL. Thus, the only difference between ΔIWC into the UT and the TL is the amount of ice measured by MLS at 01:30 or 13:30 LT. The following discussion has been added L211:

… altitude is larger over land (by a factor 6) than over sea (by a factor 3). We can note that the similar pattern between the two layers come from the diurnal

cycle of Prec in the calculation of ΔIWC at 146 and 100 hPa. Only the measured value of IWC^MLS at 146 and 100 hPa can explain the observed differences in ΔIWC values at these two levels. Thus, similar ΔIWC patterns are expected between the two levels because, according to the model developed in Dion et al. (2019),  the deep convection is the main process transporting ice into the UT and the TL during the growing phase of the convection. Convective processes associated to  land and sea are further discussed in Sect. 6.

**- Section 4.3 :** 228 "this shows that..." Ok but what can we learn about the diurnal cycle or the intensity of deep convection from this correlation?

We deleted the following sentence :

>

**- Section 5** l231: potential energy –> electric potential energy OK

**- Section 5.2,** l254. The choice of 5 pixels over the sea from the land limits: why this choice? Was a sentivity test made on the number of pixel to infer the behavior of coastal regions?

We did a sensitivity test in order to select the exact number of pixels  that were the most representative of the coastal areas. We observed that considering less than 5 pixels is too low and decrease the signal-to-noise ratio, while considering more than 5 pixels presents no differences with the offshore sea signal. The following sentence has been changed:

> MariCont-C is the average of all coastlines defined as 5 pixels extending into the sea from the land limit. This choice of 5 pixels has been taken applying some sentivity tests in order to have the best compromise between a high signal-to-noise ratio and a good representation of the coastal region.

- L255: 10 pixels offshore for oceanic behavior. Please justify.

We justified it in the following sentence :

> The MariCont_O is the average of all offshore pixels defined as sea pixels excluding 10 pixels (~2000 km off the land) over the sea from the land coasts, thus coastline pixels are excluded as well as all the coastal influences. MariCont_L is the area of all land pixels.

- P14 fig 7 and results from it. There is a mismatch between the titles of the middle and bottom panels and the corresponding figure captions.

We already answered to this mismatch into the replies to the reviewer #1.

Middle panel is entitled Mari-Con_C and It is captioned MariCont_O. The other way round

for the bottom panel. In a general manner, the results for the coasts are relatively close to the one offshore. At least the time shift is weaker between MariCont_O Vs MariCont_C than for MariCont_C Vs. MariCont_L. Is the choice of 5 pixels from the land to define the MariCon_C has something to do with it? What if you had chosen 3 or 2 pixels only? Would the coastal diurnal cycle of Prec and FLash be closer to the Land cycle?

Diurnal cycles of Prec or Flash considering 2 or 3 pixels presented a too low signal-to-noise ratio to be interpreted. A number of pixels greater than 5 in the definition of the coastal region produced diurnal cycles of Prec or Flash to be the same than that over the offshore region.

- P15: References Liu and Zipser (2008) and (2009) appear in the text but not in the reference list.

P3, L71 has been changed has follow :

 ...an alternative proxy for deep convection as shown by Liu and Zipser (20098)

P15 : Liu and Zipser (2008) is now in the reference list.

- Fig 8 and 9. The ERA5 IWC is also presented in the figures and is not commented in section 5. This does not make any problem since it is commented later on in section 6 but at least, a sentence should inform that the ERA5 results would be commented later.

A sentence L.291 has been added as follow:

 Diurnal cycles of Prec and Flash are presented over land for a) Java, b) Borneo, c) New Guinea, d) Sulawesi and e) Sumatra as shown in Figure 8 and over sea for the a) Java Sea, b) North Australia Sea (NAuSea), c) Bismark Sea, d) West Sumatra Sea (WSumSea) and e) China Sea as shown in Figure 9. Diurnal cycles of IWC from ERA5 ($IWC^{ERA5}$) are also presented in Fig. 8 and 9 and will be discussed in Section 6.

About the same figs in **section 6**, L346: it would be interesting to over plot an equivalent value of the MLS IWC and comment it. As written in my major comment, there no real estimation of the ice product in ERA5. Adding the MLS IWC here could give an idea of a potential bias in the reanalysis.

Thank you, it is good idea. However, we have chosen not to add the diurnal cycle of IWC estimated from $IWC^{MLS}$ and Prec, nor the one estimated from $IWC^{MLS}$ and Flash because the figure becomes unreadable due to many curves on each figure. Furthermore, the model developed on Dion et al. (2019) although producing a full diurnal cycle of IWC, is only valid during the increasing phase. Thus, the Fig. 11  is relevant to show the differences between the estimated IWC and the one of $IWC^{ERA5}$ during the increasing phase of the diurnal cycle. Finally, by definition of eq (1), the shape of the diurnal cycle of the estimated IWC from $IWC^{MLS}$ and Prec and from $IWC^{MLS}$ and Flash would be exactly the same as the shape of the diurnal cycle of Prec and Flash, respectively.

In **section 6** the authors comment on the consistency or the inconsistency of the ERA5 IWC diurnal cycle with the Prec one. But no reason or hypothesis are given to explain this disagreement. I wish a discussion appeared at the end of section 6 on that point.

We are not able to explain or hypothesize the differences observed over sea. However we can clarify the sentence L.359 as follow (the deleted sentence was deleted and changed following a comment by Michelle Santee explained on P13-14 of the present document):

> Over Bismark Sea, the diurnal maxima of IWC$^{ERA5}$ is found at 04:00 LT with a second peak later at noon. Over West Sumatra Sea, two diurnal maxima are found at 08:00 LT and 17:00 LT. Over China Sea, the diurnal maximum of IWC$^{ERA5}$ is found at 16:00 LT and a second peak is found at 08:00 LT.  These differences in the timing of the maximum of the diurnal cycle of Prec, Flash and IWC$^{ERA5}$ observed at small-scale over sea of the MariCont are not well understood. However, these differences do not impact on the calculation of the ΔIWC$^{Prec}$, ΔIWC$^{Flash}$ or ΔIWC$^{ERA5}$. Results are presented Section 7.

**- Section 7.** Before reading the whole paper, I did not understand why results from ΔIWC offshore could be given for flash since it was shown previously that IWC$^{Flash}$ are not synchronous over seas. Thus, one could deduce that the method to estimate IWC (and ΔIWC) cannot be applied offshore. We understand later that some regions are better described by the IWC$^{Flash}$ approach than by the IWC$^{Prec}$, due for example, to the higher contribution of stratiform Prec. So in section 7.1, the part where Fig11 is presented and commented, it should be justified more clearly why ΔIWC$^{Prec}$ and ΔIWC$^{Flash}$ can be presented as a range of observational ΔIWC.

We clarified the use of the two proxies L363:

> Figure 11 synthesizes ΔIWC in the UT and the TL over the 5 islands and 5 seas of the MariCont studied in the previous section. Eqs. (1-3) are used to calculate ΔIWC from Prec (ΔIWC$^{Prec}$) and from Flash (ΔIWC$^{Flash}$). As presented in the previous section, Prec and Flash can be used as two proxies of deep convection, with differences more or less accentuated in their diurnal cycles as a function of the region considered. Thus, the observational ΔIWC range calculated between ΔIWC$^{Prec}$ and ΔIWC$^{Flash}$ provides an upper and lower bound of ΔIWC calculated from observational datasets.

In **section 7.2** p20 and 22. It would be interesting to recall here the number of model levels from 150 to 100 hPa to have an idea of the vertical resolution of the undegraded ERA5 data.

The previous paragraph has been changed as follow according to the review of Michelle Santee. We added the last sentence in clear blue in order to recall the number of model level used in the calculation:

> In order to compare $IWC^{MLS}$ and $IWC^{ERA5}$, we firstly degraded the horizontal resolution of ERA5 from 0.25°x0.25° to 2°x2° (~200 kmx200 km) and secondly, we degraded the vertical distribution of $IWC^{ERA5}$ ($IWC^{ERA5}(z_0)$) following a unitary box function whose width is 5 and 4 km at 100 and 146 hPa, respectively.

Thus, the available $IWC^{ERA5}$ at 175 hPa and the one at 70 hPa have been used in the calculation of the convolved $IWC^{ERA5}$.

- **7.3** Obviously, a comparison section is needed here, but without describing how ERA5 ice is produced/calculated, the results presented here are meaningless. I do not say it is out of interest to do so. There is probably something to learn in this comparison (for example from the fact that over seas, ERA5 ΔIWC is systematically lower than the observational ΔIWC), but here one must be aware the meaning of the product used.

We detailed how $IWC^{ERA5}$ is calculated in section 2.4 ERA5 Ice. At present, no more interpretation can be done regarding the differences highlighted.

- L418. Considering the large range of ERA5 to <ERA5>, the numbers given may not be representative.

The sentence L417-418 has been deleted:

>

- **Section 8.3** L459. I would have added Corti et al. (2008) for the references concerning Hector.

This part of the paragraph has been deleted. See answer P.20-21 of the present document.

- Section 8.3 from L465. See my major comment about the NCEP winds.

This part of the paragraph has been deleted. See answer P.20-21 of the present document.

- L471 and 472. Ice in the UT and at the tropopause is not a passive tracer. So, to state that, an estimation of the lifetime of ice particles for both altitudes should be given.

This part of the paragraph has been delated. See answer P.20-21 of the present document.

- L503 "consistent to within 75 % over seas". This corresponds to a relatively fair agreement, and the use of "consistent" seams exaggerated.

The percentage has been corrected as follow :

... consistent to within 30-50 % over seas in the UT ...

- L517. "ΔIWC is a combination..." Again, you can write this statement only if you show that the lifetime of ice particle is long enough for such a transport.

This sentence L517 has been deleted.

**Suggested references:**

- Jensen, E.J., A.S. Ackerman, and J.A. Smith, 2007: Can overshooting convection dehydrate the tropical tropopause layer? J. Geophys. Res.,112, D11209, doi:10.1029/2006JD007943.

- Corti, T., et al. ( 2008), Unprecedentedevidence for deep convection hydrating the tropical stratosphere, Geophys. Res. Lett.,35, L10810, doi:10.1029/2008GL033641.

---

## Referee Report (RR1)

**Re-review of "Ice injected into the tropopause by deep convection – Part 2: Over the Maritime Continent" by Dion et al.**

The authors have made extensive revisions in response to the comments of the two referees. In some cases, reviewer comments have been fully addressed and the manuscript has been improved. Unfortunately, however, there are many instances in which that is not the case. As was the original, the revised manuscript is again very sloppily prepared, with numerous errors – not only minor typographical mistakes, but also more serious misstatements and inaccuracies that the authors (including co-authors) should have caught themselves had they carefully proofread their work before submission. Moreover, in at least half a dozen places the authors did not actually enact the changes in the revised text that they assert they have made in their response letter, so several issues pointed out by the referees in the previous draft have not been remedied in the revised manuscript. In addition, new errors have been introduced through the revision process. As a result, the manuscript remains confusing and hard to read and still requires considerable editing and correction before it can be considered ready for publication. The list of detailed comments is therefore longer than is typically the case for a revised manuscript. Finally, although I recommend some minor typographical corrections below, I would also like to reiterate that the final manuscript will need extensive copy-editing to improve the English.

**Specific comments and questions (both major and minor comments on each Section are listed together here, rather than separately as in the review of the original manuscript):**

Abstract:
(1) L8: delete the comma after "Version 4.2)"; measurement from --> measured by.
(2) L12: resolutions --> resolution.
(3) L13: ($\Delta IWC^{ERA5}$) degrading the vertical resolution to that of MLS --> ($\Delta IWC^{ERA5}$), with the vertical resolution degraded to that of MLS.
(4) L14: lands and --> land but.
(5) L17-18: Where do the values (4–29%, 55–78%) quoted in these lines come from? They appear nowhere else in the manuscript. In particular, they do not match any of the numbers given in the relevant discussion in Section 7.2 or 7.3.
(6) L19-21: See point #7 in the comments on Section 7.2 about the phrase "xx% of variability per study zone", which is used four times in these lines.
(7) L21: See point #4 in the comments on Section 7.3 about these values (0.6 and 3.9 mg m$^{-3}$).
(8) L22: found larger than --> found to be larger than.
(9) L22-25: It is stated that $\Delta IWC$ over land is larger than $\Delta IWC$ over sea "with a limit at 4.0 mg m$^{-3}$ in the UT". What does "limit" mean in this context? Where does the value of 4.0 mg m$^{-3}$ come from? In Section 7.1, the minimum $\Delta IWC$ over land is given as 4.9 mg m$^{-3}$ and the maximum over sea as 4.4 mg m$^{-3}$, leading to a maximum difference of only 0.5 mg m$^{-3}$.
(10) L24: strongest amounts --> largest amounts.
(11) L25: It is stated that Java sees the largest $\Delta IWC$ in the UT ("7.7 – 9.5 mg m$^{-3}$ daily mean"). Again, these numbers are not drawn from the main text. Indeed, in Section 7.1, it is stated that "In the UT … over Java, $\Delta IWC$ reaches 7.9–8.7 mg m$^{-3}$".

Introduction:

(1) L54: center in the tropics with --> centers in the tropics, with.

(2) L57: though authors have done a comprehensive work around the study of --> though those authors have done a comprehensive study of.

(3) L67-68: To avoid unnecessary repetition: improve the methodology of Dion et al. (2019) --> improve their methodology.

(4) L69: assessing --> comparing.

(5) L72: Here and in 11 other lines throughout the manuscript, "Bismark" is spelled incorrectly (my apologies for not noticing this error in my original review).  Only in L510 in Section 8 is it spelled correctly: "Bismarck".

(6) L78: recalled --> reviewed.

Section 2.1:  The MLS description has been improved but still needs work:

(1) The sentence in L96-98 ("Although optimal estimation is used to retrieve almost all other MLS products, a cloud-induced radiance technique is used to validate the MLS IWC (Wu et al., 2008; Wu et al., 2009).") should be moved to become the second sentence in the paragraph.

(2) In the above-mentioned sentence, the word "validate" should be replaced by "derive".

(3) Neither of the Wu et al. papers cited in that sentence have been added to the reference list.

(4) L89: Either delete "provides" or delete "are given at".

(5) L89: Add the units "hPa" after 215.

(6) L93: tropopshere --> troposphere.

(7) L94: instrument is crossing twice a day the equator at fixed time --> instrument crosses the equator twice a day at fixed times.

(8) The sentence in L95-96 ("The horizontal resolution of IWC$^{MLS}$ measurements is ~300 and 7 km along and across the track, respectively.") is fully redundant with information given below and can be deleted.

(9) L99: allowing to average the IWC$^{MLS}$ measurements within the bins --> allowing the IWC$^{MLS}$ measurements to be averaged within bins.

(10) L101-103: resolutions … are --> resolution … is

(11) L104-105: The valid ranges quoted for 146 and 100 hPa are switched: 0.02–50.0 mg m$^{-3}$ applies to 100 hPa, while 0.1–50.0 mg m$^{-3}$ applies to 147 hPa.

Section 2.2: The revised TRMM-3B42 description needs clarification:

(1) L109: Either delete "altitude" or delete "height".

(2) L111-112: First, "composing" is not the right word in L111. Second, these two sentences seem to be contradictory.  The first states that TRMM-3B42 is "a multi-satellite precipitation analysis" but then mentions only GPM.  The second states that it is "computed from the various precipitation-relevant satellite passive Microwave (PMW) sensors" and then lists several, not including GPM.  These two sentences should be combined and clarified.  Third, why is "Microwave" capitalized?  It's part of the acronym, but so is "passive".

(3) Neither Huffman et al. (2010) nor Huffman and Bolvin (2018) have been added to the reference list.

(4) L121-122: I am still confused by the description of how 1-hour precipitation data are obtained.  As I said in my previous review, I am under the impression (based on Huffman et al. (2007) and other sources) that the TRMM-3B42 product contains gridded merged precipitation estimates with a 3-hour temporal resolution.  I had thought that only by taking advantage of the precessing orbit of TRMM and the long study period (13 years) are the authors able to bin the data in 1-hour bins.  Perhaps that is what they are getting at with these two sentences, but it is not clear.  "The granule temporal coverage of TRMM-3B42 data is 3 hours, but the temporal resolution of individual measurements is 1 minute.  Thus, it is statistically possible to degrade the resolution to 1 hour." seems to be saying that the 1-minute resolution of the individual measurements is somehow preserved in the 3-hourly averaged TRMM-3B42 data.  In addition, since they are starting from data with a 3-hour granularity, it is confusing to talk about "degrading" the resolution to 1 hour.
(5) L122: TRMM-3B42 are provided --> TRMM-3B42 data are provided.

Section 2.3: The revised TRMM-LIS description needs considerable clarification:
(1) L126: relative to --> related to.
(2) L127: Christian et al. (2000) has not been added to the reference list.
(3) L127: event --> events.
(4) L129: delete "a" before "spatial".
(5) First it is stated (L130-131) that the "instrument detects lightning with storm-scale resolution of 3–6 km (3 km at nadir, 6 km at limb) over a large region (550–550 km) of the Earth's surface".  I have no idea what "over a large region (550–550 km) of the Earth's surface" means.  Then in the next sentence (L131) it is stated that "LIS horizontal resolution is provided at 0.25°×0.25°."  Are these two sentences consistent?  Finally, it is stated several lines below (L138-139) that "The observation range of the sensor is between 38N and 38S."  The latitudinal coverage and spatial resolution of the LIS data should be described together in sentences that logically flow from one to the next.
(6) RTEP was already defined in L127, so the typo in L134 should be fixed by replacing "real-tile event processor" with "RTEP".
(7) L134: to enable the system and detect --> to enable the system to detect.
(8) First it is stated (L133-134) that "the weak lightning signals that occur during the day are hard to detect because of background illumination".  Then it is stated (L134-135) that processing to remove the background signal allows the instrument to "detect weak lightning and achieve a 90% detection efficiency during the day".  The next line (L136) states that the "TRMM LIS detection efficiency ranges from 69% near noon to 88% at night", appearing to contradict the previous sentence.  Does the detection efficiency during the day really reach as high as 90%, higher than at night?
(9) L140-141: is studied at --> is binned to.

Section 2.4:
(1) L147-148: atmosphere through a large number --> atmosphere through combining a large number.
(2) L154: into the TTL --> in the TTL.
(3) L154: Duncan and Eriksson (2018) has not been added to the reference list.

(4) L160: cloud and precipitations are used as control variable --> clouds and precipitation are used as control variables.
(5) L165: affects --> affect.
(6) 169: I'm not sure what information "unitary" is meant to convey here, or why this word is needed. In any case, "an unitary" should be "a unitary".

Section 4.1:
(1) L200: Somehow the authors are under the impression that I suggested that "the New Guinea" be changed to "NewGuinea", which is not the case. I was merely asking for "the" to be deleted. Here and elsewhere in the manuscript a space should be added: "New Guinea".
(2) L203: depending of --> depending on.
(3) L204: could illustrate the impact of the land/sea breeze within the 24 hours --> may be related to the impact of the land/sea breeze over the course of 24 hours.
(4) L205: land temperature surface is higher than oceanic temperature surface --> land surface temperature is higher than oceanic surface temperature.
(5) L206-207: sea surface temperature becomes larger than the land surface temperature and the land breeze favours systematically the convection development over coast. These observations are consistent with results presented in Qian (2008), explaining that … --> sea surface temperature rises above the land surface temperature, and the land breeze systematically favours the development of convection over coasts. These observations are consistent with results presented by Qian (2008), who explained that ….

Section 4.2: This section requires editing and reorganization to improve the clarity and flow.
(1) For one thing, it would be easier to follow if the Prec panels were described in detail first and then the similar panels for IWC mentioned; thus, the sentence in L219-221 should be moved. In addition, the description of the calculation of the anomalies in the response letter is actually clearer than that in the revised manuscript. In particular, I find the sentence in L221-222 awkward and difficult to understand. The wording in other places is clumsy as well. Therefore, I suggest rewriting much of this paragraph, such that L217-225 are replaced by: "… each pixel. Figures 3a and b present the anomaly (deviation from the mean) of Prec measured by TRMM-3B42 over the MariCont for the pixels where convection is in the growing phase at 01:30 LT and 13:30 LT, respectively. Anomalies are calculated relative to the average computed over the entire MariCont region. Thus, red colors signify regions that are experiencing the growing phase of convection and whose Prec value is greater than the overall MariCont mean at the respective time (01:30 LT or 13:30 LT), whereas blue colors signify those regions where there is little precipitation compared to the overall MariCont mean during the growing phase of convection. The gray color denotes pixels for which convection is not ongoing. Pixels can be represented in the panels for both local times when: 1) the onset of the convection is before 01:30 LT and the end is after 13:30 LT, or 2) the onset of the convection is before 13:30 LT and the end is after 01:30 LT. Similar anomalies of IWC measured by MLS over the MariCont are shown in Figs. 3c and d, over pixels when the convection is in the growing phase at 01:30 LT and 13:30 LT, respectively. Note …"
(2) L225: whithin --> within.

(3) L227 states that the Prec anomaly "varies between −0.15 and +0.15 mm h$^{-1}$".  The corresponding range for the IWC anomalies is not given until L231-232, but intervening sentences discuss both Prec and IWC anomalies.  Therefore, the sentence in L231-232 stating the IWC range should be moved to after the Prec anomaly statement in L227.
(4) L227-228: The growing phase of the convection over land is mainly at 13:30 LT --> At 13:30 LT, the growing phase of the convection is found mainly over land (I suggest this because the parallel structure with the 01:30 LT sentence below helps the reader follow the discussion.)
(5) L229: I was confused when I first read this sentence, because "north of Australia" sounds like it is referring to offshore regions.  This sentence is specifically talking about land, so this should be "northern Australia".

Section 4.3: In some places in this section the revisions have made the writing less clear, and other points also require clarification/correction.
(1) As in the preceding point, the wording in L242 makes the discussion pertaining to land confusing.  South of Sumatra, Sulawesi, North of New Guinea and North of Australia --> Southern Sumatra, Sulawesi, northern New Guinea, and northern Australia.
(2) In L242, the amount of IWC injected over seas is stated to be < 10 mg m$^{-3}$.  In L244, the largest amount of IWC injected over seas is stated to be 7–15 mg m$^{-3}$, contradicting the first statement.
(3) L245-246: The units on the Prec anomaly of 0.2 should be mm h$^{-1}$, not mg m$^{-3}$.
(4) L245-246: It is not necessary to point to Fig. 3a or Fig. 3c here since this sentence is specifically talking about 13:30 LT.
(5) L249: pattern … come --> pattern … comes.
(6) L250-251: I suggest rewriting this sentence as "The differences in the magnitudes of the ΔIWC values at 100 and 147 hPa arise from the different amounts of IWC measured by MLS at those two levels.  That is, similar …"
(7) L252: the deep convection --> deep convection.
(8) L260: diurnal cycle of Prec with low value of Prec --> diurnal cycle of Prec, with low values of Prec.
(9) L264: South of Sumatra --> Southern Sumatra.
(10) L267: It is stated that "For pixels with large values of ΔIWC, IWC observed by MLS is between 4.5 and 5.7 mg m$^{-3}$ over North Australia Sea, South Sumatra and New Guinea".  Fig. 5b shows North Australia, not the North Australia Sea.
(11) As I pointed out in the original review, the above statement applies only to New Guinea point #1.  It is not true for New Guinea point #2, for which the IWC value is much lower.
(12) For the pixels with low ΔIWC, the IWC$^{MLS}$ ranges from 1.9 to 4.7 mg m$^{-3}$.  Thus the highest IWC for these points is larger than the bottom of the range of the high-ΔIWC points.  I note that the revised summary in this section (L269-270) mentions the length of the growing phase of deep convection and the amplitude of the diurnal cycle in Prec as factors associated with large ΔIWC values, but not the magnitude of the IWC values themselves, even though the summary immediately follows discussion of the MLS IWC values. I appreciate that such a statement has been removed in revision.  I agree with that deletion and I am NOT suggesting that it be put back in.  However, it might strike readers as odd to spend several sentences (L266-269) talking about the MLS IWC values and then end that paragraph with a summary that completely

ignores them. The authors should consider adding a sentence or two noting that the IWC$^{MLS}$ ranges overlap for the high and low ΔIWC pixels, and thus no definitive conclusion about the relationship between IWC and ΔIWC can be drawn.

(13) Furthermore, language linking high-ΔIWC points to high (> 4.5 mg m$^{-3}$) IWC values has not been removed from the Conclusions section (L539-540).

(14) In my original review, I suggested that it would be easier to read values off of Fig. 5 if the y-axis for Prec had 4 (not 3) minor tickmarks, as the IWC axis does. I still feel that way.

Section 5: L273: into --> in.

Section 5.1:

(1) L281: I think "similar" would be better than "the same".

(2) L282: North Australia lands --> northern Australia.

(3) L284: It would be better to use either "~" or ">", not both.

(4) L285: It is stated that "over NewGuinea where the number of Flash is relatively low (~10$^{-2}$ – 10$^{-3}$ flashes day$^{-1}$)", but I think there are several inland areas of New Guinea where Flash exceeds 10$^{-2}$ flashes day$^{-1}$.

Section 5.2: Again in this section the revisions have made the writing less clear in places, and other points also require clarification/correction.

(1) L296: choice of 5 pixels has been taken applying some sensitivity tests --> choice of 5 pixels was made after consideration of some sensitivity tests.

(2) L298: offshore pixels defined as sea pixels excluding 10 pixels ( 2000 km off the land) over the sea from the land coasts, thus coastline --> offshore pixels defined as sea pixels excluding 10 pixels (2000 km) over the sea from the land, thus coastline.

(3) L300-302: These sentences ("In that case, we can easily discriminate between land and coastlines or sea and coastlines by applying the land/ocean/coastlines filters. Consequently, this particular pixel will be flagged both as land and coastlines or sea and coastlines") are awkwardly worded, hard to follow, and appear to contradict previous statements. I understand that pixels can straddle land/coastline and sea/coastline boundaries. I believe that this description is saying that a pixel containing both land and coastline information will be "bookkept" in both the MariCont_L and MariCont_C categories. However, as noted above, in L298 offshore pixels are defined in such a manner as to exclude the 10 pixels nearest land, so I do not see how in this context there could be any sea/coastline confusion. The areas within 5 pixels from land are put in the MariCont_C category, and those more than 10 pixels away are put in the MariCont_O category; presumably the areas stretching between 5 and 10 pixels from land are omitted from this part of the analysis.

(4) The Flash lines in Fig. 7 are described in the caption as being "dashed", but in actuality they are dotted. This causes confusion when the reader gets to Figs. 8 and 9 and expects the similar Flash curves to also be dotted. However, in those plots Flash lines are dashed and IWC$^{ERA5}$ lines are dotted. For consistency, it would be better to depict Flash data in Fig. 7 with dashed lines.

(5) As I mentioned in my previous review, it would be very helpful to have more minor tickmarks on both the x- and y-axes in Fig. 7 (as well as Figs. 8 and 9).

(6) L307: longer times --> longer.

(7) L310: Fig. 7c --> Fig. 7b.
(8) It is stated (L310-311) that "Prec minimum is around 18:00 LT".  It is then asserted (L312-314) that "These results are consistent with Mori et al. (2004) showing … a diurnal minimum of precipitation around 11:00 LT".  To me, 18:00 LT is not consistent with 11:00 LT.
(9) L317: Fig. 7b --> Fig. 7c.
(10) L324-326: Consistently, model results from Love et al. (2011) have shown the suppression of the deep convection over offshore area in West of Sumatra from the early afternoon due to downwelling wavefront highlighted by deep warm anomalies around noon. According to the authors … --> Consistent with that picture, model results from Love et al. (2011) have shown the suppression of deep convection over the offshore area west of Sumatra from the early afternoon due to a downwelling wavefront characterized by deep warm anomalies around noon. According to those authors ….

Section 5.3:
(1) L344: 6-h consistent with that of Flash and elsewhere, the duration --> 6-h, consistent with that of Flash, whereas elsewhere the duration.
(2) L346: Prec (6 h), that is --> Prec (6 h), which is.
(3) L357-360: In the case of this sentence, the authors did respond to my comments on the previous draft by making a change elsewhere, but they made no alterations to the sentence in question.  The statement is made that "The diurnal maxima of Prec found separately over the 5 islands of the MariCont are much higher than the diurnal maxima of Prec found over tropical land (South America, South Africa and MariCont_L) from Dion et al. (2019)".  I am wondering whether these differences in diurnal maximum values are attributable at least in part to the much greater horizontal resolution used in the present study.  Thus I suggest that the resolution information along with some other qualifiers be added here: "The diurnal maxima of Prec found separately over the 5 islands of the MariCont (at 0.25°×0.25° resolution) are much higher than the diurnal maxima of Prec found over broad tropical land regions (South America, South Africa and MariCont_L, at 2°×2° resolution) from Dion et al. (2019)".
(4) L361: a diurnal cycle of Prec and Flash either coastline or offshore areas depending onthe area --> a diurnal cycle of Prec and Flash similar to that of either coastline or offshore areas depending on the region considered.
(5) L362: Fig. 7c --> Fig. 7b.
(6) L373: $0.1 \times 0.2 \ 10^{-3}$ --> $0.1 - 0.2 \ 10^{-3}$.
(7) L376: over island --> over islands.

Section 6:
(1) Both referees questioned whether, given that $IWC^{ERA5}$ is itself unvalidated at this point, $\Delta IWC^{ERA5}$ can really be used to validate the observationally derived values, or might this study rather in some sense serve to use the $\Delta IWC^{Prec}$ and $\Delta IWC^{Flash}$ to validate the new ERA5 values. The authors responded in their reply letter that they are not using $\Delta IWC^{ERA5}$ to validate $\Delta IWC^{Prec}$ and $\Delta IWC^{Flash}$ but to "assess the amounts estimated by our model", and they have added to the manuscript (L384-386) the sentence "The diurnal cycle of IWC over the MariCont from ERA5 will be used to calculate ΔIWC from ERA5 in order to assess the horizontal distribution and the amount of ice injected in the UT and the TL deduced from our model combining MLS ice and

TRMM Prec or MLS ice and LIS flash." To me, there is no difference between "assessing the distribution and amount of" the ΔIWC deduced from their model and "validating" it. I still feel that it would be appropriate to acknowledge that ERA5 IWC data cannot be considered "truth", their quality has not yet been fully evaluated, and the consistency or lack thereof found in the comparisons between $\Delta IWC^{ERA5}$ and both $\Delta IWC^{Prec}$ and $\Delta IWC^{Flash}$ may have implications for both their methodology and ERA5.

(2) L388: consistently --> consistent.

(3) In both L389 and L390, the maximum value reached by $IWC^{ERA5}$ in the UT is stated to be 6.4 mg m$^{-3}$. But I wonder where the authors are getting that number from. Clearly there are values larger than 6.4 mg m$^{-3}$ in Fig. 10a (as evidenced by the white patches in the middle of the reddish colors).

(4) L393: Java, that is --> Java, which is.

(5) L402: consistently with the diurnal cycle of Prec and … --> consistent with the diurnal cycle of Prec, and …

(6) maximum … are --> maximum … is.

(7) L408: it is stated that the differences in the timing of the maximum of the diurnal cycle of Prec, Flash and $IWC^{ERA5}$ "do not impact on the calculation of the ΔIWCPrec, ΔIWCFlash or ΔIWCERA5." Is the timing unimportant because only the magnitude of the diurnal cycle (max-min) matters for the ΔIWC calculation? There is also a pervasive lack of superscripting in this sentence.

(8) L409: presented Section 7 --> presented in Section 7.

Section 7.1:

(1) It is stated (L418) that over Java, ΔIWC "reaches 7.9–8.7 mg m$^{-3}$". This sentence pertains to only two values; thus, I presume that $\Delta IWC^{Prec}$=8.7 and $\Delta IWC^{Flash}$=7.9 mg m$^{-3}$. In that case, $\Delta IWC^{Flash}-\Delta IWC^{Prec}$=0.8 mg m$^{-3}$.

(2) The sentence in L419-420 is not complete: "except for Java where is larger than $\Delta IWC^{Flash}$ by 0.7 mg m$^{-3}$ (−8%)".

(3) Assuming that the above sentence should read "except for Java where $\Delta IWC^{Prec}$ is larger than $\Delta IWC^{Flash}$", then the difference is 0.8 mg m$^{-3}$, not 0.7 as stated.

(4) Since no sign is attached to the raw difference in L420, why is the percent difference negative? It doesn't make sense to say that "$\Delta IWC^{Prec}$ is larger than $\Delta IWC^{Flash}$ by −8%".

(5) Furthermore, the percent difference as defined here should be 0.8/7.9=10%, not 8% as stated (L420).

(6) More fundamentally, why did the authors choose to use $\Delta IWC^{Flash}$ in the denominator for the percent differences? $\Delta IWC^{Flash}$ is no more "correct" than $\Delta IWC^{Prec}$. Thus, a perhaps less arbitrary approach would have been to use the average of the two ΔIWC estimates in the denominator rather than picking one of them.

(7) L422: "$\Delta IWC^{Flash}$ is almost twice as large as than $\Delta IWC^{Prec}$ (53%)" – to me it looks like $\Delta IWC^{Flash}$ is ~4.3 mg m$^{-3}$, whereas $\Delta IWC^{Prec}$ is ~2 mg m$^{-3}$ (which does work out to ~53% by the percent difference definition used here), so that $\Delta IWC^{Flash}$ is more than twice as large as $\Delta IWC^{Prec}$, not almost twice as large. Also note: large as than --> large as.

(8) The lack of any tickmarks whatsoever on the right-hand y-axes of Fig. 11 makes it quite difficult for readers to judge any of the values quoted in the text for themselves.

(9) L428-429: island --> islands; sea --> seas.
(10) L429: contamination of stratiform --> contamination by stratiform.

Section 7.2:
(1) L433-434: This is an awkward and unclear sentence. I suggest rewriting it as: "We can use the ERA5 IWC to assess the impact of the vertical resolution of the MLS measurements on the observationally derived ΔIWC estimates."
(2) L434-435: IWC$^{MLS}$ estimation derived from MLS --> estimates of IWC derived from MLS.
(3) L435: 300×7×4 --> ~300×7×4.
(4) L436-437: … we degraded: 1) the horizontal resolution of ERA5 from 0.25°×0.25° to 2°×2° ( 200 km×200 km) and 2) ERA5 data by connecting the vertical profiles … --> … two steps were taken: 1) the horizontal resolution of ERA5 was degraded from 0.25°×0.25° to 2°×2° (200 km×200 km), and 2) the vertical resolution of ERA5 was degraded by convolving the vertical profiles ….
(5) L439: The first sentence on this line ("Consistently … respectively.") is redundant and unnecessary. It should be deleted.
(6) L439-440: There is no need for the remaining sentence ("The ice injected … ⟨IWC$^{ERA5}$⟩.") to be a separate paragraph. It should be joined to the end of the preceding paragraph.
(7) L444: The phrase "xx% of variability per study zone", used here and in eight other places in Sections 7.2 and 7.3 as well as four times in the Abstract, makes no sense to me. I believe that the authors wish to quantify the range of the relative differences between ΔIWC$^{ERA5}$ and ⟨ΔIWC$^{ERA5}$⟩ in the various study zones (that is, the smallest and the largest difference between the convolved and unconvolved ΔIWC estimates from the 5 islands for the island zone and similarly from the 5 seas for the sea zone), but "xx% of variability" means something else and is not the way to convey that information. If my understanding of what the authors are intending is correct, then perhaps something like this would work: "with relative differences between ΔIWC$^{ERA5}$ and ⟨ΔIWC$^{ERA5}$⟩ of ~19-22% over the island study zone".
(8) L445: same comment as above for "xx% of variability per study zone".
(9) L449: Again, without tickmarks on the right-hand axis it is hard to tell, but the value for New Guinea looks higher than 3.7 mg m$^{-3}$ to me, more like 3.9.
(10) L449: same comment as above for "xx% of variability per study zone".
(11) L450: It is stated that ⟨ΔIWC$^{ERA5}$⟩ is larger than ΔIWC$^{ERA5}$ "by less than 2.1 mg m$^{-3}$". To me the difference between the two estimates looks more like 2.4 mg m$^{-3}$ for Sulawesi and 2.5 mg m$^{-3}$ for Java. Thus the statement "by as much as 2.5 mg m$^{-3}$" would be more accurate.
(12) L451: same comment as above for "xx% of variability per study zone".
(13) L451-452: Again, the value of 0.2 mg m$^{-3}$ should be checked here, as it looks as though it might be larger than that for the Java and North Australia seas, and "by as much as" would be better than "by less than".

Section 7.3:
(1) L458: delete stray ")" after "respectively".
(2) L459: I do not understand what a single percentage value (75%) means in the context of the fairly large ΔIWC range (1.0 – 2.2 mg m$^{-3}$) quoted here. Other percentages that originally appeared in this paragraph have been deleted; this one probably should be too.

(3) L461: same comment as above for "xx% of variability per study zone".

(4) L462: I note that the authors' response letter indicates that the values 0.6 and 3.9 mg m$^{-3}$ in this line have been changed to 0.5 and 3.7 mg m$^{-3}$. However, such changes have not actually been made in the revised text.

(5) L462: same comment as above for "xx% of variability per study zone" (twice in this line).

(6) L463: same comment as above for "xx% of variability per study zone".

(7) L465: In statements characterizing differences as "within 1.7 to 0.7 mg m$^{-3}$", "within" would typically be taken to imply small differences, but in this case the differences are referred to as being "significant". I suggest instead "as much as 0.7 to 1.7 mg m$^{-3}$".

(8) I agree that differences of that magnitude (0.7 to 1.7 mg m$^{-3}$) are substantial. This brings up the issue that in the previous paragraph (L456-457), agreement "within 0.1 to 1.0 mg m$^{-3}$" was characterized as highlighting the robustness of the model over land. But 1.0 mg m$^{-3}$ is arguably not a small difference.

(9) L466: It is stated that significant differences are found over all individual offshore study zones "within 0.7 to 2.1 mg m$^{-3}$". But differences for the sea zone were given as "1.0 – 2.2 mg m$^{-3}$" on L459. (And again "as much as" would be better than "within".)

(10) The previous couple of points lead me to wonder why the information in the sentence in L464-466 is repeated, as these values were already given in the previous paragraph. I think that these two paragraphs could be combined and written more efficiently.

(11) L470-471: Based on the reanalysis ΔIWC range, it is suggested that Sulawesi and New Guinea may also reach high ΔIWC values comparable to those over Java, even though the observationally derived ΔIWC estimates do not indicate such strong injections over those two islands. But might it not also be the case that the reanalysis might be in error in those regions? Perhaps they pose particularly challenging environments for the models to get right. Again, it seems to me that the authors see the comparisons with ERA5 only as a means of validating (or assessing) their model, and not as a two-way street that possibly highlights regions of potential issues in the reanalysis.

Overall comment on Section 7: Given the fairly large number of places where corrections to values appear to still be needed, I am not convinced that sufficient care was taken in revising the manuscript, and I encourage the authors to double-check all of the values as well as the absolute and relative differences quoted in the text.

Section 8.2:
(1) L496: "The authors" in this line is unclear – this wording immediately follows a reference to Awaka (1998), but I assume that it is really pointing to Mori et al. (2004). Please clarify.
(2) L499: Sumatra West Sea --> West Sumatra Sea.

Section 8.3:
(1) L505-508: There is a pervasive lack of superscripting in these lines.
(2) L508: Not only is mg m-3 not superscripted here, but also in the second instance the unit is wrong: m−2 --> m$^{-3}$.
(3) L508: same comment as above for "xx% of variability".

(4) L510: as for Java Sea or Bismarck Sea, North Australia Sea has the particularity to be surrounding by several islands --> as for Java Sea or Bismarck Sea, North Australia Sea is surrounded by several islands.

(5) L514-515: composed by storms with lightning but precipitationd are weak or do not reach the surface and evaporating before --> composed of storms with lightning but precipitation is weak or does not reach the surface before evaporating.

Section 9:

(1) L523-525: It struck me that the summary in the Conclusions does not mention that a key assumption in the model is that deep convection is in the growing phase. Since many readers may only skim the paper and focus mostly on the Conclusions, and many may also be unfamiliar with the earlier Dion et al. (2019) paper, the authors should make clear that they are only applying their model during the increasing phase of the diurnal cycle of deep convection.

(2) L524: impacting --> injecting.

(3) L525: into the --> in the.

(4) L539-540: As noted in point #13 in the comments on Section 4.3, similar language to that in point (ii) here has been deleted from Section 4.3.

(5) L542: delete "within" in front of both "4–22%" and "7–53%".

(6) L544: included into --> included in.

(7) L546-547: It is stated that "the observational ΔIWC range has been shown to be consistent with the reanalysis ΔIWC range to within 23 % over land and to within 30–50 % over sea in the UT and to within 49% over land and to within 39% over sea in the TL". It is not clear where these numbers are derived from, as none of them have appeared previously in the manuscript.

(8) L547: combination between --> combination of.

(9) L549-551: I think that the wording "to within xx % per study zones" (which appears four times in these lines) will be unclear, especially to readers who skip to the conclusions without going carefully through the entire manuscript.

(10) L550: As noted in point #4 in the comments on Section 7.3, the values 0.6 and 3.9 mg m$^{-3}$ have been changed to 0.5 and 3.7 mg m$^{-3}$ in the response to referees but not the main text.

(11) L551-552: Although the differences between ΔIWC$^{ERA5}$ and ⟨ΔIWC$^{ERA5}$⟩ at the two levels suggest that the vertical resolution of the observations has a stronger impact in the TL than in the UT, the total ΔIWC variation range being discussed in this sentence does not. Shortcomings in the methodology, Prec, or Flash could all contribute to the total ΔIWC variation range.

(12) L554-556: Are the values quoted in these lines (0–0.6, 1.0, 0.3 mg m$^{-3}$) consistent with the corresponding numbers given in Section 7 (I don't think so)?

(13) L557: ice injected with ERA5 than Java in the UT and even larger ranges of values as Java in the TL --> ice injected with ERA5 as Java in the UT and even larger ranges of values than Java in the TL

(14) Concerning the sentence discussed above, see point #11 in the comments on Section 7.3.

References: The pervasive lack of proper capitalization and bizarre and unnecessary hyphenation that I commented on in my previous review persist in the revised manuscript.

---

## Referee Report (RR2)

**Review of Second Revision of "Ice injected into the tropopause by deep convection – Part 2: Over the Maritime Continent" by Dion et al.**

Once again, the authors have made extensive revisions in response to referee comments. Many of those comments have been addressed and the manuscript has been improved. In particular, the new approach to and discussion of the quantification of uncertainty in the various ΔIWC estimates is much clearer now. However, the author team's carelessness in manuscript preparation – which has plagued this paper from the start – persists in this latest draft. Not all issues noted previously have been fixed, and new errors (some of which are major) have been introduced. The most serious is a mischaracterization of the accuracy of MLS IWC data (see specific comments below). As a consequence, the authors have not successfully replied to points #1 and #2 raised by Referee #2 in their review of the previous draft.

In addition, the authors' new approach to quantifying the range of variability in ΔIWC estimates means that all of the associated values (in the Abstract and Sections 7, 8, and 9) have changed, and given how many times the quoted numbers have been misstated before, it is difficult for me to have complete faith in them now. Moreover, in some case the values quoted in the response to referees do not match those in the corresponding line in the text, and it is difficult to judge which might be correct. I did a few spot checks on the numbers in the manuscript and found some mistakes (see specific comments below). Thus, I implore the authors to carefully check all of their arithmetic again before submitting a final manuscript for publication.

Finally, I hope that it is clear that I am working hard to be of assistance in improving this manuscript. I am not deliberately being difficult. I believe that this study represents a novel and clever application of MLS data in addressing an important issue. However, the lack of care demonstrated by the authors in writing and revising this manuscript is frustrating. In my opinion, publishing a paper with as many inaccuracies as remain in this draft – the second revision – would undermine their credibility in the community, cast doubt on all aspects of the analysis they have conducted, and ultimately weaken the impact of this work.

**Specific comments (both major and minor comments are listed together for each Section):**

**Abstract:**
(1) L6-7: ice injected (ΔIWC) up to the TL by combining ice water content (IWC) measured twice a day in local time in tropical UT and TL by --> ice injected (ΔIWC) up to the tropical UT and TL by combining ice water content (IWC) measured twice a day by
(2) L8: (Prec) measurement --> (Prec) measurements
(3) L12: resolutions --> resolution
(4) L14-21: These sentences are confusing and hard to read. I recommend re-writing as:
  Our study shows that the diurnal cycles of Prec and Flash are consistent with each other in timing and phase over land but different over offshore and coastal areas of the MariCont. The observational ΔIWC range between ΔIWC$^{Prec}$ and ΔIWC$^{Flash}$, interpreted as the uncertainty of our model in estimating the amount of ice injected, is smaller over land (where the two estimates agree to within –6 to –22 %) than over ocean (where

relative differences are +6 to –71 %) in the UT and TL. The impact of the MLS vertical resolution on the estimation of ΔIWC is greater in the TL (differences between ΔIWC$^{ERA5}$ and ⟨ΔIWC$^{ERA5}$⟩ of 32 to 139%) than in the UT (difference of 9 to 33%). Considering all the methods, in the UT estimates of ΔIWC span 4.2 to 10.0 mg m$^{-3}$ over land and 0.3 to 4.4 mg m$^{-3}$ over sea, and in the TL estimates of ΔIWC span 0.5 to 3.7 mg m$^{-3}$ over land and 0.1 to 0.7 mg m$^{-3}$ over sea.

(5) In the above, are the values of 0.3 (min of the range in the UT over sea) and 3.7 (max of the range in the TL over land) correct? From Fig. 11, to me these values look more like 0.4 and 3.9, respectively.

(6) L21-23: First, the statement that ΔIWC is smaller than 4 mg m$^{-3}$ over sea directly contradicts the previous sentence, where the top of the range over the sea in the UT is correctly stated to be 4.4 mg m$^{-3}$. Second, it is not clear that these numbers apply only to the UT, not the TL. I recommend instead using the wording in Section 7.3 (L480): "At both levels, ΔIWC estimated over land is more than twice that estimated over sea."

(7) L23: present the largest ΔIWC such as the Java Island (7.7 to 9.5 mg m$^{-3}$ in the UT) --> present the largest ΔIWC (e.g., Java Island, with values of 7.7 to 9.5 mg m$^{-3}$ in the UT)

**Section 1:**
(1) L43: "twice daily in local times" – I do not think that the addition of "in local times" here is helpful, so it should either be deleted or changed to "twice daily (at 01:30 and 13:30 local time)"

(2) L52: center in the tropics with --> centers in the tropics, with

**Section 2.1:**
(1) The authors have confused accuracy (systematic error) and precision (random noise). Precision is generally improved by averaging; accuracy is not. That is, the precision of an average of N profiles is 1/sqrt(N) times the precision of an individual profile. Since their analysis involves averaging in both space and time, the precision (measurement noise) of the MLS IWC data is of essentially no consequence for this study. But, contrary to what has been written here, such averaging does nothing to mitigate the 100% systematic uncertainty (accuracy) of the IWC measurements. Referee #2 asked what the implications of the large (100%) uncertainty in the MLS IWC data are for this analysis. The authors have failed to address this point correctly in their revised manuscript.

(2) This section was heavily edited in revision, but unfortunately the changes do not represent an improvement. The overall flow is poor, and the repetitiveness and seemingly random arrangement of sentences (with multiple instances of unrelated points being interposed between sentences that should have been connected) make it hard to follow. The wording is also incorrect in places (besides the accuracy issue), and some quoted values are wrong.

(3) To address the above comment (2), I recommend re-ordering / re-writing this paragraph as:
> The Microwave Limb Sounder (MLS) was launched on NASA's Earth Observing System Aura platform in 2004 (Waters et al., 2006). MLS follows a sun-synchronous near-polar orbit, obtaining daily global coverage. Ascending (northbound) portions of the orbit cross the equator at 13:30 local time (LT); descending portions of the orbit cross the equator at 01:30 LT. Among other products, MLS provides

measurements of ice water content (IWC$^{MLS}$, mg m$^{-3}$). Although optimal estimation is used to retrieve almost all other MLS products, a cloud-induced radiance technique is used to derive IWC$^{MLS}$ (Wu et al., 2008, 2009). Here we use version 4.2 IWC data, filtered following the recommendations of the MLS team described by Livesey et al. (2018). We select IWC$^{MLS}$ during all austral convective seasons DJF between 2004 and 2017. MLS data processing provides IWC$^{MLS}$ at 6 levels in the UTLS (82, 100, 121, 146, 177 and 215 hPa). We have chosen to study only two levels: an upper and a lower level of the TTL. Because the level at 82 hPa does not provide enough significant measurements of IWC to achieve good signal-to noise, we have selected 100 hPa as the upper level of the TTL (named TL, for tropopause level) and 146 hPa as the lower level of the TTL (named UT, for upper troposphere). The resolution of IWC$^{MLS}$ (horizontal along the path, horizontal perpendicular to the path, vertical) measured at 146 and 100 hPa is 300×7×4 km and 200×7×5 km, respectively. In our study, high horizontal resolution is now possible because we consider 13 years of MLS data, allowing the IWC$^{MLS}$ measurements to be averaged in bins with 2°×2° (~230 km$^2$) horizontal resolution. Typical single-profile precisions are 0.08–0.18 mg m$^{-3}$ at 146 hPa and 0.20–0.65 mg m$^{-3}$ at 100 hPa, and the accuracy is 100% for values less than 10 mg m$^{-3}$ at both levels. The valid IWC range is 0.1–50.0 mg m$^{-3}$ at 146 hPa and 0.02–50.0 mg m$^{-3}$ at 100 hPa (Livesey et al., 2018).

(4) Note that my suggested re-writing of this section does not address the concern about the accuracy of the MLS IWC measurements raised by Reviewer #2, which I leave to the authors to answer.

**Section 2.2**

(1) The organization of this section is also awkward, with a sentence about the Prec product not differentiating between stratiform and convective precipitation coming in between two sentences about horizontal resolution and binning, then a couple sentences about averaging in time, followed by a sentence pointing back to the spatial binning methodology. As I stated in previous reviews, the authors should arrange this description in a more logical manner that steps through all related points before moving on to other aspects.

(2) L115: averaged over a 1-hour interval --> averaged over 1-hour intervals

(3) I still think it will not be clear to all readers how this 1-hr resolution for Prec is achieved. As noted in my previous reviews, the authors are able to take advantage of the precessing orbit of the TRMM satellite and the long (13-yr) study period to bin the data into 1-hr bins. They have now included a sentence to this effect in the LIS description (L132-133), and I think it would be helpful to include something along those lines here as well.

(4) L117: is provided --> are provided

**Section 2.3**

(1) L119: aboard of --> aboard

(2) L121: pixel representing --> pixel, representing

(3) L123-125: Confusing aspects of the LIS description previously mentioned have not been rectified in the revised manuscript. It is stated that: "The instrument detects lightning with storm-scale resolution of 3-6 km (3 km at nadir, 6 km at limb) over a large region (550×550

km) of the Earth's surface. The LIS horizontal resolution is provided at 0.25°×0.25°." Are these two sentences consistent with one another?

(4) L131: consistent to … we are using --> consistent with … we use
(5) L133-134: "In our study, Flash measured by LIS is binned at 0.25°×0.25° horizontal resolution to be compared to Prec from TRMM-3B42." As stated in L125, 0.25°×0.25° is the LIS native resolution. I assume that 2°×2° is meant here.

**Section 4.1**
(1) L188: associated to --> associated with
(2) L193: instead of fixing "NewGuinea", it was deleted: (e.g. over ) --> (e.g. over New Guinea)

**Section 5.2**
(1) Once again, the order of the panels in Fig. 7 is mischaracterized. This error had been fixed in the previous revision, but the figure has now been redrawn so it has reappeared in this draft. Consequently, references to Fig. 7 in L290, L307, L315, and L362 are all wrong, as is the figure caption.
(2) L297-299: I still find the wording in these sentences contradictory and confusing. "At the border between the land and the coast areas, a given 0.25°×0.25° pixel can contain information from both land and coastlines. In that case, we can easily discriminate between land and coastlines by applying the land/coastlines filters. Consequently, this particular pixel will be flagged both as land and coastlines." If in fact you could easily discriminate between land and coastlines, then you would not need to "double count" these pixels by placing them in both categories. Isn't it because they cannot be easily differentiated that they need to be flagged as being in both regimes?
(3) L304: Why does this sentence start with "Consequently"? That word does not seem appropriate to me here; perhaps "Nonetheless" might be better, or nothing.

**Section 5.3**
(1) L335: value --> values
(2) L346: of altitude --> altitude
(3) L348-349: air masses cooled in altitude are transported to the sea favoring the dissipating stage of the convection. Sulawesi is also a small island with high topography as Java --> air masses cooled at higher altitudes are transported to the sea, favoring the dissipating stage of the convection. Like Java, Sulawesi is a small island with high topography.
(4) L356: over tropical land --> over broad tropical land regions
(5) L367: instead of fixing the spelling of "Bismark Sea", it was deleted: NAusSea, Sea and WSumSea --> NAusSea, Bismarck Sea and WSumSea
(6) L373: over the Sea --> over the Bismarck Sea

**Section 6**
(1) L386-388: This wording is unclear and awkward. I suggest: "In assessing the consistency or lack thereof in the comparisons between $\Delta IWC^{ERA5}$ and both $\Delta IWC^{Prec}$ and $\Delta IWC^{Flash}$, it should be kept in mind that $IWC^{ERA5}$ data quality has not yet been fully evaluated."
(2) L390: New Guinea where --> New Guinea, where

(3) L410: impact on --> affect

**Section 7.1**
(1) Unless I missed it, nowhere in this section is it stated that the range between $\Delta IWC^{Prec}$ and $\Delta IWC^{Flash}$ is quantified as a means of characterizing the uncertainty in their model. Such a statement is made in the Abstract (L15-16), and I think it would be good to explicitly note it here (e.g., in L419, observational upper and lower bounds), as well as in the Conclusions.
(2) L424: (with $r^{Prec-Flash}$ ranges from - 6 to - 22% over the study zone) --> (with $r^{Prec-Flash}$ ranging from –6 to –22% over the study zones)
(3) L425: Of course, I did not check all of the arithmetic in this section, but I recommend that that the authors do so. According to Eqn. (4) and the values given in L423, for Java $r^{Prec-Flash}$ = 100 × [ (8.7–7.9) / 0.5*(8.7+7.9) ] = 9.6%, not 7.1% as stated here.
(4) L426: To me it looks as though $\Delta IWC^{Flash}$ is greater than $\Delta IWC^{Prec}$ by more like 2.3 mg m$^{-3}$ over the NAS, not 2.1 mg m$^{-3}$ (the max difference stated here).
(5) L428: are --> is
(6) L431: UT with --> UT, with
(7) L433-434: What is the statement "Observational $\Delta IWC$ over Java island is larger by about 1.0 mg m$^{-3}$ in the UT and about 0.3 mg m$^{-3}$ in the TL than other land study zones" based on? Do these values represent averages of the $\Delta IWC^{Prec}$ and $\Delta IWC^{Flash}$ estimates for Java vs. averages of the $\Delta IWC^{Prec}$ and $\Delta IWC^{Flash}$ estimates for all of the other islands? Or are the authors just comparing the bottom end of the estimate range for Java with the top end of the range for all of the other islands? Certainly, the estimates for Java exceed those for some of the other islands (e.g., Sumatra) by much more than 1.0 mg m$^{-3}$ in the UT. A similar question pertains to the value of 0.3 mg m$^{-3}$ in the TL.
(8) L436: largest difference --> larger difference

**Section 7.2**
(1) L446: The ice injected from ERA5 at $z_0$ --> The ERA5 amount of ice injected at $z_0$
(2) L447: we can consider --> we consider
(3) L458: larger than $\Delta IWC^{ERA5}$ by less than 2.5 mg m$^{-3}$ --> larger than $\Delta IWC^{ERA5}$ by as much as 2.5 mg m$^{-3}$ over some islands
(4) L459: To me it seems that the difference between $\Delta IWC^{ERA5}$ and $\langle \Delta IWC^{ERA5} \rangle$ might be as large as 0.3 mg m$^{-3}$ for the Java and North Australian Seas, not 0.2 as stated here.

**Section 7.3:**
(1) L465: range --> ranges
(2) L467: greater than the reanalysis by ~1.0–2.2 mg m$^{-3}$, showing a systematic larger estimate derived from observation than derived from reanalysis --> greater than that of the reanalysis by ~1.0–2.2 mg m$^{-3}$, with systematically larger estimates derived from observations than from the reanalysis
(3) L468-472: The description of the quantification of the "consistency" between the observational and reanalysis $\Delta IWC$ estimates remains confusing and poorly written. For one thing, it is presented in such a way that small values (0–25%) indicate that the two are consistent and large values (96%) indicate that they are inconsistent, which seems

counterintuitive.  In addition, the wording "the difference between x minus y" is incorrect, and several other wording issues and grammar errors make these sentences hard to understand.  I recommend re-writing L468-472 as:

> The consistency between observational and reanalysis ΔIWC ranges is calculated as the minimum value of the higher range minus the maximum value of the lower range divided by the mean of these two values.  In the UT, observational and reanalysis ΔIWC estimates are found to be consistent over land, where the relative differences between their ranges are less than 25%, but inconsistent over sea, where differences are 62–96%.  In the TL, the relative differences between the observational and reanalysis ΔIWC ranges are 0–49% over land and 0–28% over sea.

(4) L472-476: The description of $r^{Total}$ is also quite unclear and badly written.  Moreover, as originally defined, $r^{Total}$ would always be a negative number, but the values quoted for it are not negative.  I recommend re-writing as:

> In the following, we define the total range covering the observational and reanalysis ΔIWC estimates, $r^{Total}$, as the maximum value of the higher range minus the minimum value of the lower range divided by the mean of these two values. In the UT, the observational and reanalysis ΔIWC estimates span 4.2 to 10.0 mg m$^{-3}$ (with $r^{Total}$ values from 8 to 59%) over land and 0.3 to 4.4 mg m$^{-3}$ (with $r^{Total}$ values from 104 to 149%) over sea. In the TL, the observational and reanalysis ΔIWC estimates span 0.5 to 3.7 mg m$^{-3}$ (with $r^{Total}$ values from 85 to 127%) over land and 0.1 to 0.7 mg m$^{-3}$ (with $r^{Total}$ values of 142 to 160%) over sea.

(5) L476: Are the values of 0.3 mg m$^{-3}$ for the bottom of the ΔIWC range over sea in the UT and 3.7 mg m$^{-3}$ for the top of the ΔIWC range over land in the TL correct?  To me, they look more like 0.4 and 3.9 mg m$^{-3}$, respectively (as also noted in connection with the abstract).

(6) L478-479: Amounts of ice injected deduced from observations and reanalysis are consistent to each other over land in the UT and over land and sea in the TL (to within 0 to 49%) but inconsistent over sea in the UT (up to 96%) --> Amounts of ice injected deduced from observations and reanalysis are consistent (i.e., the relative differences between their respective ranges are less than 49%) over land in the UT and over land and sea in the TL but inconsistent over sea in the UT (where differences are as large as 96%)

(7) L478-479: This is backwards!  the impact of the vertical resolution on the estimation of ΔIWC is much larger in the UT than in the TL --> the impact of the vertical resolution on the estimation of ΔIWC is much larger in the TL than in the UT

(8) L481: The statement that "Java island presents the highest observational and reanalysis ΔIWC range in the UT (between 7.7 and 9.5 mg m$^{-3}$ daily mean)" is misleading – at first I interpreted it to be saying that Java shows the largest *range* of observational and reanalysis ΔIWC estimates (which, according to Fig. 11, is not true: that would be New Guinea, with values from ~5.6 to 10.0 mg m$^{-3}$).  I think the authors mean that the estimated ΔIWC *values* for Java are larger than for other islands, but that is the case only for the observational estimates, not ΔIWC$^{ERA5}$.  Also, what is meant by "daily mean" here?

(9) L482-484: The statement "assuming that ERA5 IWC data have not yet been evaluated" makes no sense in this context.  I suggest instead:

> Whatever the level considered, although Java has shown particularly high values in the observational ΔIWC range compared to other study zones, the reanalysis ΔIWC range

shows that Sulawesi and New Guinea may also reach high values of ΔIWC similar to those seen over Java.  However, as the ERA5 IWC data have yet to be extensively validated, it is also possible that the reanalysis overestimates IWC in these regions.

**Section 8.1**
(1) L494: impacts on the diurnal cycle of Prec and on the IWC --> impacts the diurnal cycle of Prec and the IWC
(2) L495: delete "and" at the end of this line
(3) L498-499: cumulus merging processes which are processes more important --> cumulus merging processes, which are more important
(4) L501: IWC is increasing proportionally with Prec consistent --> IWC increases proportionally with Prec, consistent
(5) L502-503: add commas after "(2019)" and "(Fig. 3)"

**Section 8.2**
(1) L508: precipitations --> precipitation
(2) L515-516: the calculation of ΔIWC estimated from Prec is possibly overestimated because Prec include --> ΔIWC calculated from Prec is possibly overestimated because Prec includes

**Section 8.3**
(1) L522: ~71% --> ~–71%
(2) L523: large highlighting the difficulty to estimate --> large, highlighting the difficulty of estimating

**Section 9**
(1) L537: binned at a 1-hour diurnal cycle --> binned at 1-hour resolution over the diurnal cycle
(2) L538: selected among --> during
(3) L555-556: (a) I think that "disagree" or "deviate from one another" would be better than "depart".  (b) "from" and "to" should be "by".  (c) –6% over sea should be +6%.  (d) If the sign of these relative differences is specified, then the fact that Prec is usually smaller needs to be made clear.  (e) largest --> larger.  Thus, taking these issues into account, I recommend that these lines be re-written as: "ΔIWC$^{Prec}$ is typically smaller than ΔIWC$^{Flash}$, with the two estimates disagreeing by –6 to –22% over land and +6 to –71% over sea.  The larger …"
(4) L561: inconsistent to within 96 % over sea in the UT. Thus, thanks to the combination --> inconsistent over sea in the UT, where relative differences are as large as 96%. Thus, considering the combination
(5) L563: 0.3 might be 0.4 and 3.7 might be 3.9, as mentioned earlier
(6) L564: found higher --> found to be greater
(7) L567-568: Java with … the Java Island --> Java, with … Java Island
(8) L568-569:  See comment #6 in Section 7.1.
(9) L570: than the Java Island keeping in mind --> than Java Island, although it must be kept in mind

---

## Referee Report (RR3)

**Review of Third Revision of "Ice injected into the tropopause by deep convection – Part 2: Over the Maritime Continent" by Dion et al.**

The authors have again made extensive revisions (not all of which were directly in response to referee comments), and the manuscript has been greatly improved. However, there are a few remaining points of clarification, as well as some instances of awkward or grammatically incorrect wording, that I think should be addressed before the paper is published. Some of the issues noted below have just arisen in this revision, while others were present in earlier drafts but have become more obvious now that the more serious issues have been remedied. I recommend publication after the specific points detailed below have been resolved.

**Abstract:**
(1) L1-3: Results presented in a companion paper (Part I) have used … and shown --> A companion paper (Part 1) used … and showed
(2) L9: "binned" would be better than "averaged" here.
(3) L17: First, I find the term "absolute relative differences" confusing. Second, stating absolute values for the relative differences is inconsistent with what was done in the rest of the paper and is somewhat misleading. Third, see comment #2 in Section 7 about how these ranges are specified. Please enact changes consistently throughout the manuscript.
(4) L19: It would be good to add ", depending on the study zone" after "139%".

**Section 1:**
(1) L36: With the edits made to the beginning of this paragraph, "CPT" is no longer defined.
(2) L75: an other --> another

**Section 2:**
(1) L96: of tropopause which --> of the tropopause, which
(2) L97: Lower Stratosphere … do --> lower stratosphere … does
(3) L115-117: The awkward grammar and the redundancy in these sentences make them hard to read and confusing. I suggest instead: "The TRMM-3B42 product (version V7) is a multi-satellite precipitation analysis that extends the precipitation product through 2019 by merging microwave and infrared spaceborne observations, including TRMM measurements from 1997 to 2015."
(4) L118: are provided --> is provided
(5) L124-126: In addition to some typos/grammar issues, this sentence gives the impression that the TRMM-3B42 data record is only 13 years long, which is not the case. I suggest re-writing as: "This was possible because of the combination of the precessing orbit of the TRMM satellite and the availability of precipitation estimates from the other satellites included in the TRMM-3B42 analysis during our 13-year study period."
(6) L130-134: Although it is much clearer in this draft, I remain confused by aspects of the LIS description (e.g., how the 550 × 550 km region mentioned relates to the stated 3–6 km resolution of the LIS measurements), so I went to download the Christian et al. [2000] reference (LIS ATBD). The URL given in the reference section (L619-620) does not seem to

point to an active site.  However, I was able to obtain the LIS ATBD from
https://eospso.nasa.gov/sites/default/files/atbd/atbd-lis-01.pdf.

(7) I looked through the LIS ATBD only briefly and did not thoroughly read it, but I note that many of the values given in these lines (e.g., resolution at nadir and limb, detection efficiencies at noon and at night, latitude range) do not appear in that document.  Clearly the authors have relied on another source of LIS information besides the referenced ATBD.  If there is a published paper that contains relevant information, it should be cited here.

(8) L148: I'm not sure exactly what is meant by the ERA5 "process" – data assimilation system?

(9) L154: radiances data --> radiance data (or, radiances data --> radiances)

(10) L157: Delete "by".

(11) L163: Delete "and in the present study"; also, add "profiles" or "data" or something similar between "IWC$^{ERA5}$" and "have been degraded".

(12) L164: the MLS vertical resolution of IWC$^{MLS}$ --> the vertical resolution of IWC$^{MLS}$

**Section 4:**

(1) L207: between Prec low values (4–8 mm day$^{-1}$) and IWC$^{MLS}$ large concentrations (4–7 mg m$^{-3}$) --> between low values of Prec (4–8 mm day$^{-1}$) and large values of IWC$^{MLS}$ (4–7 mg m$^{-3}$)

(2) L210: Delete the comma after "analysis".

(3) L211-213: These two sentences ("From …pixel.") are fully redundant with the newly added sentence in L209-211 and should be deleted.

(4) L216: with Prec value is --> with Prec values

(5) L217: on the contrary --> in contrast

(6) L236: section 2.4 --> section 3

(7) L251: Figure --> figure

(8) L256-257: It seems to me that the sentence about the low value of ΔIWC over the sea is out of place here.  I think it would go better at the end of the previous paragraph, since what is currently the last sentence of that paragraph also discusses pixels with low values of ΔIWC.

(9) L258: This sentence is missing essential information.  I think the authors mean "when **ΔIWC** is large" and "when **ΔIWC** is small" ("small" is a more appropriate word than "weak").

(10) L259: Since this sentence begins "More precisely", the implication is that it will elaborate further on the immediately preceding discussion.  But this is not the case – the previous sentence talks about the duration of the increasing phase of the Prec diurnal cycle, whereas this sentence is about its amplitude.  I would delete "More precisely".

(11) L262: IWC$^{MLS}$ … are --> IWC$^{MLS}$ … is

**Section 5:**

(1) L277: Flash takes --> Flash has (or, Flash takes --> Flash is characterized by)

(2) L300: The times when Flash and Prec reach their maxima are not the same as those given in the previous section (L284-285).  I realize that the earlier estimates are based on Figs. 6b and 2d, whereas the numbers here are from the broader averages of Fig. 7, but it is still a bit confusing for the reader, so some words of clarification would help.

(3) L352-353: In addition to a couple of other minor wording issues in this sentence, I don't think it is correct to characterize the decreasing phase of the diurnal cycle as "decreasing more rapidly" for Flash than for Prec.  To address all of the issues, I suggest something along

these lines: "However, because Flash is observed only in deep convective clouds, the decreasing phase of the Flash diurnal cycle is shorter than the decreasing phase of Prec."

(4) L358-372: The discussion in this paragraph is confusing. The sentence starting in L360 notes that the Java Sea shows the largest diurnal maxima in Prec and Flash. The sentence starting in L364 marks the contrast with NAusSea, Bismarck Sea, and WSumSea, which display diurnal cycles with small amplitudes. It is then stated (L366-367) that "Java Sea and WSumSea present a similar diurnal cycle of Prec and Flash, with Flash growing phase starting about 4 h earlier than that of Prec." This seems to contradict the earlier statement contrasting the diurnal cycles of Java Sea and WSumSea. Perhaps WSumSea should be omitted from the list of regions with weak diurnal cycles. Moreover, while it is true that the difference in the timing of the onset of the growing phases of Flash and Prec is about 4 h for WSumSea, this sentence implies a similar timing difference for the Java Sea, but that is not what Fig. 9a seems to indicate (although the lengthy "plateau" in the diurnal cycle of Flash between about 10 and 18 LT makes it difficult to judge exactly when its growing phase should be considered to start). Then, in L367-368 it is stated "China Sea also shows a diurnal maximum of Flash shifted by about 4 hours before the diurnal maximum of Prec". The use of "also" and the 4-hour figure leads the reader to expect an apples-to-apples comparison, but of course the timing of the maximum in the diurnal cycle is not the same thing as the timing of the onset of the growing phase, and indeed the second half of the sentence – "but the time of the diurnal minimum of Prec and Flash is similar" – clearly shows that the behavior over the China Sea is much different, since the onset of the increasing phases for Flash and Prec coincide. Finally, the delay between the diurnal minimum in Flash and that in Prec over the NAusSea is estimated to be "more than 7 h" (L371), but it looks more like about 9 hours to me.

(5) L373: Flash and Prec increasing phase of convection start at the same time and increase --> the increasing phases of convection for Flash and Prec start at the same time and increase

(6) L376-377: The same comments as above for the differences in the onset of the growing phases of Flash and Prec of 4 h over the Java Sea and 7 h over NAusSea apply here too. In addition, the statement that the increasing phases of Flash and Prec start at the same time over the Bismarck Sea does not reflect the more complicated reality for the minimum in the Flash diurnal cycle discussed in L370-371.

**Section 7:**
(1) L422: Over Java, $\Delta IWC^{Prec}$ is given as 8.7 mg m$^{-3}$ and $\Delta IWC^{Flash}$ as 8.1 mg m$^{-3}$, hence their difference is 0.6, not 0.7 mg m$^{-3}$.

(2) L434: The range of $r^{Prec-Flash}$ summarized here for islands (−6 to −22%) does not include the value for Java (+6%), whereas the range summarized for seas (+6 to −71%) does include the value for Java Sea. I am puzzled by this inconsistency. I think that for both regions either the full range should be represented or the typical range excluding the outliers should be used, but in the latter case the fact that Java / Java Sea are omitted needs to be made clear.

(3) L444: in named --> is named

(4) L477-478: I have a couple of issues with the sentence "Amounts of ice injected deduced from observations and reanalysis show close absolute values over land in the UT and over land and sea in the TL but largely different over sea in the UT." First, it's not clear to me

why the focus here is on "absolute values" when much of the discussion throughout Section 7 emphasizes relative differences (or perhaps they mean absolute values of relative differences). Second, I am not convinced that "show close absolute values" is a supportable statement based on the results in Fig. 11. While the observational and reanalysis ranges do overlap in most (but by no means all) locations except over the seas in the UT, as was noted in preceding sections, fairly large differences between the observational $\Delta$IWC and $\Delta$IWC$^{ERA5}$ are not uncommon, and in some cases even $\Delta$IWC$^{Prec}$ and $\Delta$IWC$^{Flash}$ do not agree particularly well. I think a more appropriate statement here would be: "Amounts of injected ice deduced from observations and reanalysis are fairly consistent over land in the UT and over land and sea in the TL but are inconsistent over sea in the UT."

(5) L479: While it is true that over land "$r^{Total}$ is larger in the TL than in the UT", over sea the upper end of the $r^{Total}$ range in the TL (160%) is not greatly different from that in the UT (156%), according to the values quoted in L473-476 (I tried to check a few of these values by eyeballing Fig. 11 but didn't get the exact values given in these lines, so the authors might want to double-check them again).

(6) L480: "At any considered level" should be "At both considered levels", but to avoid repeating the same phrase at the beginning of two sentences in a row, I suggest that it simply be deleted here.

(7) L481: form --> from

**Section 9:**
(1) L530: (TRMM), the number --> (TRMM), and the number
(2) L554: I think it would be good to emphasize here that the largest $\Delta$IWC occurs over land. I suggest changing "are related to" to "are found over land and are shown to be related to".
(3) L555-556: See comment #2 in Section 7 about the ranges quoted for islands and seas.
(4) L557-558: The possibility that very low values of Flash over sea may contribute to the larger discrepancies between $\Delta$IWC$^{Prec}$ and $\Delta$IWC$^{Flash}$ there was not mentioned when these differences were first discussed in Section 7.1 (e.g., L434-435), so it should be added in those lines too. Also, it might be good to insert "per pixel" after "flashes day$^{-1}$".
(5) L559: difference between $\Delta$IWC estimated from observations and from reanalysis --> differences between $\Delta$IWC estimated from observations and that estimated from reanalysis
(6) L560-561: This sentence ("Among … retained as inconsistent.") doesn't really make sense. I think it would be better to say something like: "In light of these relative differences, $\Delta$IWC estimates from observations and reanalysis are found to be fairly consistent over land in the UT and over land and sea in the TL but inconsistent over sea in the UT."
(7) L568: maximum value of … range --> maximum values of … ranges
(8) L569: and than 0.3 mg m$^{-3}$ --> and more than 0.3 mg m$^{-3}$
(9) L571: evaluated --> fully evaluated; strongest --> largest

---

## Author Response (AR2)

We would like to thanks Dr Michelle Santee and the Anonymous Referee #2 for their review on the revised manuscript. In the present document we answer Anonymous Referee #2 first and then Dr Michelle Santee. Comments from the reviewers are in blue, our replies are in black. Deleted sentences are crossed out and added words or sentences are in bold.

**Replies to Anonymous Referee #2:**

Dion and colleagues have substantially improved the manuscript by taking into account both referees' comments. Most of them have been carefully addressed and clarifications were brought whenever comments were not relevant. The overall text is clearer and straightforward. Thus, I recommend the paper to be accepted to ACP. However, I have few remarks/questions before the paper can be published.
Wu et al. (2008) is not in the corrected reference list.
We added Wu et al. (2008) in the corrected reference list.

(1) About the 100% MSL IWC accuracy cited from Wu et al. (2008): what such a large number means for the present analysis?
(1) The accuracy here means the uncertainty for values less than 10 mg m$^{-3.}$ However, according to Wu et al., (2008) : «Most of the inhomogeneity-induced uncertainties can be reduced through averaging (e.g., in monthly maps) because of randomness of the inhomogeneity.". Thus, in our study, the accuracy in IWC is reduced considering the study zones and the long study period used.
We clarify into the manuscript L 101 as follow:
"The precision of the measurement is 0.10 mg m$^{-3}$ at 146 hPa and 0.25 to 0.35 mg m$^{-3}$ at 100 hPa. **While the accuracy is 100 % for values less than 10 mg m$^{-3}$ at both levels, it is strongly reduce by averaging over the long study period and over the study zones.** The valid range is 0.1-50.0 mg m$^{-3}$ at 146 hPa and 0.02-50.0 mg m$^{-3}$ at 100 hPa (Wu et al., 2008)."

(2) How does the accuracy compared with other satellite-borne estimations?
(2) This accuracy is large compared with other satellite-borne. For instance:
- the SMILES instrument measuring IWC from October 2009 to April 2010 present Systematic Errors in the IWC measurement between 54 and 77 % (from 180 to 80 hPa) (source : https://mls.jpl.nasa.gov/data/smiles.php).
- WV from MLS have accuracy between 9 and 20 % between 146 and 80 hPa (source: https://mls.jpl.nasa.gov/data/v4-2_data_quality_document.pdf, page 75).
However, as specified in Wu et al., 2008 and in the previous answer, this accuracy is, in the present paper, largely reduced through the averaging over the study zone and the study period.

(3) Through the corrected version of the paper, there are a lot of references or figure numbers missing, flagged with a question mark. Please reedit carefully the LaTeX version. This has overcomplicated reading.
(3) Sorry for this mistake, we carefully reedited the re-revised LaTeX version.

**Replies to Dr Michelle Santee:**

**Abstract :**

(5) L17-18: Where do the values (4−29%, 55−78%) quoted in these lines come from? They appear nowhere else in the manuscript. In particular, they do not match any of the numbers given in the relevant discussion in Section 7.2 or 7.3.

(5) The values have been changed consistently with the values recalculated and given in section 7.2 as follow:

« The reanalysis ΔIWC range between $\Delta IWC^{ERA5}$ and $<\Delta IWC^{ERA5}>$ has been also found to be small in the UT **(9 to 33 %)** but large in the TL **(32 to 139 %)**, ... »

(6) L19-21: See point #7 in the comments on Section 7.2 about the phrase "xx% of variability per study zone", which is used four times in these lines.

(6) In the abstract we decided to not show the percentage that does not have strong importance in the results. (Furthermore, the abstract is long enough). We changed the sentence into:

"**Combining observational and reanalysis ΔIWC ranges, the total ΔIWC range is estimated, in the UT, between 4.2 and 10.0 mg m$^{-3}$, over land and between 0.3 and 4.4 mg m$^{-3}$ over sea, and, in the TL, between 0.5 and 3.7 mg m$^{-3}$ over land and between 0.1 and 0.7 mg m$^{-3}$ over sea.**"

(7) L21: See point #4 in the comments on Section 7.3 about these values (0.6 and 3.9 mg m$^{-3}$).

(7) We updated the values (see point #4 of Section 7.3).

(9) L22-25: It is stated that ΔIWC over land is larger than ΔIWC over sea "with a limit at 4.0 mg m$^{-3}$ in the UT". What does "limit" mean in this context? Where does the value of 4.0 mg m$^{-3}$ come from? In Section 7.1, the minimum ΔIWC over land is given as 4.9 mg m$^{-3}$ and the maximum over sea as 4.4 mg m$^{-3}$, leading to a maximum difference of only 0.5 mg m$^{-3}$.

(9) We changed (L23) the sentence by:

« 1) ΔIWC over land has been found to be larger than ΔIWC over sea with a limit around 4.4-4.9 mg m$^{-3}$ in the UT (the limit being defined as the minimum of ΔIWC estimated over land and the maximum of ΔIWC estimated over sea), ... »

(11) L25: It is stated that Java sees the largest ΔIWC in the UT ("7.7 – 9.5 mg m-3 daily mean"). Again, these numbers are not drawn from the main text. Indeed, in Section 7.1, it is stated that "In the UT ... over Java, ΔIWC reaches 7.9–8.7 mg m$^{-3}$".

(11) In this sentence we are talking about the total ΔIWC range. However, we decided in the abstract to name it only "ΔIWC" instead of " total ΔIWC range" in order to make easier the understanding at this stage. The sentence in the abstract has been changed as follow:

« 2) small islands with high topography present the **largest ΔIWC such as the Java Island (with ΔIWC between 7.7 and 9.5 mg m$^{-3}$ in the UT).** »

And a sentence in the Section 7.3 L472) has been added has follow (see the answer Section 7.1 (6) to understand the definition of r$^{Total}$):

"**Java island presents the highest observational and reanalysis ΔIWC range in the UT (between 7.7 and 9.5 mg m$^{-3}$ daily mean, r$^{Total}$ = 21%).**"

(+) Note that to be clearer in the abstract we decided to change some sentences in the new manuscript version as follow (see Section 7.1 point (6) to understand why we change the percentage values):

"... Our study shows that, while the diurnal cycles of Prec and Flash are consistent to each other in timing and phase over land but different over offshore and coastal areas of the MariCont, the observational ΔIWC range between $\Delta IWC^{Prec}$ and $\Delta IWC^{Flash}$**, interpreted as the uncertainty of our model to estimate the ice injected,** is small (they agree to within 6 to 22 % over land and to within 6 to 71 % over ocean) in the UT and TL. The reanalysis ΔIWC range between $\Delta IWC^{ERA5}$ and $<\Delta IWC^{ERA5}>$ has been also found to be small in the UT (9 to 33 %) but large in the TL (32 to 139 %), **highlighting the larger uncertainty in the TL than in the UT due to the vertical resolution of MLS observations.** Considering estimates **of ΔIWC from all methods**, ΔIWC is estimated in the UT between 4.2 and 10.0 mg m$^{-3}$, over land, and between 0.3 and 4.4 mg m$^{-3}$ over sea, and, in the TL, between 0.5 and 3.7 mg m$^{-3}$ over land and between 0.1 and 0.7 mg m$^{-3}$ over sea. Finally, from $IWC^{ERA5}$, Prec and Flash, this study highlights 1) ΔIWC over land has been found t**o be larger than ΔIWC over sea (ΔIWC over land and ocean regions being larger and smaller**

than 4 mg m$^{-3}$, respectively), and 2) small islands with high topography present the **largest ΔIWC such as the Java Island (with ΔIWC between 7.7 and 9.5 mg m$^{-3}$ in the UT). ”**

**Introduction :**
(1) to (6) has been corrected.

**2.1 MLS :**
(8) The sentence in L95-96 (“The horizontal resolution of IWC$_{MLS}$ measurements is ~300 and 7 km along and across the track, respectively.”) is fully redundant with information given below and can be deleted.
(8) We deleted this sentence :
« ~~The horizontal resolution of IWC$^{MLS}$ measurements is ~300 and 7 km along and across the track, respectively.~~ »

**2.2 TRMM 3B42 :**
(2) L111-112: First, “composing” is not the right word in L111. Second, these two sentences seem to be contradictory. The first states that TRMM-3B42 is “a multi-satellite precipitation analysis” but then mentions only GPM. The second states that it is “computed from the various precipitation-relevant satellite passive Microwave (PMW) sensors” and then lists several, not including GPM. These two sentences should be combined and clarified. Third, why is “Microwave” capitalized? It’s part of the acronym, but so is “passive”.
(2) We changed the previous sentence by the following one:
“ ~~TRMM-3B42 is a multi-satellite precipitation analysis composing a Global Precipitation Measurement (GPM) Mission. TRMM-3B42 is computed from the various precipitation-relevant satellite passive Microwave (PMW) sensors using GPROF2017 computed at the Precipitation Processing System (PPS) (e.g., GMI, DPR, Ku, Ka, Special Sensor Microwave Imager/Sounder [SSMIS], etc.) and including TRMM measurements from 1997 to 2015 (Huffman et al., 2007, 2010; and Huffman and Bolvin, 2018).~~ “
« TRMM-3B42 is a multi-satellite precipitation **analysis. The analysis merges microwave and infrared space borne observations and included TRMM measurements from 1997 to 2015 (Huffman et al., 2007, 2010; and Huffman and Bolvin, 2018).** »

(4) L121-122: I am still confused by the description of how 1-hour precipitation data are obtained. As I said in my previous review, I am under the impression (based on Huffman et al. (2007) and other sources) that the TRMM-3B42 product contains gridded merged precipitation estimates with a 3-hour temporal resolution. I had thought that only by taking advantage of the precessing orbit of TRMM and the long study period (13 years) are the authors able to bin the data in 1-hour bins. Perhaps that is what they are getting at with these two sentences, but it is not clear. “The granule temporal coverage of TRMM-3B42 data is 3 hours, but the temporal resolution of individual measurements is 1 minute. Thus, it is statistically possible to degrade the resolution to 1 hour.” seems to be saying that the 1-minute resolution of the individual measurements is somehow preserved in the 3-hourly averaged TRMM-3B42 data. In addition, since they are starting from data with a 3-hour granularity, it is confusing to talk about “degrading” the resolution to 1 hour.
(4) In order to be clearer and to delete all confusing sentences, we modified the sentence as follow:
«  »
« The TRMM-3B42 data has been averaged over a 1-hour interval from 0 to 24 hours. »

**Section 2.3 :**
(5) First it is stated (L130-131) that the “instrument detects lightning with storm-scale resolution of 3−6 km (3 km at nadir, 6 km at limb) over a large region (550−550 km) of the Earth’s surface”. I have no idea what “over a large region (550−550 km) of the Earth’s surface” means. Then in the next sentence (L131) it is stated that “LIS horizontal resolution is provided at 0.25°×0.25°.” Are these two sentences consistent? Finally, it is stated several lines below (L138-139) that “The observation range of the sensor is between 38N and 38S.” The latitudinal coverage and spatial resolution of the LIS data should be described together in sentences that logically flow from one to the next.
(5) : We changed L126:

« over a large region (550~550 km) of the Earth's surface »
 by
« over a large region (550x550 km) of the Earth's surface »

And we moved sentences in the text as follow:
« The observation range of the sensor is between 38ºN and 38ºS.The instrument detects lightning with storm-scale resolution of 3-6 km (3 km at nadir, 6 km at limb) over a large region (550x550 km) of the Earth's surface. **The LIS horizontal resolution is provided** at 0.25º x0.25º. »

(8) First it is stated (L133-134) that "the weak lightning signals that occur during the day are hard to detect because of background illumination". Then it is stated (L134-135) that processing to remove the background signal allows the instrument to "detect weak lightning and achieve a 90% detection efficiency during the day". The next line (L136) states that the "TRMM LIS detection efficiency ranges from 69% near noon to 88% at night", appearing to contradict the previous sentence. Does the detection efficiency during the day really reach as high as 90%, higher than at night?
(8) : The previous sentences were confusing. To be clearer, we changed the paragraph as follow:
"A RTEP removes the background signal to enable the system to detect weak lightning and **achieve a 90\%** detection efficiency during the day. LIS is thus able to provide the number of flashes (Flash) measured. The TRMM LIS detection efficiency ranges from 69\% near noon to 88\% at night."

 « A RTEP removes the background signal to enable the system to detect weak lightning and **improves the** detection efficiency during the day. LIS is thus able to provide the number of flashes (Flash) measured. The TRMM LIS detection efficiency ranges from 69% near noon to 88% at night. »

**Section 2.4**
(6) 169: I'm not sure what information "unitary" is meant to convey here, or why this word is needed. In any case, "an unitary" should be "a unitary".
(6) : The term « unitary » has been deleted in the sentence L165:
« a  box function »

**Section 4.3 :**
(2) In L242, the amount of IWC injected over seas is stated to be < 10 mg m-3. In L244, the largest amount of IWC injected over seas is stated to be 7−15 mg m$^{-3}$, contradicting the first statement.
(2) : We changed in the sentence L242, as follow:
« than over seas (< 10 mg m$^{-3}$). »
« than over seas (< **15** mg m$^{-3}$). »

(10) L267: It is stated that "For pixels with large values of ΔIWC, IWC observed by MLS is between 4.5 and 5.7 mg m-3 over North Australia Sea, South Sumatra and New Guinea". Fig. 5b shows North Australia, not the North Australia Sea. (11) As I pointed out in the original review, the above statement applies only to New Guinea point #1. It is not true for New Guinea point #2, for which the IWC value is much lower.
(10) and (11) : We changed the sentence as follow:
« For pixels with large values of ΔIWC, IWC observed by MLS is between 4.5 and 5.7 mg m$^{3}$ over North Australia, South Sumatra and New Guinea 1. »

(12) For the pixels with low ΔIWC, the IWCMLS ranges from 1.9 to 4.7 mg m-3. Thus the highest IWC for these points is larger than the bottom of the range of the high-ΔIWC points. I note that the revised summary in this section (L269-270) mentions the length of the growing phase of deep convection and the amplitude of the diurnal cycle in Prec as factors associated with large ΔIWC values, but not the magnitude of the IWC values themselves, even though the summary immediately follows discussion of the MLS IWC values. I appreciate that such a statement has been removed in revision. I agree with that deletion and I am NOT suggesting that it be put back in. However, it might strike readers as odd to spend several sentences (L266-269) talking about the MLS IWC values and then end that paragraph with a summary that completely ignores them. The authors should consider adding a sentence or two noting that the IWCMLS ranges overlap for the high and low ΔIWC pixels, and thus no definitive conclusion about the relationship between IWC and

ΔIWC can be drawn.

(12) The sentences of synthesis have been changed as follow :

« To summarize, large values of ΔIWC are observed over land in combination to i) longer growing phase of deep convection (> 9 hours) and/or ii) large diurnal amplitude of Prec (> 0.5 mm h$^{-1}$). **However, as IWC$^{MLS}$ ranges overlap for the high and low ΔIWC, no definitive conclusion about the relationship between IWC and ΔIWC can be drawn.** »

(13) The sentence has been removed.

(14) The number of tickmarks have been changed.

**Section 5.1**

(4) We changed the sentence as follow :

« Differences between Flash and Prec distributions are found over North Australia Sea, with relatively large number of Flash (>10$^{-2}$ flashes day$^{-1}$) compared to low Prec (4 – 10 mm day$^{-1}$) (Fig. 2c), and over **several inland areas of New Guinea** where the number of Flash is relatively low (~10$^{-2}$- 10$^{-3}$ flashes day$^{-1}$) ... »

**Section 5.2**

(3) You are right, it is a mistake to say that pixel in the coastline and in the sea can be mixed. We changed the sentence as follow:

«  »

« At the border between the land and the coast areas, a given 0.25◦×0.25◦ pixel can contain information from both land and coastlines. In that case, we can easily discriminate between land and coastlines by applying the land/coastlines filters. Consequently, this particular pixel will be flagged both as land and coastlines. »

(4) and (5) Figures have been improved: dashed lines have been added into Fig. 7 and minor tickmarks have been added in Figs. 7, 8 and 9.

(8) It is stated (L310-311) that "Prec minimum is around 18:00 LT". It is then asserted (L312- 314) that "These results are consistent with Mori et al. (2004) showing ... a diurnal minimum of precipitation around 11:00 LT". To me, 18:00 LT is not consistent with 11:00 LT.

(8) Indeed, we corrected the description of the Figure 2 in Mori et al. (2004). We changed the sentence as follow L311:

"These results are consistent **with the work of** Mori et al. (2004) showing a diurnal maximum of precipitation in the early morning between 02:00 **and** 03:00 LT and a diurnal minimum of precipitation **between** 11:00 **and 21:00 LT**, over coastal zones of Sumatra."

**Section 6 :**

(1) Both referees questioned whether, given that $IWC^{ERA5}$ is itself unvalidated at this point, $\Delta IWC^{ERA5}$ can really be used to validate the observationally derived values, or might this study rather in some sense serve to use the $\Delta IWC^{Prec}$ and $\Delta IWC^{Flash}$ to validate the new ERA5 values. The authors responded in their reply letter that they are not using $\Delta IWC^{ERA5}$ to validate $\Delta IWC^{Prec}$ and $\Delta IWC^{Flash}$ but to "assess the amounts estimated by our model", and they have added to the manuscript (L384-386) the sentence "The diurnal cycle of IWC over the MariCont from ERA5 will be used to calculate $\Delta IWC$ from ERA5 in order to assess the horizontal distribution and the amount of ice injected in the UT and the TL deduced from our model combining MLS ice and TRMM Prec or MLS ice and LIS flash." To me, there is no difference between "assessing the distribution and amount of" the $\Delta IWC$ deduced from their model and "validating" it. I still feel that it would be appropriate to acknowledge that ERA5 IWC data cannot be considered "truth", their quality has not yet been fully evaluated, and the consistency or lack thereof found in the comparisons between $\Delta IWC^{ERA5}$ and both $\Delta IWC^{Prec}$ and $\Delta IWC^{Flash}$ may have implications for both their methodology and ERA5.

(1) : As you were suggesting, we decided to add the following sentence L386 :

« The ERA5 reanalyses provide hourly IWC at 150 and 100 hPa ($IWC^{ERA5}$). The diurnal cycle of IWC over the MariCont from ERA5 will be used to calculate $\Delta IWC$ from ERA5 in order to **support** the horizontal distribution and the amount of ice injected in the UT and the TL deduced from our model combining MLS ice and TRMM Prec or MLS ice and LIS flash. **Since $IWC^{ERA5}$ data quality has not yet been fully evaluated, this may impact on the consistency or lack thereof found in the comparisons between $\Delta IWC^{ERA5}$ and both $\Delta IWC^{Prec}$ and $\Delta IWC^{Flash}$ may have implications for both our methodology and ERA5.** »

(3) In both L389 and L390, the maximum value reached by $IWC^{ERA5}$ in the UT is stated to be 6.4 mg m-3. But I wonder where the authors are getting that number from. Clearly there are values larger than 6.4 mg m-3 in Fig. 10a (as evidenced by the white patches in the middle of the reddish colors).

(3) We changed « reaching » by « exceeding ».

(7) L408: it is stated that the differences in the timing of the maximum of the diurnal cycle of Prec, Flash and $IWC^{ERA5}$ "do not impact on the calculation of the $\Delta IWC^{Prec}$, $\Delta IWC^{Flash}$ or $\Delta IWC^{ERA5}$." Is the timing unimportant because only the magnitude of the diurnal cycle (max- min) matters for the $\Delta IWC$ calculation? There is also a pervasive lack of superscripting in this sentence.

(7) We changed the sentence as follow exactly as you were suggesting:

"However, these differences do not impact on the calculation of the $\Delta IWC^{Prec}$, $\Delta IWC^{Flash}$ or $\Delta IWC^{ERA5}$, **because only the magnitude of the diurnal cycle (max-min) matters for the $\Delta IWC$ calculation."**

**Section 7.1**

(1) It is stated (L418) that over Java, $\Delta IWC$ "reaches 7.9−8.7 mg m$^{-3}$". This sentence pertains to only two values; thus, I presume that $\Delta IWC^{Prec}$=8.7 and $\Delta IWC^{Flash}$=7.9 mg m$^{-3}$. In that case, $\Delta IWC^{Flash}−\Delta IWC^{Prec}$=0.8 mg m$^{-3}$.

(1) We corrected the sentence as follow:

«  »

« $\Delta IWC^{Flash}$ is generally greater than $\Delta IWC^{Prec}$ by less than **0.8** mg m$^{-3}$ »

(2) The sentence in L419-420 is not complete: "except for Java where is larger than $\Delta IWC^{Flash}$ by 0.7 mg m$^{-3}$ (−8%)".

(2) We completed the sentence as follow:
« ... except for Java where $\Delta\mathbf{IWC^{Prec}}$ is larger than $\Delta\mathrm{IWC^{Flash}}$ by 0.7 mg m$^{-3}$. »

(3) Assuming that the above sentence should read "except for Java where $\Delta\mathrm{IWC^{Prec}}$ is larger than $\Delta\mathrm{IWC^{Flash}}$", then the difference is 0.8 mg m$^{-3}$, not 0.7 as stated.
(3) It has been changed to 0.8 mg m$^{-3}$

(4) Since no sign is attached to the raw difference in L420, why is the percent difference negative? It doesn't make sense to say that "$\Delta\mathrm{IWC^{Prec}}$ is larger than $\Delta\mathrm{IWCFlash}$ by −8%".
(4) The sign has been deleted L425.

(5) Furthermore, the percent difference as defined here should be 0.8/7.9=10%, not 8% as stated (L420).
(5) It has been corrected to 10%.

(6) More fundamentally, why did the authors choose to use $\Delta\mathrm{IWC^{Flash}}$ in the denominator for the percent differences? $\Delta\mathrm{IWC^{Flash}}$ is no more "correct" than $\Delta\mathrm{IWC^{Prec}}$. Thus, a perhaps less arbitrary approach would have been to use the average of the two $\Delta\mathrm{IWC}$ estimates in the denominator rather than picking one of them.
(6) As you were suggesting we recalculate all the percentage values, and renamed its as follow:

→ Equation has been added L421 in the new manuscript version :
$r^{Prec-Flash} = ((\Delta\mathrm{IWC^{Prec}}-\Delta\mathrm{IWC^{Flash}})/((\Delta\mathrm{IWC^{Prec}}+ \Delta\mathrm{IWC^{Flash}})/2)\times100$

→ L449:
$r^{ERA5-\langle\Delta\mathrm{IWCERA5}\rangle} = ((\Delta\mathrm{IWC^{ERA5}}- \langle\Delta\mathrm{IWC^{ERA5}}\rangle)/((\Delta\mathrm{IWC^{ERA5}}+\langle\Delta\mathrm{IWC^{ERA5}}\rangle)/2)\times100$,

→ L466: " The consistency between observational and reanalysis $\Delta\mathrm{IWC}$ range is calculated as the difference between the minimal value of the largest range minus the maximum value of the lowest range divided by the mean of these two values. In the UT, over land, observational and reanalysis $\Delta\mathrm{IWC}$ are found consistent to within 0 to 25% while over sea they are inconsistent (to within 62 to 96%) in the UT."

" In the following we will consider rTotal as the relative differences between the minimal value of the lower range minus the maximum value of the largest range divided by the mean of these two values. The range between observational and reanalysis ranges is named the total IWC range, and is estimated in the UT between 4.2 and 10.0 mg m$^{-3}$ ($r^{Total}$ from 8 to 59%) over land ..."

(7) L422: "$\Delta\mathrm{IWC^{Flash}}$ is almost twice as large as than $\Delta\mathrm{IWC^{Prec}}$ (53%)" – to me it looks like $\Delta\mathrm{IWC^{Flash}}$ is ~4.3 mg m$^{-3}$, whereas $\Delta\mathrm{IWCPrec}$ is ~2 mg m$^{-3}$ (which does work out to ~53% by the percent difference definition used here), so that $\Delta\mathrm{IWC^{Flash}}$ is more than twice as large as $\Delta\mathrm{IWC^{Prec}}$, not almost twice as large. Also note: large as than --> large as.
(7) It was a mistake in the sentence. The sentence has been changed as suggested and the percentage value has been recalculated as a function of the comments Section 7.2 point #(7). We changed the sentence as follow (L518 of the new manuscript):
"Additionally, Fig. 11 shows that the strongest differences between $\Delta\mathrm{IWC^{Prec}}$ and $\Delta\mathrm{IWC^{Flash}}$ are found over the North Australia Sea, with $\Delta\mathrm{IWC^{Flash}}$ **greater than** $\Delta\mathbf{IWC^{Prec}}$ **by 2.3 mg m$^{-3}$ in the UT ($r^{Prec-Flash} = $~71%) and by 0.4 mg m$^{-3}$ in the TL ($r^{Prec-Flash} = $- 75%)."**

(8) The lack of any tickmarks whatsoever on the right-hand y-axes of Fig. 11 makes it quite difficult for readers to judge any of the values quoted in the text for themselves.
(8) The figure has been completed as suggested.

**Section 7.2**
(5) L439: The first sentence on this line ("Consistently ... respectively.") is redundant and unnecessary. It should be deleted.
(5) We deleted the sentence.

(7) L444: The phrase "xx% of variability per study zone", used here and in eight other places in Sections 7.2 and 7.3 as well as four times in the Abstract, makes no sense to me. I believe that the authors wish to quantify the range of the relative differences between $\Delta IWC^{ERA5}$ and $\langle\Delta IWC^{ERA5}\rangle$ in the various study zones (that is, the smallest and the largest difference between the convolved and unconvolved $\Delta IWC$ estimates from the 5 islands for the island zone and similarly from the 5 seas for the sea zone), but "xx% of variability" means something else and is not the way to convey that information. If my understanding of what the authors are intending is correct, then perhaps something like this would work: "with relative differences between $\Delta IWC^{ERA5}$ and $\langle\Delta IWC^{ERA5}\rangle$ of ~19-22% over the island study zone".

(7) We changed the phrase "xx% of variability per study zone" in the section 7.2, 7.3 and 9 in order to be clearer on what has been considered in the percentages. Our answer here, details also the answer to the question Section 7.1 point #(6). (Differences between the previous manuscript version and the new version are also clearly highlighted in the Supplement document provided.)

- In the Section 7.1, from L417:

[revised manuscript text omitted]

(9) L449: Again, without tickmarks on the right-hand axis it is hard to tell, but the value for New Guinea looks higher than 3.7 mg m$^{-3}$ to me, more like 3.9. (10) L449: same comment as above for "xx% of variability per study zone".
(9) It was a mistake. The percentage does not change. We corrected the sentence as follow:
"In the TL, over land, $\Delta IWC^{ERA5}_{100}$ and $<\Delta IWC^{ERA5}_{100}>$ vary from 0.5 to **3.9** mg m$^{-3}$ …"

(11) L450: It is stated that 〈$\Delta IWC^{ERA5}$〉 is larger than $\Delta IWC^{ERA5}$ "by less than 2.1 mg m$^{-3}$". To me the difference between the two estimates looks more like 2.4 mg m$^{-3}$ for Sulawesi and 2.5 mg m$^{-3}$ for Java. Thus the statement "by as much as 2.5 mg m$^{-3}$" would be more accurate. (12) L451: same comment as above for "xx% of variability per study zone".
(11) We changed the value by 2.11 in the sentence as follow:
« ...being larger than $\Delta IWC^{ERA5}$ by less than **2.5** mg m$^{-3}$. »

(13) L451-452: Again, the value of 0.2 mg m$^{-3}$ should be checked here, as it looks as though it might be larger than that for the Java and North Australia seas, and "by as much as" would be better than "by less than".
(13) We have changed as suggested:
« Over sea, $\Delta IWC^{ERA5}$ and $<\Delta IWC^{ERA5}>$ vary from 0.05 to 0.4 mg m$^{-3}$ ($r^{ERA5}$ = ~71 %) with $\Delta IWC^{ERA5}$ lower than $<\Delta IWC^{ERA5}>$ by **as much as** 0.2 mg m$^{-3}$. »

**Section 7.3**
(4) L462: I note that the authors' response letter indicates that the values 0.6 and 3.9 mg m$^3$ in this line have been changed to 0.5 and 3.7 mg m$^3$. However, such changes have not actually been made in the revised text.
(4) It was a mistake. We corrected by the true values of 0.5 and 3.7 mg m$^3$

(9) L466: It is stated that significant differences are found over all individual offshore study zones "within 0.7 to 2.1 mg m$^{-3}$". But differences for the sea zone were given as "1.0 – 2.2 mg m$^{-3}$" on L459. (And again "as much as" would be better than "within".)
(9) All the sentence has been deleted. See answer Section 7.3 #(10).

(10) We deleted the sentence L464:

""
and sentences in the Section 7.3 synthesis has been changed. See below the answer Section 7.3 point #(7).

(11) L470-471: Based on the reanalysis ΔIWC range, it is suggested that Sulawesi and New Guinea may also reach high ΔIWC values comparable to those over Java, even though the observationally derived ΔIWC estimates do not indicate such strong injections over those two islands. But might it not also be the case that the reanalysis might be in error in those regions? Perhaps they pose particularly challenging environments for the models to get right. Again, it seems to me that the authors see the comparisons with ERA5 only as a means of validating (or assessing) their model, and not as a two-way street that possibly highlights regions of potential issues in the reanalysis.

(11) : We completed the sentence L477 in order to consider the eventual error from the reanalysis, as follow:

**"However, whatever the level considered, although Java has shown particularly high values in the observational ΔIWC range compared to other study zones, the reanalysis ΔIWC range shows that Sulawesi and New Guinea would also be able to reach similar high values of ΔIWC as Java (assuming that ERA5 IWC data have not yet been evaluated). "**

**Section 8.2**
(1) L505-508: There is a pervasive lack of superscripting in these lines.

(1) Yes, we were refering to about Mori et al. (2004). The sentences have been clarified as follow:

« The diurnal cycle of stratiform and convective precipitations over West Sumatra Sea has been studied by Mori et al. (2004) using 3 years of TRMM precipitation radar (PR) datasets, following the 2A23 Algorithm (Awaka, 1998). **Mori et al. (2004)** have shown that rainfall over Sumatra is characterized by convective activity with a diurnal maximum between 15:00 and 22:00 LT while ... »

(3) L508: same comment as above for "xx% of variability".

(3) This sentence has been change and detailed in Section 7.1 point #(7) (see L517 in the new manuscript version).

**Section 9**
(1) L523-525: It struck me that the summary in the Conclusions does not mention that a key assumption in the model is that deep convection is in the growing phase. Since many readers may only skim the paper and focus mostly on the Conclusions, and many may also be unfamiliar with the earlier Dion et al. (2019) paper, the authors should make clear that they are only applying their model during the increasing phase of the diurnal cycle of deep convection.

(1) : The sentences L523-525 have been modified as follow:

« ...troposphere (UT) and the tropopause level (TL) over the MariCont, from the method proposed in a companion paper (Dion et al., 2019).  While Dion et al. (2019) have... »

« ...troposphere (UT) and the tropopause level (TL) over the MariCont, from the method proposed in a companion paper (Dion et al., 2019). **The study is focused on the austral convective season of DJF from 2004 to 2017. In the model used (Dion et al., 2019), Prec is considered as a proxy of deep convection impacting ice (ΔIWC^Prec) in the UT and the TL. ΔIWC^Prec is firstly calculated by the correlation between the growing phase of the diurnal cycle of Prec from TRMM (obtained at a 1-hour resolution**

**diurnal cycle) and the value of IWC measured by MLS (IWC$^{MLS}$ , provided at the temporal resolution of 2 observations in local time per day) selected among the growing phase of the diurnal cycle of Prec.** While Dion et al. (2019) have ... »

(7) L546-547: It is stated that "the observational $\Delta$IWC range has been shown to be consistent with the reanalysis $\Delta$IWC range to within 23 % over land and to within 30−50 % over sea in the UT and to within 49% over land and to within 39% over sea in the TL". It is not clear where these numbers are derived from, as none of them have appeared previously in the manuscript.

(7) These values have been deleted by error. Thus, we added back these values in to the section 7.3 Synthesis, L469 as follow (note that the percentages values have been changed considering the new calculation of the percentage as defined in the sentence):

**« The consistency between observational and reanalysis $\Delta$IWC range is calculated as the difference between the minimal value of the largest range minus the maximum value of the lowest range divided by the mean of these two values. In the UT, over land, observational and reanalysis $\Delta$IWC are found consistent to within 0 to 25% while over sea they are inconsistent (to within 62 to 96%) in the UT. In the TL, observational and reanalysis $\Delta$IWC ranges are consistent to within 0 to 49% over land and to within 0 to 28% over sea. »**

These values has been changed in the conclusion section 9 as follow:

**"Finally, $\Delta$IWC estimated from observations has been shown to be consistent with $\Delta$IWC estimated from reanalysis to within 25% over land in the UT, to within 49 % over land in the TL and to within 28 % over sea in the TL, but inconsistent to within 96 % over sea in the UT."**

(11) L551-552: Although the differences between $\Delta$IWC$^{ERA5}$ and 〈$\Delta$IWC$^{ERA5}$〉 at the two levels suggest that the vertical resolution of the observations has a stronger impact in the TL than in the UT, the total $\Delta$IWC variation range being discussed in this sentence does not. Short comings in the methodology, Prec, or Flash could all contribute to the total $\Delta$IWC variation range.

(11) To be clearer the confusing sentence L551-552 has been deleted.

(12) L554-556: Are the values quoted in these lines (0–0.6, 1.0, 0.3 mg m$^3$) consistent with the corresponding numbers given in Section 7 (I don't think so)?

(12) Firstly, we corrected the consistencies by adding the following sentence in Section 7.1, (L432 in the new manuscript version):

«To summarize, independently of the proxies used for the calculation of $\Delta$IWC, and at both altitudes, Java island shows the largest injection of ice over the MariCont. **Observational $\Delta$IWC over Java island is larger by about 1.0 mg m$^{-3}$ in the UT and about 0.3 mg m$^{-3}$ in the TL than other land study zones.** Furthermore, it has been shown ...»

Then, we correct the consistencies by correcting the values (L566 in the new manuscript version) in Section 9 as follow:

« The study at small scale over islands and seas of the MariCont has shown that $\Delta$IWC from ERA5, Prec and Flash in the UT agree to within **0.1 – 1.0** mg m$^{-3}$ over MariCont_L, Sumatra, Borneo and Java with the largest values obtained over Java Island. »

(14) Concerning the sentence discussed above, see point #11 in the comments on Section 7.3.

(14) As previously, we changed the text as follow (L555):

" The study at small scale over islands and seas of the MariCont has shown that $\Delta$IWC from ERA5, Prec and Flash in the UT agree to within 0.1 – 1.0 mg m$^{-3}$ over MariCont_L, Sumatra, Borneo and Java with the largest values obtained over Java Island. **Based on observations,** the Java Island presents the largest amount of ice in the UT and the TL (larger by about 1.0 mg m$^{-3}$ in the UT and about 0.3 mg m$^{-3}$ in the TL than other land study zones). **Based on the reanalysis,** New Guinea and Sulawesi reach similar ranges of ice injection in the UT and even larger ranges of values in the TL than the Java Island keeping in mind that ERA5 IWC

data have not yet been evaluated. »

**References**
We carefully checked all references in the list.

**Supplement**
All changes in the manuscript have been highlighted below.

[revised manuscript text omitted]

---

## Author Response (AR3)

**Review of Second Revision of "Ice injected into the tropopause by deep convection – Part 2:**
**Over the Maritime Continent" by Dion et al., October 2020.**

We would like to sincerely thank Dr. Michelle Santee for the consequent work done during the proofreading of this study. We would also like to apologize for any inconvenience caused by the errors and omissions left in the latest version of the manuscript.

Our corrections and answers are detailed below. Reviewer comments are copied in blue. Our answers are in black. Main changes in the text are presented below in bold.

**Abstract:**
(1) L6-7: ice injected ($\Delta$IWC) up to the TL by combining ice water content (IWC) measured twice a day in local time in tropical UT and TL by --> ice injected ($\Delta$IWC) up to the tropical UT and TL by combining ice water content (IWC) measured twice a day by
(2) L8: (Prec) measurement --> (Prec) measurements
(3) L12: resolutions --> resolution
(4) L14-21: These sentences are confusing and hard to read. I recommend re-writing as: Our study shows that the diurnal cycles of Prec and Flash are consistent with each other in timing and phase over land but different over offshore and coastal areas of the MariCont. The observational $\Delta$IWC range between $\Delta$IWCPrec and $\Delta$IWCFlash, interpreted as the uncertainty of our model in estimating the amount of ice injected, is smaller over land (where the two estimates agree to within –6 to –22 %) than over ocean (where relative differences are +6 to –71 %) in the UT and TL. The impact of the MLS vertical resolution on the estimation of $\Delta$IWC is greater in the TL (differences between $\Delta$IWCERA5 and $\langle\Delta$IWCERA5$\rangle$ of 32 to 139%) than in the UT (difference of 9 to 33%).
Considering all the methods, in the UT, estimates of $\Delta$IWC span 4.2 to 10.0 mg m-3 over land and 0.3 to 4.4 mg m-3 over sea, and, in the TL, estimates of $\Delta$IWC span 0.5 to 3.7 mg m-3 over land and 0.1 to 0.7 mg m-3 over sea.
(1)-(4) : We changed the text according to your suggestions.

(5) In the above, are the values of 0.3 (min of the range in the UT over sea) and 3.7 (max of the range in the TL over land) correct? From Fig. 11, to me these values look more like 0.4 and 3.9, respectively.
(5): We have corrected the values to 0.4 and 3.9, respectively.

(6) L21-23: First, the statement that $\Delta$IWC is smaller than 4 mg m$^{-3}$ over sea directly contradicts the previous sentence, where the top of the range over the sea in the UT is correctly stated to be 4.4 mg m$^{-3}$. Second, it is not clear that these numbers apply only to the UT, not the TL. I recommend instead using the wording in Section 7.3 (L480): "At both levels, $\Delta$IWC estimated over land is more than twice that estimated over sea."
(7) L23: present the largest $\Delta$IWC such as the Java Island (7.7 to 9.5 mg m-3 in the UT) → present the largest $\Delta$IWC (e.g., Java Island, with values of 7.7 to 9.5 mg m-3 in the UT)
(6) and (7): L21-23 of the previous manuscript has been changed by the following sentence L21-23 of the new manuscript as suggested :

"Finally, based on IWC from MLS and ERA5, Prec and Flash, this study highlights that 1) **at both levels, $\Delta$IWC estimated over land is more than twice that estimated over sea,** and 2) small islands with high topography **present the largest $\Delta$IWC (e.g., Java Island, with values of 7.7 to 9.5 mg m$^{-3}$ in the UT).**"

**Section 1:**
(1) L43: "twice daily in local times" – I do not think that the addition of "in local times" here is

helpful, so it should either be deleted or changed to "twice daily (at 01:30 and 13:30 local time)"

(1): L43 of the previous manuscript has been changed as suggested (see L45 of the new manuscript version).

**Section 2.1:**

(1) The authors have confused accuracy (systematic error) and precision (random noise). Precision is generally improved by averaging; accuracy is not. That is, the precision of an average of N profiles is 1/sqrt(N) times the precision of an individual profile. Since their analysis involves averaging in both space and time, the precision (measurement noise) of the MLS IWC data is of essentially no consequence for this study. But, contrary to what has been written here, such averaging does nothing to mitigate the 100% systematic uncertainty (accuracy) of the IWC measurements. Referee #2 asked what the implications of the large (100%) uncertainty in the MLS IWC data are for this analysis. The authors have failed to address this point correctly in their revised manuscript.

(1) : We corrected and detailed our explanations regarding the accuracy and the precision of MLS at the L101-110 of the new manuscript. We explain that by applying temporal (monthly/seasonally/yearly) and geographical averages, we can considerably lower the random error on the averages. Systematic error on the averages, of the order of 100% on each individual retrieval of $IWC^{MLS}$, will of course be unchanged. For that reason, our analysis, based on the methodology developed in Dion et al. (2019), relies on a differential method to highlight the amplitude of the diurnal cycle of IWC that is expected to be the amount of ice injected in the TL and/or the UT. By considering the difference between the maximum and the minimum of IWC obtained within 24 hours, the associated systematic error dramatically decreases. This supposes that the systematic errors are of the same order of magnitude within each temporal bin within 24 hours.

(2) This section was heavily edited in revision, but unfortunately the changes do not represent an improvement. The overall flow is poor, and the repetitiveness and seemingly random arrangement of sentences (with multiple instances of unrelated points being interposed between sentences that should have been connected) make it hard to follow. The wording is also incorrect in places (besides the accuracy issue), and some quoted values are wrong.

(3) To address the above comment (2), I recommend re-ordering / re-writing this paragraph as: The Microwave Limb Sounder (MLS) was launched on NASA's Earth Observing System Aura platform in 2004 (Waters et al., 2006). MLS follows a sun-synchronous near-polar orbit, obtaining daily global coverage. Ascending (northbound) portions of the orbit cross the equator at 13:30 local time (LT); descending portions of the orbit cross the equator at 01:30 LT. Among other products, MLS provides measurements of ice water content (IWCMLS, mg m-3 ). Although optimal estimation is used to retrieve almost all other MLS products, a cloud-induced radiance technique is used to derive IWCMLS (Wu et al., 2008, 2009). Here we use version 4.2 IWC data, filtered following the recommendations of the MLS team described by Livesey et al. (2018). We select IWCMLS during all austral convective seasons DJF between 2004 and 2017. MLS data processing provides IWCMLS at 6 levels in the UTLS (82, 100, 121, 146, 177 and 215 hPa). We have chosen to study only two levels: an upper and a lower level of the TTL. Because the level at 82 hPa does not provide enough significant measurements of IWC to achieve good signal-to noise, we have selected 100 hPa as the upper level of the TTL (named TL, for tropopause level) and 146 hPa as the lower level of the TTL (named UT, for upper troposphere). The resolution of IWCMLS (horizontal along the path, horizontal perpendicular to the path, vertical) measured at 146 and 100 hPa is 300×7×4 km and 200×7×5 km, respectively. In our study, high horizontal resolution is now possible because we consider 13 years of MLS data, allowing the IWCMLS measurements to be averaged in bins with 2°×2° (~230 km2 ) horizontal resolution. Typical single-profile precisions are 0.08− 0.18 mg m-3 at

146 hPa and 0.20–0.65 mg m-3 at 100 hPa, and the accuracy is 100% for values less than 10 mg m-3 at both levels. The valid IWC range is 0.1–50.0 mg m-3 at 146 hPa and 0.02–50.0 mg m-3 at 100 hPa (Livesey et al., 2018).
(4) Note that my suggested re-writing of this section does not address the concern about the accuracy of the MLS IWC measurements raised by Reviewer #2, which I leave to the authors to answer.
(1) to (4): We changed the whole paragraph as you were suggesting in (3). Furthermore, we modified the level definitions (L 94-96 of the new manuscript) and we added the following sentence in bold to answer the question regarding the impact of the accuracy at 100% in our study (see L101 of the new manuscript version):

From L87 of the new manuscript :
"The Microwave Limb Sounder (MLS) was launched on NASA's Earth Observing System Aura platform in 2004 (Waters et al., 2006). MLS follows a sun-synchronous near-polar orbit, obtaining daily global coverage. Ascending (northbound) portions of the orbit cross the equator at 13:30 local time (LT); descending portions of the orbit cross the equator at 01:30 LT. Among other products, MLS provides measurements of ice water content (IWC$^{MLS}$, mg m$^{-3}$). Although optimal estimation is used to retrieve almost all other MLS products, a cloud-induced radiance technique is used to derive IWC$^{MLS}$ (Wu et al., 2008, 2009). Here we use version 4.2 IWC data, filtered following the recommendations of the MLS team described by Livesey et al. (2018). We select IWC$^{MLS}$ during all austral convective seasons DJF between 2004 and 2017. MLS data processing provides IWC$^{MLS}$ at 6 levels in the UTLS (82, 100, 121, 146, 177 and 215 hPa). **We have chosen to study only two of the available levels: 146 hPa as representative of the lower part of the TTL (named UT for upper troposphere) and 100 hPa as representative of tropopause which lies in the middle of the TTL (named TL for tropopause level). Note that the level at 82hPa, representing the Lower Stratosphere, would have been also very interesting to study but do not provide enough significant measurements of IWC to achieve acceptable signal-to noise ratio.**
The resolution of IWC$^{MLS}$ (horizontal along the path, horizontal perpendicular to the path, vertical) measured at 146 and 100 hPa is 300×7×4 km and 200×7×5 km, respectively. **In our study, we consider 13 years of MLS data, which allows the IWC$^{MLS}$ measurements to be averaged in bins of 2° ( ~220 km) zonal and meridional extent, over all study zones. The valid IWC range is 0.02–50.0 mg m$^{-3}$ at 100 hPa and 0.1–50.0 mg m$^{-3}$ at 146 hPa (Livesey et al., 2018). Typical single-profile precisions (i.e. random noise) are 0.10 mg m$^{-3}$ at 100 hPa and 0.20–0.35 mg m$^{-3}$ at 146 hPa, and the accuracy (i.e. systematic error) is 100 % for values less than 10 mg m$^{-3}$ at both levels. The fact that our study is based on 13-year averages of all observations within each 2° x 2° bin implies that the uncertainty on the averages due to measurement precision is drastically reduced. On the other hand, the systematic error on the averages will be unchanged. But our analysis, based on the methodology developed in Dion et al. (2019), relies on a differential method to highlight the amplitude of the diurnal cycle of IWC that is expected to be the amount of ice injected in the TL and/or the UT. By considering the difference between the maximum and the minimum of IWC obtained within 24 hours, the associated systematic error dramatically decreases. This supposes that the systematic errors are of the same order of magnitude within each temporal bin within 24 hours.**"

**Section 2.2:**

(1) The organization of this section is also awkward, with a sentence about the Prec product not differentiating between stratiform and convective precipitation coming in between two sentences about horizontal resolution and binning, then a couple sentences about averaging in time, followed by a sentence pointing back to the spatial binning methodology. As I stated in previous reviews, the authors should arrange this description in a more logical manner that steps through all related points

before moving on to other aspects.

(2) L115: averaged over a 1-hour interval --> averaged over 1-hour intervals
(3) I still think it will not be clear to all readers how this 1-hr resolution for Prec is achieved. As noted in my previous reviews, the authors are able to take advantage of the precessing orbit of the TRMM satellite and the long (13-yr) study period to bin the data into 1-hr bins. They have now included a sentence to this effect in the LIS description (L132-133), and I think it would be helpful to include something along those lines here as well.
(4) L117: is provided --> are provided
(1) to (4): We re-organized the paragraph as follow (L.112 of the new manuscript):

"The Tropical Rainfall Measurement Mission (TRMM) was launched in 1997 and provided measurements of precipitation until 2015.  The TRMM satellite carried five instruments, three of which (PR, TMI, VIRS) formed a complementary sensor suite for rainfall. TRMM had an almost circular orbit at 350 km altitude performing a complete revolution in one and a half hour. The 3B42 algorithm product (TRMM-3B42) (version V7) is a multi-satellite precipitation analysis, created to estimate the precipitation and extends the precipitation product through 2019. The analysis merges microwave and infrared spaceborne observations and included TRMM measurements from 1997 to 2015 (Huffman et al., 2007, 2010; Huffman and Bolvin, 2018). Precipitation from TRMM-3B42 (Prec) are provided at a 0.25° (~ 29.2 km ) horizontal resolution, extending from 50° S to 50° N (https://pmm.nasa.gov/data-access/downloads/trmm, last access: April 2019). Details of the binning methodology of TRMM-3B42 are provided by Huffman and Bolvin (2018).  The precipitation estimates do not distinguish between stratiform and convective precipitation and the implications of this will be discussed later. Work is currently underway with NASA funding to develop  more appropriate estimators for random error, and to introduce estimates of bias error (Huffman and Bolvin, 2018). In our study, Prec from TRMM-3B42 was selected over the austral convective seasons (DJF) from 2004 to 2017 and at each location was binned into 1-hour intervals according to local time (LT). This was possible because of the combinaison between the precessing orbit of the TRMM satellite and the precipitation analysis from the other satellites included into TRMM-3B42 long duration (13 years)."

(3) I still think it will not be clear to all readers how this 1-hr resolution for Prec is achieved. As noted in my previous reviews, the authors are able to take advantage of the precessing orbit of the TRMM satellite and the long (13-yr) study period to bin the data into 1-hr bins. They have now included a sentence to this effect in the LIS description (L132-133), and I think it would be helpful to include something along those lines here as well.
(3): We added the following sentence L123 of the new manuscript version:

"In our study, Prec from TRMM-3B42 was selected over the austral convective seasons (DJF) from 2004 to 2017 and at each location was binned into 1-hour intervals according to local time (LT). This was possible because of the combinaison between the **precessing orbit** of the TRMM satellite and the precipitation analysis from the other satellites included into TRMM-3B42 long duration (13 years)."

**Section 2.3:**

(1) L119: aboard of --> aboard
(2) L121: pixel representing --> pixel, representing
(1)-(2): We changed the text according to your suggestions.

(3) L123-125: Confusing aspects of the LIS description previously mentioned have not been rectified in the revised manuscript. It is stated that: "The instrument detects lightning with storm-scale resolution of 3-6 km (3 km at nadir, 6 km at limb) over a large region (550x550 km) of the

Earth's surface. The LIS horizontal resolution is provided at 0.25°x0.25°." Are these two sentences consistent with one another?

(3): The horizontal resolution is not given at 0.25° × 0.25°, but we binned the LIS measurements at this resolution. Thus, the sentence L124 of the previous manuscript has been deleted. The sentence L133 has been kept (see L136 of the new manuscript) as follow:

L124 of the previous manuscrit is deleted: ""

L136 of the new manuscript: **"The measurements could be further binned at either 0.25°x 0.25° or at 2°x 2° horizontal resolution to allow comparison with  Prec from TRMM-3B42."**

(5) L133-134: "In our study, Flash measured by LIS is binned at 0.25°x0.25° horizontal resolution to be compared to Prec from TRMM-3B42." As stated in L125, 0.25°x0.25° is the LIS native resolution. I assume that 2°x2° is meant here.

(5): LIS has been binned at 0.25° × 0.25° horizontal resolution in Figure 6 in order to be easily compared to Prec from TRMM-3B42 in Figure 2. However, LIS has been also binned at 2° × 2° horizontal resolution in order to be compared to Prec at  2° × 2° horizontal resolution and in order to calculate IWC$^{Flash}$.

We changed the sentence L133 of the previous manuscript into the following sentence (see L136 of the new manuscript):

"**The measurements could be further binned at either 0.25°x 0.25° or at 2°x 2° horizontal resolution to allow comparison with  Prec from TRMM-3B42.**"

**Section 4.1:**

(1) L188: associated to --> associated with

(2) L193: instead of fixing "NewGuinea", it was deleted: (e.g. over ) --> (e.g. over New Guinea)

Corrections have been done as suggested.

**Section 5.2:**

(1) Once again, the order of the panels in Fig. 7 is mischaracterized. This error had been fixed in the previous revision, but the figure has now been redrawn so it has reappeared in this draft. Consequently, references to Fig. 7 in L290, L307, L315, and L362 are all wrong, as is the figure caption.

(1): We corrected the order of the panels in Figure 7 a, b and c.

(2) L297-299: I still find the wording in these sentences contradictory and confusing. "At the border between the land and the coast areas, a given 0.25°x0.25° pixel can contain information from both land and coastlines. In that case, we can easily discriminate between land and coastlines by applying the land/coastlines filters. Consequently, this particular pixel will be flagged both as land and coastlines." If in fact you could easily discriminate between land and coastlines, then you would not need to "double count" these pixels by placing them in both categories. Isn't it because they cannot be easily differentiated that they need to be flagged as being in both regimes?

(2): We made an error in the explanation here. The distinction between MariCont_L and MariCont_C does not need a special flag. We deleted the entire sentence as follow L297-299 of the previous manuscript:

"MariCont_L is the area of all land pixels. "

(3) L304: Why does this sentence start with "Consequently"? That word does not seem appropriate to me here; perhaps "Nonetheless" might be better, or nothing.

(3): We deleted « Consequently » L304 of the previous manuscript (L303 of the new manuscript).

**Section 5.3:**

(1) L335: value --> values

(2) L346: of altitude --> altitude

(3) L348-349: air masses cooled in altitude are transported to the sea favoring the dissipating stage of the convection. Sulawesi is also a small island with high topography as Java --> air masses cooled at higher altitudes are transported to the sea, favoring the dissipating stage of the convection. Like Java, Sulawesi is a small island with high topography.

(4) L356: over tropical land --> over broad tropical land regions

(5) L367: instead of fixing the spelling of "Bismark Sea", it was deleted: NAusSea, Sea and WSumSea --> NAusSea, Bismarck Sea and WSumSea

(6) L373: over the Sea --> over the Bismarck Sea

(1) to (6): Corrections have been done as suggested.

**Section 6:**

(1) L386-388: This wording is unclear and awkward. I suggest: "In assessing the consistency or lack thereof in the comparisons between $\Delta IWC^{ERA5}$ and both $\Delta IWC^{Prec}$ and $\Delta IWC^{Flash}$, it should be kept in mind that $IWC^{ERA5}$ data quality has not yet been fully evaluated."

(1): The sentence has been changed by the one that you proposed (see L384 of the new manuscript).

(2) L390: New Guinea where --> New Guinea, where

(3) L410: impact on --> affect

(2) and (3): Corrections have been done as suggested.

**Section 7.1:**

(1) Unless I missed it, nowhere in this section is it stated that the range between $\Delta IWC^{Prec}$ and $\Delta IWC^{Flash}$ is quantified as a means of characterizing the uncertainty in their model. Such a statement is made in the Abstract (L15-16), and I think it would be good to explicitly note it here (e.g., in L419, observational upper and lower bounds), as well as in the Conclusions.

(1): We completed the sentence L419 of the previous manuscript as follow (L416 of the new manuscript):

"The observational $\Delta IWC$ range calculated between $\Delta IWC^{Prec}$ and $\Delta IWC^{Flash}$ **provides a quantitative characterisation of the uncertainty** in our model. In the following we will consider ..."

We also complemented the L547 of the conclusion section as follow (L546 of the new manuscript version):

"$\Delta IWC$ calculated by using Flash as a proxy of deep convection ($\Delta IWC^{Flash}$) is compared to $\Delta IWC^{Prec}$ over five islands and five seas of the MariCont. **Over each study zone, the range of values between $\Delta IWC^{Prec}$ and $\Delta IWC^{Flash}$, the observational $\Delta IWC$ range, allows us to characterize the uncertainty of our model.**"

(2) L424: (with $r_{Prec–Flash}$ ranges from - 6 to - 22% over the study zone) --> (with $r_{Prec–Flash}$ ranging from $-6$ to $-22\%$ over the study zones)

(2) Corrections have been done as suggested.

(3) L425: Of course, I did not check all of the arithmetic in this section, but I recommend that the authors do so. According to Eqn. (4) and the values given in L423, for Java $r^{Prec-Flash} = 100 * [ (8.7-7.9) / 0.5*(8.7+7.9) ] = 9.6\%$, not 7.1% as stated here.

(3): We double checked all the calculations. The error for Java came from the values presented L423 of the previous manuscript. 7.9 should have been 8.1. Thus the calculation of $r^{Prec-Flash}$ becomes: $r^{Prec-Flash} = 100 \times [ (8.7-\mathbf{8.1}) / 0.5*(8.7+\mathbf{8.1}) ] = 7.1 \%$ as stated here. (However, to be consistent with the others percentages values in the text, we will replace « 7.1% » by « 7% » to be in integer value). Thus, we corrected the sentence L425 of the previous manuscript as follow (see L420 of the new manuscript version):

"In the UT (Fig. 11a), over islands, $\Delta$IWC calculated over Sumatra, Borneo, Sulawesi and New Guinea varies from 4.9 to 7.1 mg m$^{-3}$ whereas, over Java, $\Delta$IWC reaches $\mathbf{8.1}$–8.7 mg m$^{-3}$. $\Delta$IWC$^{Flash}$ is generally greater than $\Delta$IWC$^{Prec}$ by less than 1.4 mg m$^{-3}$ (with $r^{Prec-Flash}$ ranging from -6 to -22% over the study zones) for all the islands, except for Java where $\Delta$IWC$^{Prec}$ is larger than $\Delta$IWC$^{Flash}$ by $\mathbf{0.7}$ mg m$^{-3}$ ($r^{Prec-Flash} = \mathbf{7}$ %)."

(4) L426: To me it looks as though $\Delta$IWC$_{Flash}$ is greater than $\Delta$IWC$_{Prec}$ by more like 2.3 mg m$^{-3}$ over the NAS, not 2.1 mg m$^{-3}$ (the max difference stated here).

(4): This is a mistake. We double checked all the values over the sea study zones and we corrected the value L426 of the previous manuscript to 2.3 mg m$^{-3}$ (see L424 of the new manuscript).

(5) L428: are --> is
(6) L431: UT with --> UT, with
(5) and (6) Corrections have been done as suggested.

(7) L433-434: What is the statement "Observational $\Delta$IWC over Java island is larger by about 1.0 mg m$^{-3}$ in the UT and about 0.3 mg m$^{-3}$ in the TL than other land study zones" based on? Do these values represent averages of the $\Delta$IWC$^{Prec}$ and $\Delta$IWC$^{Flash}$ estimates for Java vs. averages of the $\Delta$IWC$^{Prec}$ and $\Delta$IWC$^{Flash}$ estimates for all of the other islands? Or are the authors just comparing the bottom end of the estimate range for Java with the top end of the range for all of the other islands? Certainly, the estimates for Java exceed those for some of the other islands (e.g., Sumatra) by much more than 1.0 mg m$^{-3}$ in the UT. A similar question pertains to the value of 0.3 mg m$^{-3}$ in the TL.

(7): We clarified the sentence L433 of the previous manuscript as follow (see L430 of the new manuscript):

"To summarize, independently of the proxies used for the calculation of $\Delta$IWC, and for both UT and TL, Java island shows the largest injection of ice over the MariCont. **The minimum value of the** observational $\Delta$IWC range over Java island **is larger than the maximum value of the observational $\Delta$IWC range of other land study zones by more than 1.0 mg m$^{-3}$ in the UT and more than 0.3 mg m$^{-3}$ in the TL.**"

**Section 7.2:**

(1) L446: The ice injected from ERA5 at $z_0$ --> The ERA5 amount of ice injected at $z_0$
(2) L447: we can consider --> we consider
(3) L458: larger than $\Delta$IWC$_{ERA5}$ by less than 2.5 mg m$_{-3}$ --> larger than $\Delta$IWC$_{ERA5}$ by as much as
(1)-(3) : We changed the text according to your suggestions.

(4) L459: To me it seems that the difference between $\Delta$IWC$^{ERA5}$ and 〈$\Delta$IWC$^{ERA5}$〉 might be as large as 0.3 mg m$^{-3}$ for the Java and North Australian Seas, not 0.2 as stated here.
(4): The indicated values has been corrected to 0.3 mg m$^{-3}$ (L459 of the new manuscript).

**Section 7.3:**

(1) L465: range --> ranges

(2) L467: greater than the reanalysis by ~1.0–2.2 mg m$_{-3}$, showing a systematic larger estimate derived from observation than derived from reanalysis --> greater than that of the reanalysis by ~1.0–2.2 mg m$_{-3}$, with systematically larger estimates derived from observations than from the reanalysis

(3) L468-472: The description of the quantification of the "consistency" between the observational and reanalysis ΔIWC estimates remains confusing and poorly written. For one thing, it is presented in such a way that small values (0–25%) indicate that the two are consistent and large values (96%) indicate that they are inconsistent, which seems counterintuitive. In addition, the wording "the difference between x minus y" is incorrect, and several other wording issues and grammar errors make these sentences hard to understand. I recommend re-writing L468-472 as:

The consistency between observational and reanalysis ΔIWC ranges is calculated as the minimum value of the higher range minus the maximum value of the lower range divided by the mean of these two values. In the UT, observational and reanalysis ΔIWC estimates are found to be consistent over land, where the relative differences between their ranges are less than 25%, but inconsistent over sea, where differences are 62–96%. In the TL, the relative differences between the observational and reanalysis ΔIWC ranges are 0–49% over land and 0–28% over sea.

(4) L472-476: The description of r$_{Total}$ is also quite unclear and badly written. Moreover, as originally defined, r$_{Total}$ would always be a negative number, but the values quoted for it are not negative. I recommend re-writing as:

In the following, we define the total range covering the observational and reanalysis ΔIWC estimates, r$_{Total}$, as the maximum value of the higher range minus the minimum value of the lower range divided by the mean of these two values. In the UT, the observational and reanalysis ΔIWC estimates span 4.2 to 10.0 mg m$_{-3}$ (with r$_{Total}$ values from 8 to 59%) over land and 0.3 to 4.4 mg m$_{-3}$ (with r$_{Total}$ values from 104 to 149%) over sea. In the TL, the observational and reanalysis ΔIWC estimates span 0.5 to 3.7 mg m$_{-3}$ (with r$_{Total}$ values from 85 to 127%) over land and 0.1 to 0.7 mg m$_{-3}$ (with r$_{Total}$ values of 142 to 160%) over sea.

(1)-(4) : We changed the text according to your suggestions.

(5) L476: Are the values of 0.3 mg m$^{-3}$ for the bottom of the ΔIWC range over sea in the UT and 3.7 mg m$^{-3}$ for the top of the ΔIWC range over land in the TL correct? To me, they look more like 0.4 and 3.9 mg m$^{-3}$, respectively (as also noted in connection with the abstract).

(5): Values have been corrected to 0.4 and 3.9 mg m$^{-3}$, respectively (L20, L474-475 and L563 of the new manuscript).

(6) L478-479: Amounts of ice injected deduced from observations and reanalysis are consistent to each other over land in the UT and over land and sea in the TL (to within 0 to 49%) but inconsistent over sea in the UT (up to 96%) → Amounts of ice injected deduced from observations and reanalysis are consistent (i.e., the relative differences between their respective ranges are less than 49%) over land in the UT and over land and sea in the TL but inconsistent over sea in the UT (where differences are as large as 96%)

(7) L478-479: This is backwards! the impact of the vertical resolution on the estimation of ΔIWC is much larger in the UT than in the TL → the impact of the vertical resolution on the estimation of ΔIWC is much larger in the TL than in the UT

(6) and (7) We changed the text according to your suggestions.

(8) L481: The statement that "Java island presents the highest observational and reanalysis ΔIWC

range in the UT (between 7.7 and 9.5 mg m$^{-3}$ daily mean)" is misleading – at first I interpreted it to be saying that Java shows the largest *range* of observational and
reanalysis $\Delta$IWC estimates (which, according to Fig. 11, is not true: that would be New Guinea, with values from ~5.6 to 10.0 mg m$^{-3}$). I think the authors mean that the estimated $\Delta$IWC *values* for Java are larger than for other islands, but that is the case only for the observational estimates, not $\Delta$IWC$^{ERA5}$. Also, what is meant by "daily mean" here?

(8): We deleted the incriminated sentence L481 from the previous manuscript and changed the sentence L482 by the sentence L481 of the new manuscript as follow:

" At any considered level, although Java has shown **the largest** values of $\Delta$IWC from observations compared to other study zones, the reanalysis $\Delta$IWC range shows that Sulawesi and New Guinea may also reach high values of $\Delta$IWC, similar to those seen over Java. However, as the ERA5 IWC data have yet to be extensively validated, it is also possible that the reanalysis overestimates IWC in these regions."

**Section 8.1:**

(1) L494: impacts on the diurnal cycle of Prec and on the IWC --> impacts the diurnal cycle of Prec and the IWC
(2) L495: delete "and" at the end of this line
(3) L498-499: cumulus merging processes which are processes more important --> cumulus merging processes, which are more important
(4) L501: IWC is increasing proportionally with Prec consistent --> IWC increases proportionally with Prec, consistent
(5) L502-503: add commas after "(2019)" and "(Fig. 3)"
(1) to (5): Corrections have been done.

**Section 8.2:**

(1) L508: precipitations --> precipitation
(2) L515-516: the calculation of $\Delta$IWC estimated from Prec is possibly overestimated because Prec include --> $\Delta$IWC calculated from Prec is possibly overestimated because Prec includes
(1) to (2): Corrections have been done.

**Section 8.3:**

(1) L522: ~71% --> ~-71%
(2) L523: large highlighting the difficulty to estimate --> large, highlighting the difficulty of estimating
(1) to (2): Corrections have been done.

**Section 9:**
(1) L537: binned at a 1-hour diurnal cycle --> binned at 1-hour resolution over the diurnal cycle
(2) L538: selected among --> during
(1)-(2) : We changed the text according to your suggestions.

(3) L555-556: (a) I think that "disagree" or "deviate from one another" would be better than "depart". (b) "from" and "to" should be "by". (c) −6% over sea should be +6%. (d) If the sign of these relative differences is specified, then the fact that Prec is usually smaller needs to be made clear. (e) largest --> larger. Thus, taking these issues into account, I recommend that these lines be

re-written as: "ΔIWCPrec is typically smaller than ΔIWCFlash, with the two estimates disagreeing by −6 to −22% over land and +6 to −71% over sea. The larger …"

(3): The sentence L555-556 of the previous manuscript has been changed as suggested L555 of the new manuscript.

(4) L561: inconsistent to within 96 % over sea in the UT. Thus, thanks to the combination → inconsistent over sea in the UT, where relative differences are as large as 96%. Thus, considering the combination

(5) L563: 0.3 might be 0.4 and 3.7 might be 3.9, as mentioned earlier

(6) L564: found higher --> found to be greater

(7) L567-568: Java with … the Java Island --> Java, with … Java Island

(4) to (7): We changed the text according to your suggestions.

(8) L568-569: See comment #6 in Section 7.1.

(8): As for comment (7) Section 7.1, we changed the sentence L586-589 of the previous manuscript to the sentence L568 as follow:

[revised manuscript text omitted]

---

## Author Response (AR4)

**Review of Second Revision of "Ice injected into the tropopause by deep convection – Part 2: Over the Maritime Continent" by Dion et al., 5 November 2020.**

We would like to thank again Dr. Michelle Santee for the consequent work done during all the proofreading processes of this study. Our corrections and answers are detailed below. Reviewer comments are copied in blue. Our answers are in black. Main changes in the text are presented below in bold.

**Abstract:**

(1) L1-3: Results presented in a companion paper (Part I) have used ... and shown --> A companion paper (Part 1) used ... and showed
(2) L9: "binned" would be better than "averaged" here.
(1)-(2) : We changed the text according to your suggestions.

(3) L17: First, I find the term "absolute relative differences" confusing. Second, stating absolute values for the relative differences is inconsistent with what was done in the rest of the paper and is somewhat misleading. Third, see comment #2 in Section 7 about how these ranges are specified. Please enact changes consistently throughout the manuscript.
(3) : We change the text in the abstract L.17 of the previous version by the following text (L.15-17 of the new version) :
« The observational ΔIWC range between $\Delta IWC^{Prec}$ and $\Delta IWC^{Flash}$, interpreted as the uncertainty of our model in estimating the amount of ice injected, is smaller over land (**where the two relative differences between $\Delta IWC^{Prec}$ and $\Delta IWC^{Flash}$ agree to within +7 to -22%**) than over ocean (**where relative differences are between +6 and -71%**) in the UT and TL. »

(4) L19: It would be good to add ", depending on the study zone" after "139%".
(4) : We changed the text according to your suggestions.

**Section 1:**

(1) L36: With the edits made to the beginning of this paragraph, "CPT" is no longer defined. (2) L75: an other --> another
(1) : We changed the text according to your suggestions.

**Section 2:**

(1) L96: of tropopause which --> of the tropopause, which
(2) L97: Lower Stratosphere ... do --> lower stratosphere ... does
(3) L115-117: The awkward grammar and the redundancy in these sentences make them hard to read and confusing. I suggest instead: "The TRMM-3B42 product (version V7) is a multi- satellite precipitation analysis that extends the precipitation product through 2019 by merging microwave and infrared spaceborne observations, including TRMM measurements from 1997 to 2015."
(4) L118: are provided --> is provided
(5) L124-126: In addition to some typos/grammar issues, this sentence gives the impression that the TRMM-3B42 data record is only 13 years long, which is not the case. I suggest re- writing as: "This was possible because of the combination of the precessing orbit of the TRMM satellite and the availability of precipitation estimates from the other satellites included in the TRMM-3B42 analysis during our 13-year study period."
(1)-(5) : We changed the text according to your suggestions.

(6): L130-134: Although it is much clearer in this draft, I remain confused by aspects of the LIS description (e.g., how the 550 × 550 km region mentioned relates to the stated 3–6 km resolution of the LIS measurements), so I went to download the Christian et al. [2000] reference (LIS ATBD). The URL given in the reference section (L619-620) does not seem to point to an active site. However, I was able to obtain the LIS ATBD from
https://eospso.nasa.gov/sites/default/files/atbd/atbd-lis-01.pdf.
(7) I looked through the LIS ATBD only briefly and did not thoroughly read it, but I note that many of the values given in these lines (e.g., resolution at nadir and limb, detection efficiencies at noon and at night, latitude range) do not appear in that document. Clearly the authors have relied on another source of LIS information besides the referenced ATBD. If there is a published paper that contains relevant information, it should be cited here.
(6)-(7) : We updated the link provided in the reference list by the following one :
https://eospso.gsfc.nasa.gov/sites/default/files/atbd/atbd-lis-01.pdf, and remove the previous link (http://thunder. msfc. nasa. 620 gov/bookshelf/pubs/atbd/. 2000). Furthermore, we added the following link in the text as reference of the information provided:
https://ghrc.nsstc.nasa.gov/lightning/overview_lis_instrument.html.
Thus, we verified and corrected the paragraph presenting TRMM-LIS as follow (L127 of the new manuscript version):

« The Lightning Imaging Sensor (LIS) aboard of the TRMM satellite measures several parameters related to lightning, including the number of flashes within a given time period. **Details are given in Christian et al. (2000), and, more recently, in the NASA website (https://ghrc.nsstc.nasa.gov/lightning/overview_lis_instrument.html), including how the raw measurements are processed to estimate the number of flashes (Flash), subject to a** detection efficiency of the instrument of 69% at noon to 88% at night (lower during the day because of background illumination). The instrument detects lightning with storm-scale resolution of **5-10 km (5 km at nadir), and the observation range of the sensor is between 38◦ N and 38◦ S. The LIS on TRMM views a total area exceeding 580 km × 580 km at the cloud top.** The LIS instruments obtained measurements between 1 January 1998 and 8 April 2015. To be consistent with the other parts of our study, we used the measurements only for DJF from 2004-2015. As LIS is on the TRMM platform, the measurements can be binned in 1-hour intervals of LT to obtain a full 24-hr diurnal cycle. The measurements could be further binned at either 0.25◦ × 0.25◦ or at 2◦ × 2 ◦ horizontal resolution to allow comparison with Prec from TRMM-3B42. »

(8) L148: I'm not sure exactly what is meant by the ERA5 "process" – data assimilation system?
(8): Indeed, « ERA5 "process" » would mean "data assimilation system". To be clearer, we changed "process"in the text L.148 of the new version manuscript by  "data assimilation system".

(9) L154: radiances data --> radiance data (or, radiances data --> radiances)
(10) L157: Delete "by".
(11) L163: Delete "and in the present study"; also, add "profiles" or "data" or something similar between "IWC$_{ERA5}$" and "have been degraded".
(12) L164: the MLS vertical resolution of IWC$_{MLS}$ --> the vertical resolution of IWC$_{MLS}$
(9)-(12) : We changed the text according to your suggestions.

**Section 4:**
(1) L207: between Prec low values (4–8 mm day$^{-1}$) and IWC$_{MLS}$ large concentrations (4–7 mg m$^{-3}$) --> between low values of Prec (4–8 mm day$^{-1}$) and large values of IWC$_{MLS}$ (4–7 mg m$^{-3}$)
(2) L210: Delete the comma after "analysis".
(3) L211-213: These two sentences ("From ...pixel.") are fully redundant with the newly added sentence in L209-211 and should be deleted.

(4)  L216: with Prec value is --> with Prec values
(5)  L217: on the contrary --> in contrast
(6)  L236: section 2.4 --> section 3
(7)  L251: Figure --> figure
(8)  L256-257: It seems to me that the sentence about the low value of ΔIWC over the sea is out of place here. I think it would go better at the end of the previous paragraph, since what is currently the last sentence of that paragraph also discusses pixels with low values of ΔIWC.
(9)  L258: This sentence is missing essential information. I think the authors mean "when Δ**IWC** is large" and "when Δ**IWC** is small" ("small" is a more appropriate word than "weak").
(10) L259: Since this sentence begins "More precisely", the implication is that it will elaborate further on the immediately preceding discussion. But this is not the case – the previous sentence talks about the duration of the increasing phase of the Prec diurnal cycle, whereas this sentence is about its amplitude. I would delete "More precisely".
(11) L262: IWC$_{MLS}$ ... are --> IWC$_{MLS}$ ... is
(1)-(11) : We changed the text according to your suggestions.

**Section 5:**

(1) L277: Flash takes --> Flash has (or, Flash takes --> Flash is characterized by)
(1): We changed the text according to your suggestions.

(2)  L300: The times when Flash and Prec reach their maxima are not the same as those given in the previous section (L284-285). I realize that the earlier estimates are based on Figs. 6b and 2d, whereas the numbers here are from the broader averages of Fig. 7, but it is still a bit confusing for the reader, so some words of clarification would help.
(2) : The sentence L. 284-285 of the previous manuscript version has been changed to be consistent with the range of values observed later in Fig. 7. Thus the range of value observed in Fig. 6b and Fig. 2d,  include the range of value observed later Fig.7.
L. 280 of the new manuscript version:

«Over land, the **maxima of Flash are mainly observed in the range 15:00-18:00 LT, while maxima of Prec (Fig. 2d) are mainly observed over a longer range of maxima from 18:00 to 24:00 LT.** Coastal regions show similar hours of maximum of Prec and Flash, i.e between 00:00 LT and 04:00 LT although, over the West Sumatra Coast, diurnal maxima of both Prec and Flash happen 1–4 hours earlier (from 23:00-24:00 LT) than those of other coasts. »

We change L296 as follow :
« Over land, during the growing phase of the convection, Prec and Flash start to increase at the same time (10:00 LT – 12:00 LT) but Flash **reaches a maximum earlier (from 15:00 LT) than Prec (from 18:00 LT), consistently with the range of maximum hours observed Fig. 2d and Fig. 6b.**

(3)  L352-353: In addition to a couple of other minor wording issues in this sentence, I don't think it is correct to characterize the decreasing phase of the diurnal cycle as "decreasing more rapidly" for Flash than for Prec. To address all of the issues, I suggest something along these lines: "However, because Flash is observed only in deep convective clouds, the decreasing phase of the Flash diurnal cycle is shorter than the decreasing phase of Prec."
(3) : We changed the text according to your suggestions.

(4)  L358-372: The discussion in this paragraph is confusing. The sentence starting in L360 notes that the Java Sea shows the largest diurnal maxima in Prec and Flash. The sentence starting in L364 marks the contrast with NAusSea, Bismarck Sea, and WSumSea, which display diurnal cycles with small amplitudes. It is then stated (L366-367) that "Java Sea and WSumSea present a similar

diurnal cycle of Prec and Flash, with Flash growing phase starting about 4 h earlier than that of Prec." This seems to contradict the earlier statement contrasting the diurnal cycles of Java Sea and WSumSea. Perhaps WSumSea should be omitted from the list of regions with weak diurnal cycles. Moreover, while it is true that the difference in the timing of the onset of the growing phases of Flash and Prec is about 4 h for WSumSea, this sentence implies a similar timing difference for the Java Sea, but that is not what Fig. 9a seems to indicate (although the lengthy "plateau" in the diurnal cycle of Flash between about 10 and 18 LT makes it difficult to judge exactly when its growing phase should be considered to start).

We decided to suppress the following sentence comparing Jasa Sea and WsumSea L366 of the previous manuscript version: « Java Sea and WSumSea present a similar diurnal cycle of Prec and Flash, with Flash growing phase starting about 4 h earlier than that of Prec. »

Then, in L367-368 it is stated "China Sea also shows a diurnal maximum of Flash shifted by about 4 hours before the diurnal maximum of Prec". The use of "also" and the 4-hour figure leads the reader to expect an apples-to-apples comparison, but of course the timing of the maximum in the diurnal cycle is not the same thing as the timing of the onset of the growing phase, and indeed the second half of the sentence – "but the time of the diurnal minimum of Prec and Flash is similar" – clearly shows that the behavior over the China Sea is much different, since the onset of the increasing phases for Flash and Prec coincide.

We suppressed the « also » in the sentence L 367-368 of the previous manuscript version.

Finally, the delay between the diurnal minimum in Flash and that in Prec over the NAusSea is estimated to be "more than 7 h" (L371), but it looks more like about 9 hours to me.

We changed « 7 » by « **9** » in the sentence L 371 of the previous manuscript version.

(5) L373: Flash and Prec increasing phase of convection start at the same time and increase --> the increasing phases of convection for Flash and Prec start at the same time and increase

(5) : We changed the text according to your suggestions.

(6) L376-377: The same comments as above for the differences in the onset of the growing phases of Flash and Prec of 4 h over the Java Sea and 7 h over NAusSea apply here too. In addition, the statement that the increasing phases of Flash and Prec start at the same time over the Bismarck Sea does not reflect the more complicated reality for the minimum in the Flash diurnal cycle discussed in L370-371.

(6) : We correct the sentence as follow L.372 of the new manuscript version:

« **The diurnal cycles of Flash and Prec have shown some time lag as** over North Australia Sea where the diurnal cycle of Flash is more than **9 hours ahead of the diurnal cycle of Prec.**»

The following sentence has been suppressed :

« . China Sea and Bismarck Sea present the same time of the onset of the Flash and Prec increasing phase »

**Section 7:**

(1) L422: Over Java, $\Delta IWC_{Prec}$ is given as 8.7 mg m$^{-3}$ and $\Delta IWC_{Flash}$ as 8.1 mg m$^{-3}$, hence their difference is 0.6, not 0.7 mg m$^{-3}$.

(1) : We changed 0.7 by **0.6**, L.417 of the new manuscript version.

(2) L434: The range of $r_{Prec-Flash}$ summarized here for islands (−6 to −22%) does not include the value for Java (+6%), whereas the range summarized for seas (+6 to −71%) does include the value for Java Sea. I am puzzled by this inconsistency. I think that for both regions either the full range should be represented or the typical range excluding the outliers should be used, but in the latter

case the fact that Java / Java Sea are omitted needs to be made clear.

(2) : We changed the values L.434 of the previous manuscript version by the values including Java island as follow L.27-429 of the new version :

« ….: $\Delta IWC^{Prec}$ and $\Delta IWC^{Flash}$ are more consistent to each other, both in the UT and in the TL, over islands (relative difference $r^{Prec-Flash}$ = +7 to -22 %) than over seas ( $r^{Prec-Flash}$ = +6 to -71 %). »

(3) L444: in named --> is named

(3) : We changed the text according to your suggestions.

(4) L477-478: I have a couple of issues with the sentence "Amounts of ice injected deduced from observations and reanalysis show close absolute values over land in the UT and over land and sea in the TL but largely different over sea in the UT." First, it's not clear to me why the focus here is on "absolute values" when much of the discussion throughout Section 7 emphasizes relative differences (or perhaps they mean absolute values of relative differences). Second, I am not convinced that "show close absolute values" is a supportable statement based on the results in Fig. 11. While the observational and reanalysis ranges do overlap in most (but by no means all) locations except over the seas in the UT, as was noted in preceding sections, fairly large differences between the observational $\Delta IWC$ and $\Delta IWC_{ERA5}$ are not uncommon, and in some cases even $\Delta IWC_{Prec}$ and $\Delta IWC_{Flash}$ do not agree particularly well. I think a more appropriate statement here would be: "Amounts of injected ice deduced from observations and reanalysis are fairly consistent over land in the UT and over land and sea in the TL but are inconsistent over sea in the UT."

(4): We changed the text according to your suggestions.

(5) L479: While it is true that over land "$r_{Total}$ is larger in the TL than in the UT", over sea the upper end of the $r_{Total}$ range in the TL (160%) is not greatly different from that in the UT (156%), according to the values quoted in L473-476 (I tried to check a few of these values by eyeballing Fig. 11 but didn't get the exact values given in these lines, so the authors might want to double-check them again).

(5): We double-checked all the calculations of $r^{Total}$ and the values presented previously are correct. We change the sentence L.479 of the previous manuscript version by the following one, L.474 of the new manuscript version :

« However, the impact of the vertical resolution on the estimation of $\Delta IWC$ is much larger in the TL than in the UT ($r^{Total}$ is larger in the TL than in the UT) over land, but similar over sea. »

(6) L480: "At any considered level" should be "At both considered levels", but to avoid repeating the same phrase at the beginning of two sentences in a row, I suggest that it simply be deleted here.

(6): We deleted the begining of the sentence starting by "At any considered level" L.480 of the previous manuscript version, as suggested.

(7) L481: form --> from

(6)-(7): We changed the text according to your suggestions.

**Section 9:**

(1) L530: (TRMM), the number --> (TRMM), and the number

(2) L554: I think it would be good to emphasize here that the largest $\Delta IWC$ occurs over land. I suggest changing "are related to" to "are found over land and are shown to be related to".

(1)-(2): We changed the text according to your suggestions.

(3) L555-556: See comment #2 in Section 7 about the ranges quoted for islands and seas.

(3): We change the sentence L.555 of the previous manuscript version by the following sentence

L.550-552 of the new manuscript version:

« $\Delta IWC^{Prec}$ is typically smaller than $\Delta IWC^{Flash}$, with the two estimates differing up to $-22$ % over land and up to $-71$% over sea, **excepted for Java and Java Sea, where the two estimates differ by +7 and +6 %, respectively.** »

(4) L557-558: The possibility that very low values of Flash over sea may contribute to the larger discrepancies between $\Delta IWC_{Prec}$ and $\Delta IWC_{Flash}$ there was not mentioned when these differences were first discussed in Section 7.1 (e.g., L434-435), so it should be added in those lines too. Also, it might be good to insert "per pixel" after "flashes day$^{-1}$".

(4). We added "per pixel" after "flashes day$^{-1}$" L.554 of the new manuscript version.

We added the information about the very low values of Flash in the text L. 434 of the previous manuscript version, as follow, L.429-431of the new manuscript version:

« The larger difference over seas is probably due to the larger contribution from stratiform precipitation to Prec over sea and to the very low values of Flash over seas ($<10^{-2}$ flashes day$^{-1}$ per pixel). »

(5) L559: difference between $\Delta IWC$ estimated from observations and from reanalysis --> differences between $\Delta IWC$ estimated from observations and that estimated from reanalysis

(6) L560-561: This sentence ("Among ... retained as inconsistent.") doesn't really make sense. I think it would be better to say something like: "In light of these relative differences, $\Delta IWC$ estimates from observations and reanalysis are found to be fairly consistent over land in the UT and over land and sea in the TL but inconsistent over sea in the UT."

(7) L568: maximum value of ... range --> maximum values of ... ranges

(8) L569: and than 0.3 mg m$_{-3}$ --> and more than 0.3 mg m$_{-3}$

(9) L571: evaluated --> fully evaluated; strongest --> largest

(5)-(9) : We changed the text according to your suggestions.

We did other small changes :

L. 30, we changed "up to" by "through".

L. 42, we changed "is focusing" by "focuses".

L.57, we changed "those authors" by "Yang and Slingo (2001)" to be clearer.

L.85 we changed the sentence "This section presents the instruments and the reanalyses used for this study." by " This section presents the **observational and reanalysis datasets** used for this study".

L. 89 we changed the sentence "the elevation" by "the elevation from Solar Radiation Data (SoDa, http ...".

L.255 we changed "null", by "zero".

L. 289, we delated the url kink already given previously, and changed by "SoDa elevation filter".

[revised manuscript text omitted]

---

## Author Response (AR5)

**Review of Second Revision of "Ice injected into the tropopause by deep convection – Part 2: Over the Maritime Continent" by Dion et al., 26 November 2020.**

We would like to thank the Editor Geraint Vaughan for its decision and its comments. The manuscript was revised based on his suggestions, as detailed below. Editor comments are copied in blue. Our answers are in black. Main changes in the text are presented below in bold.

Abstract line 15
"The observational ΔIWC range between ΔIWCPrec and ΔIWCFlash, interpreted as the uncertainty of our model in estimating the amount of ice injected, is smaller over land (where the two relative differences between ΔIWCPrec and ΔIWCFlash agree to within +7 to -22%) than over ocean (where relative differences are between +6 and -71%) in the UT and TL."
What does this mean? Should the brackets not read "(where ΔIWCPrec and ΔIWCFlash agree to within +7 to -22%)" and "(where differences are between +6 and -71%)"? What does the range +7 to -22% (and +6 to -71%) refer to? Do you actually mean "(where ΔIWCPrec and ΔIWCFlash agree to within 22%)" and "(where differences are up to 71%)"? This is what I understand from l. 577.

The sentence has been changed as follows:

"The observational  IWC range between  IWCPrec and  IWCFlash, interpreted as the uncertainty of our model in estimating the amount of ice injected, is smaller over land **(where IWCPrec and  IWCFlash agree to within 22%)** than over ocean **(where differences are up to 71%)** in the UT and TL."

l. 132 'on the NASA website' (not 'in'), also give last access to website

The sentence has been corrected and the last access to website has been added as follows:

"Details are given in Christian et al. (2000), and, more recently, **on** the NASA website (https://ghrc.nsstc.nasa.govlightning/overview_lis_instrument.html, **last access: November 2020**), including ..."

Fig 5 caption change to
Location of 2°x2° pixels in Fig.4 where a) ΔIWC > 15 mg m-3 and f) 5 > ΔIWC > 2 mg m-3. Diurnal cycle of Prec (solid line) over 4 pixels (in Fig.4) where ΔIWC > 15 mg m-3 (b, c, d, e) and 5 > ΔIWC > 2 mg m-3 (g, h, i, j), during DJF 2004-2017. The diamond represents IWCMLS during the increasing phase of the convection. The dashed line is the diurnal cycle of IWC estimated from the diurnal cycle of Prec and from IWCMLS.

The caption has been changed as suggested.

l.294 characterised (British spelling)
l.317 consistent (not consistently)
l.318 observed in
l.385 The China Sea

The words have been respectively changed as suggested.

[revised manuscript text omitted]